# Generalization and Scaling Laws for Mixture-of-Experts Transformers

**Mansour ZOUBEIROU A MAYAKI** [* 1]

## Abstract

We develop a theory of generalization and scaling for Mixture-of-Experts (MoE) Transformers that cleanly separates *active* per-input capacity from routing combinatorics. By conditioning on fixed routing patterns and union-bounding across them, we derive a sup-norm covering-number bound whose metric entropy scales with the active parameter budget and incurs a MoE-specific routing overhead. Combined with a standard ERM analysis for squared loss, this yields a generalization bound under a $d$-dimensional manifold data model and $C^\beta$ targets, showing that approximation and estimation trade off as in dense networks once active parameters are accounted for appropriately. We further prove a constructive approximation theorem for MoE architectures, showing that, under the approximation construction, error can decrease either by scaling active capacity or by increasing the number of experts, depending on the dominant bottleneck. From these results we derive neural scaling laws for model size, data size, and compute-optimal tradeoffs. Overall, our results provide a transparent statistical reference point for reasoning about MoE scaling, clarifying which behaviors are certified by worst-case theory and which must arise from data-dependent routing structure or optimization dynamics.

## 1. Introduction

Mixture-of-Experts (MoE) Transformers enable *conditional computation*: for each token, only a small subset of experts is activated, so the number of *active* parameters per input is much smaller than the *total* number of parameters in the model. This yields a practical advantage: compute scales primarily with the number of active experts, while the full expert pool still provides substantial capacity for specializa-tion. Despite striking empirical gains, a theory that cleanly separates the benefits of active capacity from the overheads induced by routing has been missing. In particular, existing scaling laws for dense models track error as a function of total parameters, dataset size, and compute, whereas MoE models require a more refined accounting: (i) approximation should scale with the *active* parameter budget rather than the total parameter count; (ii) generalization should reflect the multiplicity of routing patterns; and (iii) compute-optimal training should couple data and *active* capacity rather than data and total parameters. In this work, we develop approximation and uniform generalization results for MoE Transformers that make these distinctions explicit.

Classical bounds indexed by the ambient dimension $D$ predict unrealistically slow rates for high-dimensional inputs, whereas empirical scaling for Transformers is much faster. A growing body of theory and evidence shows that when data concentrate on a $d$-dimensional manifold, both approximation and estimation rates and hence the resulting scaling exponents are governed by the *intrinsic* dimension $d$ rather than $D$. Recent work ([Havrilla & Liao](#), [2024](#)) develops approximation and statistical theories for Transformers on intrinsically low-dimensional data, deriving data- and model-scaling exponents that depend explicitly on $d$ and validating these predictions on language-modeling benchmarks. Motivated by these results, we adopt $d$ as the geometric control parameter throughout: it aligns the theory more closely with practice, explains faster-than-$1/D$ behavior, and yields MoE scaling predictions that are sensitive to data geometry.

Our results should be interpreted as a conservative statistical baseline for MoE models rather than as tight predictions of training dynamics. Where empirical behavior is sharper, it must arise from data-dependent routing stability, specialization, or architectural biases not captured by worst-case analysis. Uniform generalization bounds are especially relevant for emerging architectures such as MoE, where empirical regularities are still evolving: they clarify which scaling behaviors are statistically certified independently of optimization and which require richer, data-dependent explanations.

The main theoretical contributions are:

- **Metric-entropy decomposition for MoE Transformers.** We establish a sup-norm covering-number bound in

[1]Université Lumière Lyon 2, LIRIS UMR 5205 F-69676, Lyon, France. Correspondence to: Mansour MAYAKI <mansour.mayaki at liris.cnrs.fr>.

*Proceedings of the 43rd International Conference on Machine Learning*, Seoul, South Korea. PMLR 306, 2026. Copyright 2026 by the author(s).

which the complexity of Mixture-of-Experts Transformers decomposes additively into an *active-parameter* term and a *routing* combinatorial term, thereby isolating the statistical effect of conditional computation.

- **Approximation and uniform generalization under intrinsic dimension.** Under a $d$-dimensional manifold model and $C^\beta$ targets, we derive approximation and uniform generalization bounds that separate approximation, estimation, and routing contributions, with rates governed by active capacity.
- **Neural scaling laws for sparse architectures.** We recover dense-network scaling exponents in $(d, \beta)$ when measured against the *active* parameter budget, including model, data, and compute-optimal scaling. Routing contributes an additional logarithmic term in the worst-case bound, while specialization can yield stronger empirical gains in practice.

## 2. Related Work

**Transformers: expressivity, approximation, and generalization.** The Transformer architecture (Vaswani et al., 2017) implements content-addressable computation through multi-head self-attention and positionwise MLPs. Formal expressivity results show that (suitably parameterized) Transformers are universal approximators for a broad class of sequence-to-sequence maps (Yun et al., 2020), with positional encodings playing a crucial role in breaking permutation symmetries and enabling length-dependent behaviors. Subsequent analyses refine which architectural ingredients control approximation power number of heads, attention bandwidth, depth vs. width trade-offs, and the effect of pre-norm vs. post-norm residual designs often revealing that depth amplifies compositional expressivity while LayerNorm stabilizes Lipschitz constants across layers. From the learning-theoretic perspective, generalization guarantees for deep networks leverage norm-based capacity measures (e.g., spectral-complexity/margin bounds, path norms) (Bartlett et al., 2017; Neyshabur et al., 2015) together with classical complexity tools (Rademacher/Gaussian complexities and covering numbers) (Bartlett & Mendelson, 2002b). Our analysis adopts this toolbox but tailors it to sparse Transformers by (i) conditioning on a fixed routing pattern so that the model reduces to a smooth active subnetwork and (ii) union-bounding over routing patterns to account for combinatorial choices. For approximation, we rely on deep ReLU theory that provides near-optimal rates $\|f - \mathcal{N}\|_\infty \lesssim N^{-\beta/d}$ for $C^\beta$ functions on $d$-dimensional domains (Yarotsky, 2017; Schmidt-Hieber, 2020; Petersen & Voigtlaender, 2018). We lift these Euclidean results to data on manifolds by working in local charts and gluing approximants with partitions of unity, which pairs naturally with the MoE router's $k$-sparse mixing and underlies our approximation theorem.

**Mixture-of-Experts (MoE): conditional computation and systems.** Sparsely-gated MoE layers (Shazeer et al., 2017) activate only a small subset of experts per token, decoupling *total* parameters from per-example *active* compute. This idea enabled large sparse Transformers such as GShard (Lepikhin et al., 2021), Switch Transformers (Fedus et al., 2022), and GLaM (Du et al., 2022); in vision, V-MoE validated similar benefits for ViTs (Riquelme et al., 2021). Practical deployments introduced load-balancing losses and alternative dispatchers (BASE layers, expert-choice routing) to reduce hotspotting and improve stability, and specialized runtimes (DeepSpeed-MoE, Tutel) to scale training/inference (Lewis et al., 2021; Zhou et al., 2022; Rajbhandari et al., 2022; Hwang et al., 2023). Recent systems emphasize expert specialization and the gap between total and active parameters e.g., Mixtral ($8\times7$B, top-2 routing) and DeepSeek-V3 (hundreds of billions total, tens of billions active) (Jiang et al., 2024; DeepSeek-AI, 2024). These trends motivate our theoretical focus on *active* per-input capacity as the relevant approximation budget and on an explicit routing-combinatorics term that scales like $k \log(eM/k)$ in covering-number bounds. In our framework, hard top-$k$ and softmax gating with temperature/load-balancing behave similarly at the level of uniform bounds as long as the effective sparsity is $k$.

**Intrinsic-dimension theory for Transformers.** Our analysis builds on work showing that Transformer scaling is governed by the *intrinsic* rather than ambient dimension when data concentrate on low-dimensional structure. In particular, Havrilla & Liao (2024) develop an approximation statistical framework for Transformers on intrinsically low-dimensional data, deriving data/model exponents that depend explicitly on the manifold dimension and validating these predictions empirically. We adopt this viewpoint to interpret MoE: active per-token capacity plays the role of the effective model-size axis, while routing acts as an additive overhead. Our analysis builds on their dense-Transformer framework and extends it to MoE with routing. Theorem 3.2 mirrors their approximation result but constructs MoE layers with top-$k$ routing and separates active capacity from total parameters. This aligns our MoE scaling laws with intrinsic-dimension theory recovering dense-model exponents measured against *active* capacity and clarifying when routing can shift constants without changing rates.

**Scaling laws: dense vs. MoE.** Dense language models exhibit empirical power-law scaling of loss with model size, data, and compute (Hestness et al., 2017; Kaplan et al., 2020); the Chinchilla study argues for compute-optimal training by scaling model and data jointly under a fixed budget (Hoffmann et al., 2022). For MoE, recent empirical work proposes *MoE-specific* scaling formulations that disentangle *total* from *active* parameters and incorporate

routing effects. Fine-grained MoE scaling analyzes expert granularity, token/expert budgets, and the efficiency frontier as expert shards become smaller and more numerous (Krajewski et al., 2024). Upcycling laws study converting dense checkpoints into MoE while retaining predictable scaling behavior (Liew et al., 2025). Joint dense–MoE laws integrate active parameters, dataset size, and expert count to explain memory/throughput trade-offs (Ludziejewski et al., 2025), and unified efficient-MoE laws relate expert activation ratios, granularity, and compute (Tian et al., 2025). Large-scale case studies (e.g., extending fine-grained MoE beyond 50B parameters) provide further evidence that MoE can match or exceed dense scaling at comparable or lower active compute (Krajewski et al., 2025). Our theory complements these findings: we recover the dense-network exponents but measure them against the *active* parameter budget $N_{\text{act}}$, and we make explicit a routing overhead proportional to $k \log(eM/k)$ that shifts constants and determines finite-sample crossovers (e.g., long sequences or large $M \gg k$). This yields compute–optimal prescriptions consistent with practice grow data and *active* capacity together while clarifying when adding experts primarily changes constants rather than exponents.

## 3. MoE Model and Approximation Theory

We consider i.i.d. samples $(x_i, y_i)_{i=1}^n$ drawn from an unknown distribution $Q$ over $\mathcal{X} \times \mathcal{Y}$.

**Definition 3.1** (MoE Transformer)**.** Fix hyperparameters $(\ell, d_{\text{emb}}, m, L_T, M, k)$ and FFN dimensions $(L_{\text{FFN}}, W_{\text{FFN}})$. For an input $H^{(0)} \in \mathbb{R}^{\ell \times d_{\text{emb}}}$, each block $j = 1, \dots, L_T$ applies the following residual updates:

1. **Attention:**
$$\widetilde{H}^{(j)} = H^{(j-1)} + \text{MHA}_j\Big(H^{(j-1)}\Big).$$

2. **MoE feed-forward layer:** For each token embedding $\widetilde{h}_t \in \widetilde{H}^{(j)}$,
$$h_t^{(j)} = \widetilde{h}_t + \sum_{m=1}^M g_{j,m}(\widetilde{h}_t)\, E_{j,m}(\widetilde{h}_t),$$

where $\{E_{j,m}\}_{m=1}^M$ are MLP experts of depth $L_{\text{FFN}}$ and width $W_{\text{FFN}}$.

The routing weights satisfy, for every layer $j$ and token representation $h$,

$$g_{j,m}(h) \geq 0, \qquad \sum_{m=1}^M g_{j,m}(h) = 1, \qquad \big|\{m : g_{j,m}(h) \neq 0\}\big| \leq k,$$

Thus at most $k$ experts are active per token.

Definition 3.1 follows standard sparse MoE architectures (Shazeer et al., 2017; Lepikhin et al., 2021; Fedus et al., 2022). While this formulation captures expert sparsity, it is broader than what is needed for the statistical analysis. We therefore analyze a restricted MoE Transformer class in which: (i) submodules have fixed depth and width; (ii) routing is deterministic top-$k$; (iii) all parameters are uniformly bounded; and (iv) the routing family is expressive enough to implement a $k$-sparse partition of unity. These assumptions retain the active-computation structure of practical MoE models while allowing explicit control of covering numbers and routing combinatorics.

### 3.1. Approximation Bound Under Hölder Smoothness

**Theorem 3.2.** *Let* $\mathcal{M} \subset \mathbb{R}^D$ *be a compact $d$-dimensional $C^1$ submanifold, and let $f \in C^\beta(\mathcal{M})$ satisfy $\|f\|_{C^\beta} \leq B$, where $\beta > 0$. Consider the MoE Transformer class $\mathcal{T}_{\text{MoE}}$ (Definition 3.1) with $L_T$ blocks, sequence length $\ell$, and an MoE FFN sublayer per block with $M$ experts, of which at most $k$ are active per token. Assume each expert is a ReLU MLP of depth $L_{\text{FFN}}$, width $w_{\text{FFN}}$, and parameter budget $\Pi_{\text{exp}}$, and let $\Pi_{\text{attn}}$ denote the number of attention / non-expert parameters per block. Assume further that the router can implement a $k$-sparse partition of unity and that all weights are bounded by $\kappa$.*

*Define the* active attention+expert budget
$$N_{\text{act}} := L_T \Pi_{\text{attn}} + L_T k \Pi_{\text{exp}}.$$

*Then, as shown in Appendix F, there exist constants $C, c > 0$ depending on $(B, \beta, d, \mathcal{M})$ and polynomially on the architectural quantities $(L_T, w_{\text{FFN}}, L_{\text{FFN}}, \kappa)$ such that*

$$\inf_{T \in \mathcal{T}_{\text{MoE}}} \|T - f\|_\infty^2 \ \leq\ C \cdot \min\Big\{ N_{\text{act}}^{-2\beta/d},\ M^{-2\beta/d} \Big\}. \quad (1)$$

*Remark 3.3.* The exponent $-2\beta/d$ is the classical minimax exponent for $C^\beta$ approximation on a $d$-dimensional manifold. The constants in Theorem 3.2 depend on $(B, \beta, d, \mathcal{M})$ and polynomially on the architectural parameters. Standard manifold-approximation arguments imply that this dependence is at most polynomial in $d$ for manifolds with bounded regularity; the only exponential-in-$d$ behavior is the unavoidable covering-number dependence that produces the exponent itself.

*Proof sketch.* Appendix F gives a constructive proof of Theorem 3.2. $\square$

### 3.2. MoE Generalization Bound

**Theorem 3.4.** *Let $\mathcal{T}(\varepsilon)$ be a class of MoE Transformers such that for every $f \in C^\beta(\mathcal{M})$ there exists $T \in \mathcal{T}(\varepsilon)$ with*

$$\|T - f\|_{L^2(Q)} \leq \varepsilon.$$

*Let $\hat{T}_n \in \mathcal{T}(\varepsilon)$ denote an empirical risk minimizer under squared loss based on the i.i.d. sample $(x_i, y_i)_{i=1}^n$. Then*

$$\mathbb{E}\|\hat{T}_n - f\|_{L^2(Q)}^2 \leq \varepsilon^2 + \tilde{O}\left(\frac{N_{\text{act}}}{n} + \frac{L_T \ell k \log(eM/k)}{n}\right)$$

Here $\tilde{O}(g)$ hides polylogarithmic factors:

$$\tilde{O}(g) = O\left(g \cdot \text{polylog}\left(n, N_{\text{act}}, L_T \ell k, M/k, \kappa, R, M_0\right)\right).$$

**Corollary 3.5** (Generalization bound with MoE approximation rate). *Under the same router expressivity assumptions as in Theorem 3.2, substituting*

$$\varepsilon^2 \asymp N_{\text{act}}^{-2\beta/d}$$

*gives*

$$\mathbb{E}\|\hat{T}_n - f\|_{L^2(Q)}^2 \lesssim N_{\text{act}}^{-2\beta/d} + \frac{N_{\text{act}}}{n} + \frac{L_T \ell k \log(eM/k)}{n},$$

*up to polylogarithmic factors.*

*Remark* 3.6 (Looseness of the routing union bound). The factor $L_T \ell k \log(eM/k)$ comes from a worst-case union bound over all top-$k$ routing patterns:

$$\log|\Pi| \leq L_T \ell k \log(eM/k).$$

This contribution is conservative and should be interpreted as a worst-case upper bound rather than a tight characterization of practical routing behavior. In practice, router specialization can substantially reduce the effective number of routing patterns, so the bound is expected to be loose but safe.

*Proof sketch.* The proof (Appendix E) conditions on fixed routing patterns, reducing the MoE to deterministic active subnetworks. For each pattern, Lipschitz bounds and a covering argument over the active parameter space yield the dense-style rate, while a union bound over patterns adds the routing term $L_T \ell k \log(eM/k)$. □

**Lemma 3.7** (MoE covering number). *Let $\mathcal{X} \subset \mathbb{R}^D$ be compact with $\|x\|_\infty \leq M_0$. Consider the Mixture-of-Experts Transformer class*

$$\mathcal{T}_{\text{MoE}}(D, M, k, L_T, L_{\text{FFN}}, w_{\text{FFN}}, d_{\text{emb}}, m, \kappa, R)$$

*defined as in Definition 3.1, except that each feed-forward sublayer is an MoE layer with $M$ experts and the router selects hard top-$k$ experts per token per layer. Assume each learned parameter entry satisfies $\|\theta\|_\infty \leq \kappa$ and each network output is bounded by $\|T(x)\|_\infty \leq R$.*

*Let $\Pi_{\text{attn}}$ be the number of scalar parameters in the attention/non-expert parts per block, and let $\Pi_{\text{exp}}$ be the number of scalar parameters of a single expert MLP of depth $L_{\text{FFN}}$,*

*width $w_{\text{FFN}}$, and input/output dimension $d_{\text{emb}}$. Then for any $\delta \in (0,1)$, the covering number of $\mathcal{T}_{\text{MoE}}$ under the sup-norm satisfies*

$$\log \mathcal{N}(\delta, \mathcal{T}_{\text{MoE}}, \|\cdot\|_\infty) \leq C_1\left(\Pi_{\text{attn}} + L_T k \Pi_{\text{exp}}\right) \log\left(\frac{C_2 \kappa R M_0}{\delta}\right) + C_3 L_T \ell k \log\left(\frac{eM}{k}\right) \quad (2)$$

*for absolute constants $C_1, C_2, C_3 > 0$ depending polynomially on $d_{\text{emb}}, w_{\text{FFN}}, m, L_T, L_{\text{FFN}}, \ell$. Small polylogarithmic factors may be hidden in the $\tilde{O}(\cdot)$ notation.*

*Proof sketch.* The proof (Appendix H) conditions on a fixed top-$k$ routing pattern, reducing the MoE to a deterministic active subnetwork, and then union-bounds over all routing patterns. For a fixed pattern, a parameter-to-function Lipschitz bound for attention and FFN layers implies that an $\eta$-grid of $[-\kappa, \kappa]^{N_{\text{act}}}$ induces a $\delta$-cover of the corresponding function subclass, yielding (2) by standard covering-number arguments (Bartlett & Mendelson, 2002b; Vaswani et al., 2017; Ba et al., 2016). □

**Scope and interpretation of the bounds.** The theoretical analysis is developed for a structured class of MoE Transformers, including bounded parameters, hard top-$k$ routing, and explicit routing partitions. These assumptions are standard in uniform generalization analysis and are introduced to enable tractable covering-number arguments.

The resulting guarantees should be interpreted as *worst-case statistical bounds*. In particular, the routing complexity term reflects a union bound over all possible routing patterns and therefore provides a conservative upper bound rather than a tight characterization of practical MoE behavior.

# 4. Neural Scaling Laws

We derive explicit scaling laws for Mixture-of-Experts transformers that expose the dependence on sample size $n$, number of experts $M$, number of *active* experts per token $k$, and parameter budgets. Throughout we use the generalization bound from theorem 3.4 in the shorthand form

$$\mathbb{E}\|\hat{T}_n - f\|_{L^2(Q)}^2 \lesssim \underbrace{N_{\text{act}}^{-2\beta/d}}_{\text{approximation}} + \underbrace{\frac{N_{\text{act}}}{n}}_{\text{estimation}} + \underbrace{\frac{R_{\text{route}}}{n}}_{\text{routing}}. \quad (3)$$

where $R_{\text{route}} := L_T \ell k \log\left(\frac{eM}{k}\right)$.

## 4.1. Data Scaling

Let us study how test error decreases with the number of samples $n$ when the architecture is fixed or mildly tuned with $n$. Ignoring $R_{\text{route}}$ for the moment (we reintroduce it in §4.4), we balance approximation and estimation by

minimizing, over the active budget $N_{\mathrm{act}}$. Differentiating and setting to 0 gives

$$-\frac{2\beta}{d}N_{\mathrm{act}}^{-\frac{2\beta}{d}-1} + \frac{c}{n} = 0 \implies N_{\mathrm{act}}^{\star} \asymp \left(\frac{n}{c}\right)^{\frac{d}{2\beta+d}} \Rightarrow N_{\mathrm{act}}^{\star} \asymp n^{\frac{d}{2\beta+d}}$$

Substituting $N_{\mathrm{act}}^{\star}$ back we obtain the *data scaling law*

$$\mathbb{E}\|\hat{T}_n - f\|_{L^2(Q)}^2 \asymp n^{-\alpha_D}, \qquad \alpha_D = \frac{2\beta}{2\beta+d} \quad (4)$$

This exponent matches the dense-network exponent under the same intrinsic-dimension assumptions (Havrilla & Liao, 2024); the MoE twist is that $N_{\mathrm{act}}$ is the *active* budget per input (depending on $k$), not the total parameter count.

### 4.2. Model Scaling

We quantify how the error scales with the *active parameter budget* when the sample size $n$ is fixed.

**Proposition 4.1** (Model-scaling law in the approximation–dominated regime). *For a fixed $n$, let us suppose that $N_{\mathrm{act}}$ lies in the regime where the estimation and routing terms are dominated by the approximation term, namely there exist absolute constants $c_1, c_2 > 0$ such that*

$$\frac{N_{\mathrm{act}}}{n} \leq c_1 N_{\mathrm{act}}^{-\frac{2\beta}{d}} \quad and \quad \frac{R_{\mathrm{rt}}}{n} \leq c_2 N_{\mathrm{act}}^{-\frac{2\beta}{d}} \quad (5)$$

*Then we get*

$$\mathbb{E}\|\hat{T}_n - f\|_{L^2(Q)}^2 \asymp N_{\mathrm{act}}^{-\alpha_N}, \qquad \alpha_N = \frac{2\beta}{d} \quad (6)$$

**Proposition 4.2** (Characterization of the approximation–dominated regime). *Since $N_{\mathrm{act}}$ is the active parameter budget, the first condition in (5) is equivalent to*

$$N_{\mathrm{act}}^{1+2\beta/d} \lesssim n \iff N_{\mathrm{act}} \lesssim n^{\frac{d}{2\beta+d}} =: N_{\mathrm{act}}^{\star}(n),$$

*that is, below the* estimation crossover $N_{\mathrm{act}}^{\star}(n)$.

*The routing condition reads*

$$R_{\mathrm{route}} \lesssim n\, N_{\mathrm{act}}^{-2\beta/d} \iff N_{\mathrm{act}} \lesssim \left(\frac{n}{L_T \ell k \log(eM/k)}\right)^{\frac{d}{2\beta}} =: N_{\mathrm{act}}^{\star}(n, M, k, L_T, \ell).$$

*Therefore, the model-scaling law (6) holds whenever*

$$N_{\mathrm{act}} \leq \min\{N_{\mathrm{act}}^{\star}(n),\, N_{\mathrm{act}}^{\star}(n, M, k, L_T, \ell)\}. \quad (7)$$

*Remark* 4.3 (Crossover and beyond). At the estimation crossover $N_{\mathrm{act}} \approx N_{\mathrm{act}}^{\star}(n)$, the approximation and estimation terms balance:

$$N_{\mathrm{act}}^{-2\beta/d} \approx \frac{N_{\mathrm{act}}}{n} \implies N_{\mathrm{act}} \asymp n^{\frac{d}{2\beta+d}}, \quad \mathbb{E}\|\hat{T}_n - f\|_{L^2(Q)}^2 \asymp n^{-\frac{2\beta}{2\beta+d}}$$

For $N_{\mathrm{act}} \gg N_{\mathrm{act}}^{\star}(n)$ (holding $n$ fixed), the bound is dominated by $N_{\mathrm{act}}/n$ (plus routing), so increasing $N_{\mathrm{act}}$

*worsens* the bound. Thus (6) describes the *subcritical* (approximation-limited) regime, analogous to the dense case (Havrilla & Liao, 2024) but with $N_{\mathrm{act}}$ the *active* parameter budget.

**Corollary 4.4** (Model-scaling with experts-only approximation). *If attention does not contribute to approximation (so $N_{\mathrm{act}} = N_{\mathrm{exp}}$), then in the subcritical regime (Eq. (7))*

$$\mathbb{E}\|\hat{T}_n - f\|_{L^2(Q)}^2 \asymp (L_T k \Pi_{\mathrm{exp}})^{-\frac{2\beta}{d}} \quad (8)$$

### 4.3. Optimizing the Number of Active Experts $k$

Let $A := L_T \Pi_{\mathrm{exp}}$ and absorb the attention budget into constants. With experts dominate approximation, $N_{\mathrm{act}} \propto k$, the bound (3) reduces to the function of $k$:

$$\mathcal{E}(k) \lesssim \underbrace{(Ak)^{-2\beta/d}}_{\text{approx}} + \underbrace{\frac{Ak}{n}}_{\text{estimation}} + \underbrace{\frac{L_T \ell\, k\, \log(eM/k)}{n}}_{\text{routing}} \quad (9)$$

**Ignoring the log first.** Lets drop $\log(eM/k)$ and treat it as a slowly varying constant near the optimizer. Minimize $g(k) = (Ak)^{-2\beta/d} + \frac{\tilde{B}}{n}k$ over $k > 0$. Differentiating gives (up to architecture constants)

$$k^{\star} \asymp n^{\frac{d}{2\beta+d}} \quad and \quad \mathcal{E}(k^{\star}) \asymp n^{-\frac{2\beta}{2\beta+d}} \quad (10)$$

Enforce $k^{\star} \leq M$; otherwise the optimum saturates at $k = M$.

**Reintroducing the log.** With the routing term present, the derivative of the first-order condition gives

$$\frac{2\beta}{d}A^{-2\beta/d}k^{-2\beta/d-1} = \frac{1}{n}\left(A + L_T \ell \log\frac{eM}{k} - L_T \ell\right) \quad (11)$$

This matches the stated first-order condition whose closed form involves a Lambert-$W$ factor if solved exactly. Since $\log(eM/k)$ varies slowly near the optimizer, the solution is

$$k^{\star} \asymp \min\left\{M, \left(\frac{n}{A + L_T \ell \log(eM/k^{\star})}\right)^{\frac{d}{2\beta+d}}\right\} \quad (12)$$

Thus, within the worst-case bound, the dependence of $k^{\star}$ on $M$ enters only logarithmically. This does not preclude stronger empirical gains from larger expert pools arising through specialization effects outside the scope of the uniform analysis. The full proof of these calculations are provided in appendix G.1.

We stress that this conclusion holds at the level of the worst-case bound only; the expanded routing ablations in Section 5 show that, for sufficiently large $M/k$, empirical performance can improve substantially due to expert specialization effects outside the scope of the present analysis.

### 4.4. Routing-dominated vs. Power-law Regime

Comparing the routing term to the power-law term $n^{-2\beta/(2\beta+d)}$ yields the threshold

$$\frac{R_{\mathrm{rt}}}{n} \gtrsim n^{-\frac{2\beta}{2\beta+d}} \iff n \lesssim n_{\mathrm{thr}} := \left(L_T \ell k \log \frac{eM}{k}\right)^{\frac{2\beta+d}{d}} \quad (13)$$

$$\mathbb{E}\|\hat{T}_n - f\|^2 \approx \begin{cases} \frac{L_T \ell k \log(eM/k)}{n} & n \ll n_{\mathrm{thr}} \\ n^{-\frac{2\beta}{2\beta+d}} & n \gg n_{\mathrm{thr}} \end{cases} \quad (14)$$

The bound indicates that routing dominance is mitigated when factors such as $L_T$, $\ell$, $k$, and $\log(eM/k)$ are small, a pattern that is consistent with architectural choices observed in prior MoE systems. These strategies align with empirical system-level practices in sparse Transformers: for instance, Switch Transformers (Fedus et al., 2022) and GShard (Lepikhin et al., 2021) emphasized top-1 or top-2 routing for efficiency, while later refinements such as BASE Layers (Lewis et al., 2021) and expert-choice routing (Zhou et al., 2022) reduced routing variance and overhead. Large-scale MoE deployments (Riquelme et al., 2021; Du et al., 2022) also highlight the importance of balancing $M$ and $k$ to avoid routing bottlenecks. From the theoretical side, our bound formalizes these intuitions: the routing-dominated regime is suppressed precisely when the effective expert activation ($k$) and the combinatorial explosion in $M$ are controlled, allowing the system to operate in the favorable power-law regime where sample complexity scales as $n^{-2\beta/(2\beta+d)}$.

While the theory predicts a single smoothness parameter $\beta$, empirical estimates obtained from model scaling and data scaling may differ, as these regimes probe different components of the error decomposition (approximation vs estimation and optimization effects). We therefore focus on structural consistency of the scaling exponents rather than exact agreement of inferred $\beta$ values.

### 4.5. Sample Complexity for Target Error $\varepsilon$

In the power-law regime, where routing overhead is negligible relative to the approximation and estimation terms, the classical nonparametric rate applies. Specifically, achieving a target population error $\varepsilon$ requires

$$n(\varepsilon) \asymp \varepsilon^{-\frac{2\beta+d}{\beta}}, \qquad N_{\mathrm{act}}(\varepsilon) \asymp \varepsilon^{-\frac{d}{\beta}} \quad (15)$$

The first relation quantifies the number of samples needed as a function of smoothness $\beta$ and intrinsic dimension $d$; the second gives the corresponding effective parameter budget scaling to match the approximation rate. Both coincide with the dense-network theory (Yarotsky, 2017; Schmidt-Hieber, 2020), except that in MoE models $N_{\mathrm{act}}$ is interpreted as

the *active* parameter budget per input rather than the total parameter count.

In contrast, in the routing-dominated regime the error floor is determined by the combinatorial overhead of selecting experts. In this case, sample complexity to achieve accuracy $\varepsilon$ grows only linearly in $1/\varepsilon$:

$$n(\varepsilon) \asymp \frac{L_T \ell k \log(eM/k)}{\varepsilon} \quad (16)$$

This expression highlights that the overhead scales with the number of MoE layers ($L_T$), sequence length ($\ell$), and active experts per layer ($k$), while depending only logarithmically on the total number of experts $M$. Thus, if $M \gg k$, routing dominates and inflates sample complexity significantly, while $M = \Theta(k)$ mitigates this effect. In practice, this explains why architectures such as Switch Transformers (Fedus et al., 2022) or GLaM (Du et al., 2022) adopt small $k$ (top-1 or top-2 routing) and balance $M$ against $k$ to remain in the favorable power-law regime. Our theoretical analysis therefore provides a principled characterization of when MoE gains translate into efficient sample usage and when routing combinatorics instead dictate learning dynamics.

### 4.6. Compute-optimal Trade-offs

We derive the optimal allocation of *active parameters per input* $N_{\mathrm{act}}$ and *number of training samples* $n$ under a fixed training compute budget $C$. Throughout we use the bound in (3) and we adopt the standard compute model in which per-token (or per-sample) FLOPs scale linearly with the active parameter budget (Hoffmann et al., 2022; Kaplan et al., 2020), so the total training compute satisfies

$$C \asymp n \cdot N_{\mathrm{act}}. \quad (17)$$

**Proposition 4.5** (Compute-optimal allocation of $N_{\mathrm{act}}$ and $n$)**.** *Assume the compute budget* (17)*. Then the excess error bound is minimized (up to constants) by*

$$N_{\mathrm{act}}^{\star}(C) \asymp C^{\frac{d}{2\beta+2d}} \quad and \quad n^{\star}(C) = \frac{C}{N_{\mathrm{act}}^{\star}(C)} \asymp C^{\frac{2\beta+d}{2\beta+2d}} \quad (18)$$

*The resulting compute error scaling law is*

$$\mathbb{E}\|\hat{T}_{n^{\star}} - f\|_{L^2(Q)}^2 \asymp C^{-\frac{\beta}{\beta+d}} \quad (19)$$

Thus, under a fixed compute budget, both the dataset size and the per-sample active parameters should grow with $C$ according to the exponents in (18). The error decays as a power law in compute with exponent $\beta/(\beta+d)$, matching nonparametric theory and providing the MoE analogue of compute scaling laws. See appendix G.3 for proof.

## 5. Empirical Scaling and Evaluation on LLMs

Our empirical study is designed to examine the scaling exponents and routing effects suggested by the theoretical

analysis under controlled yet practically relevant conditions. We evaluate the framework on a family of causal, decoder-only Transformer language models with Mixture-of-Experts (MoE) feed-forward layers. We consider three public text corpora with differing levels of complexity and estimated intrinsic dimension: **TinyStories**, a low-dimensional synthetic dataset of children's stories (Eldan & Li, 2023); **WikiText-103**, a medium-scale corpus of moderately heterogeneous Wikipedia articles (Merity et al., 2016); and **OpenWeb-Text**, a large and highly heterogeneous corpus reconstructed from Reddit-linked web pages (Gokaslan & Cohen, 2019). Additional details are provided in Appendix A.

**Intrinsic dimension $d$ estimates in practice.** We estimate the intrinsic dimension $d$ of the data manifold from hidden representations using standard intrinsic-dimension estimators. Concretely, we pass corpus tokens through a fixed pretrained GPT-2 model, collect hidden states at a given layer, and apply the Levina–Bickel $k$-nearest-neighbor maximum-likelihood estimator, which is known to be more robust than TwoNN in high-dimensional settings (Fukunaga & Olsen, 1971; Levina & Bickel, 2004; Facco et al., 2017). Distances are computed using FAISS, and intrinsic dimension is evaluated on a layer-by-layer basis. To reduce variance, we repeat the procedure over multiple random subsamples and report the **median** intrinsic dimension. Unless stated otherwise, we use the **middle Transformer layers** as a representative summary of $d$, following common practice in prior work. The resulting estimates are stable across layers and subsamples. The estimated intrinsic dimensions are lowest for TinyStories and higher for WikiText-103 and OpenWebText, consistent with their relative heterogeneity; exact values are reported in Table 1, with depth-wise trends reported in Figures 2 and 3 in Appendix A.

We emphasize that these estimates are representation-dependent and should be viewed as a coarse proxy for data complexity rather than as a definitive characterization of the learned MoE representations.

**Experimental setup.** We operate in a small- to medium-scale regime to enable dense sweeps over the capacity grid. Unless noted otherwise, we use sequence lengths $\ell \in \{128, 256\}$, $m = 4$ attention heads, the AdamW optimizer, and a learning rate of $3 \times 10^{-4}$. For model scaling, we vary $(L_T, d_{\text{ff}})$ over a broad grid and consider multiple MoE configurations $(M, k)$. For data scaling, we fix a representative architecture and vary the token budget $D$ over the range $5 \times 10^4$–$8 \times 10^5$. We examine three empirical regimes corresponding to the theoretical analysis: (i) *model scaling*, where we estimate the exponent $\widehat{\alpha}_N$ by regressing $\log L$ on $\log N_{\text{act}}$; (ii) *data scaling*, where we regress $\log L$ on $\log D$ to extract $\widehat{\alpha}_D$; and (iii) *routing ablations*, where we vary $(M, k)$ to probe the contribution of routing combinatorics and identify the transition from routing-influenced behavior

to a power-law-like regime.

Empirical estimates of $\beta$ obtained from model scaling and data scaling do not coincide exactly, which is expected since these regimes probe different components of the error decomposition. We therefore treat $\beta$ as an effective smoothness parameter and focus on structural consistency of the exponents rather than exact parameter recovery. Additional discussion is provided in Appendix B.

### 5.1. Fitting Exponents and Comparison to Theory

In all scaling experiments, we operated in regimes where routing overhead was either negligible relative to the validation loss or explicitly accounted for. When needed, we introduced a small loss floor $c$ and fit linear models to $\log(L - c)$ as a function of the relevant scaling variable ($\log N_{\text{act}}$ or $\log D$ ), yielding empirical exponents $(\widehat{\alpha}_N, \widehat{\alpha}_D)$. We compared these exponents to theoretical values $(\alpha_N, \alpha_D)$ derived from the intrinsic-dimension framework using an estimated dimension $d$ and a range of smoothness parameters $\beta$. Across datasets, the empirical and theoretical exponents are of comparable magnitude and approximately satisfy the compute-optimal consistency relation $\alpha_D \approx \alpha_N/(1 + \alpha_N)$. These observations suggest that the intrinsic-dimension-based theory provides a coherent statistical reference for observed model and data-scaling behavior, without implying tight quantitative agreement. Additional sensitivity analyses over $\beta$ are reported in Appendix A.6.

### 5.2. Routing ablation

To probe routing effects, we vary the total number of experts $M$ and the number of active experts $k$. Figure 1 reports routing ablations over $(M, k)$ at fixed training compute. The results reveal two regimes.

In the moderate regime ($M/k \leq 8$), validation loss increases approximately monotonically with the routing term $k \log(eM/k)$, consistent with the theoretical interpretation of routing as a statistical overhead in the uniform bound.

For larger values of $M/k$, we observe a reversal: increasing the expert pool can improve performance despite higher routing complexity. This specialization-dominated regime is not captured by the worst-case analysis and suggests that practical MoE models can benefit from structured routing and expert specialization beyond what uniform bounds certify.

Overall, this two-regime behavior supports interpreting the routing term as a conservative upper bound rather than a tight characterization of practical performance.

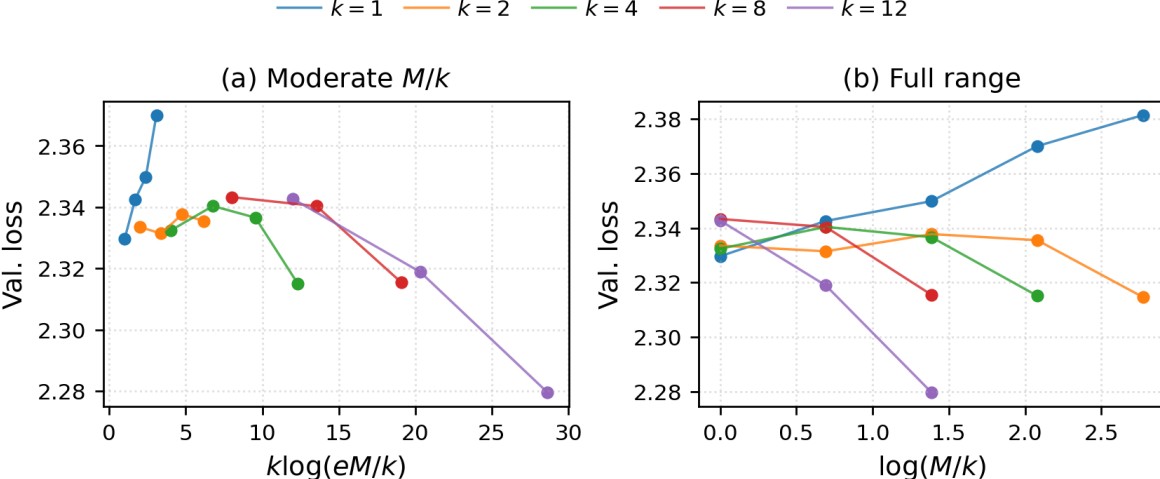

*Figure 1.* **Routing ablation across expert pool size** $M$ **and active experts** $k$**. (a)** In the moderate regime ($M/k \leq 8$), validation loss increases with the routing complexity term $k \log(eM/k)$, consistent with its interpretation as a worst-case routing overhead. **(b)** Over the full range, performance improves again for sufficiently large $M/k$, suggesting gains from expert specialization not captured by the worst-case analysis.

## 6. Discussion and Limitations

Our results indicate that *active per-token capacity* plays a central role in MoE performance, while routing contributes an additional overhead whose effect depends on the regime. When routing complexity remains modest, MoE models admit dense-style scaling behavior when measured against active parameters rather than total model size. Across TinyStories, WikiText-103, and OpenWebText, the observed model- and data-scaling exponents are of comparable magnitude to those suggested by intrinsic-dimension analysis. The expanded routing ablations further reveal two regimes: a moderate regime in which performance degrades with the routing term $k \log(eM/k)$, consistent with its role in the worst-case theory, and a specialization-dominated regime for larger $M/k$, in which increasing the expert pool improves performance. The constants in our bounds are conservative, and the regimes studied remain far from asymptotic.

From a practical standpoint, three architectural factors appear central to MoE behavior: activation sparsity, expert granularity, and routing diversity. Within worst-case statistical guarantees, expert pools that grow much larger than the number of activated experts are not certified to improve rates beyond a logarithmic term; however, our routing ablations show that larger expert pools can yield substantial empirical gains through specialization when $M/k$ is sufficiently large. Routing entropy and load balance therefore remain useful diagnostics during training. For long sequences, smaller values of $k$ can help limit routing overhead and delay the onset of routing-dominated regimes, although data-dependent effects may substantially alter these trade-offs in practice.

Our analysis focuses on statistical complexity rather than

optimization dynamics. In practical MoE systems, mechanisms such as expert specialization, load balancing, and routing regularization can significantly influence behavior, potentially reducing effective routing complexity and improving performance beyond worst-case predictions.

**Limitations and future directions.** Our analysis relies on several simplifying assumptions, including bounded parameters, hard top-$k$ routing, and squared loss. While these assumptions are standard in uniform convergence analysis, they do not fully capture practical MoE training settings. In particular, the derived bounds are likely conservative: the linear dependence on sequence length $\ell$ and the number of active experts $k$ arises from worst-case covering arguments and may overestimate the effective complexity of practical models. Moreover, the analysis does not capture approximation gains from expert specialization, which we observe empirically in regimes where $M/k$ is large. Developing tighter data-dependent analyses, together with corresponding lower bounds, remains an important direction for future work.

**What the theory certifies.** Our results should be interpreted as worst-case statistical guarantees rather than tight predictions of practical MoE training. The analysis certifies that, under uniform assumptions, the dominant approximation–estimation tradeoff is controlled by the active parameter budget, while routing contributes an additional combinatorial overhead of order $L_T \ell k \log(eM/k)$. This explains why dense-style scaling exponents reappear when model size is measured in active parameters.

**What remains beyond the theory.** The bounds do not certify gains from expert specialization, routing stability,

load balancing, or optimization dynamics. The routing ablations in Figure 1 suggest that such effects become important when the expert pool is large relative to the number of active experts. This points to a natural next step: replacing worst-case routing counts with data-dependent notions of effective routing entropy and deriving guarantees that capture specialization-driven gains.

## 7. Conclusion

We developed a statistical framework for Mixture-of-Experts (MoE) Transformers that separates *active* per-token capacity from routing combinatorics. Under standard regularity assumptions, we derive approximation and uniform generalization bounds in which excess risk decomposes into approximation, estimation, and routing terms. When expressed in terms of active parameters, these bounds imply scaling exponents that mirror dense-model behavior, while isolating routing as the distinct MoE-specific statistical overhead. Within this framework, we characterize the scaling of the optimal activation level up to architectural and logarithmic factors, and show that, under worst-case guarantees, increasing the total number of experts affects the bound only logarithmically through routing complexity. At the same time, our expanded routing ablations reveal a richer empirical picture: in a moderate regime, performance degrades with the routing term $k \log(eM/k)$ as predicted by the theory, whereas for sufficiently large $M/k$, larger expert pools improve performance through specialization effects that are not captured by the worst-case analysis.

Overall, our work provides a conservative statistical reference point for reasoning about sparse Transformer scaling. It clarifies when MoE models should behave like dense models measured in active capacity, and when additional data-dependent structure in routing and expert specialization can lead to behavior beyond uniform guarantees.

## Impact Statement

This paper presents theoretical and empirical work aimed at improving the scientific understanding of Mixture-of-Experts Transformers and sparse model scaling. Its primary contribution is methodological rather than directly deployable. A possible positive impact is improved compute efficiency in large-scale machine learning through better understanding of active capacity and routing. As with other work on scaling advanced models, these insights could also indirectly support the development of more capable systems, with downstream risks similar to those already associated with large language models, including misuse, bias propagation, and unequal access. We do not believe this paper raises ethical concerns beyond those already well established for research on large-scale machine learning.

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

# A. Empirical Validation on MoE Transformer LLMs

This section provides a detailed description of the architectures, hyperparameter grids, and scaling protocols used to ensure the empirical results are consistent with our theoretical predictions. Our goal is to cleanly isolate the roles of (i) effective model capacity, (ii) dataset size, and (iii) routing combinatorics in MoE Transformers.

## A.1. Datasets.

We consider three publicly available text corpora with varying complexity and intrinsic dimension: **Stories / TinyStories** is a corpus of short, synthetically generated children's stories, following (Eldan & Li, 2023). This dataset is known to lie on an intrinsically low-dimensional manifold, and has been used in prior work on theoretical scaling laws for Transformers. **WikiText-103** is a curated subset of English Wikipedia articles (Merity et al., 2016), commonly used for language modeling. It provides a medium-scale, moderately heterogeneous domain. **OpenWebText** is an open recreation of the WebText corpus (Gokaslan & Cohen, 2019), constructed from outgoing Reddit links. It is substantially larger and more heterogeneous than WikiText-103, with higher intrinsic dimension. For each corpus we extract a plain-text training stream (one document per line). For scaling experiments we sub-sample the stream to match the desired token budgets.

## A.2. Intrinsic Dimension Estimation (Extended Description)

The intrinsic dimension (ID) of a representation manifold quantifies the effective degrees of freedom of the learned features. To estimate the ID of transformer representations, we employ the Levina–Bickel Maximum Likelihood Estimator (MLE) (Levina & Bickel, 2004), which models the local likelihood of $k$-nearest-neighbor distances on a smooth manifold. Compared to the TwoNN estimator, which relies only on the ratio of the first two nearest neighbors, the MLE approach uses the full local neighborhood and exhibits substantially lower variance in high-dimensional settings an essential property when studying large language models.

We extract hidden states from each block of a pretrained GPT-2 model (gpt2-medium) by sampling tokens from a large text corpus. For each layer $\ell$, we compute all pairwise distances to its $k$ nearest neighbors using FAISS in 32-bit precision, and apply the MLE estimator to the resulting distance matrix $D_\ell$. The estimation pipeline is repeated for multiple random subsamples to mitigate sampling noise; we report the **median MLE estimate** for each layer, along with the **median absolute deviation (MAD)** as a robust measure of uncertainty.

Following established practice in intrinsic-dimension analyses of deep networks, we summarize the model-level intrinsic dimension using the **middle layers** 40–60% depth). Shallow layers typically show inflated dimensionality due to local embedding variation, whereas deeper layers tend to collapse. This procedure yields stable and reproducible ID estimates that are consistent with theoretical predictions from neural scaling laws, allowing us to define a characteristic ID ($d$) for each corpus that captures its inherent complexity, as summarized in the depth curves (Figure 2) and the scaling law fits (Figure 3).

*Table 1.* **Intrinsic Dimension (ID) Summary across Models and Datasets.** The ID estimates remain remarkably stable across different model sizes for each dataset. For the subsequent scaling analysis (Figure 5 and 4), we chose the median ID values for each dataset.

| Model (number of parameters) | Tinystories | | Wikitext-103 | | OpenWebText | |
|---|---|---|---|---|---|---|
| | Mean (CI) | Median | Mean (CI) | Median | Mean (CI) | Median |
| gpt2 (117 million) | $22.9 \pm 2.9$ | 23.1 | $31.6 \pm 2.9$ | 32.4 | $46.7 \pm 6.2$ | 49.8 |
| gpt2-medium (345 million) | $21.4 \pm 1.1$ | 21.9 | $31.6 \pm 1.7$ | 32.9 | $43.7 \pm 2.6$ | 45.0 |
| gpt2-large (774 million) | $22.1 \pm 0.9$ | 22.4 | $31.5 \pm 1.6$ | 32.1 | $44.7 \pm 2.6$ | 47.7 |
| gpt2-xl (1558 million) | $19.7 \pm 0.6$ | 19.6 | $30.3 \pm 1.0$ | 31.1 | $43.0 \pm 1.7$ | 43.9 |
| **Values Adopted for Scaling Analysis** | | | | | | |
| **Chosen $d$ (Median ID)** | $d = 23$ | | $d = 32$ | | $d = 45$ | |
| **Best-Fit $\beta$** | $\beta \approx 1.0$ | | $\beta \approx 1.0$ | | $\beta \approx 1.0 - 1.5$ | |

## A.3. Model family: MoE decoder-only Transformers

All experiments use a causal decoder-only Transformer with Mixture-of-Experts (MoE) feed-forward layers. Each model consists of token and positional embeddings, followed by $L_T$ Transformer blocks. Each block contains:

*Table 2.* Theoretical Exponents derived from the Chosen $d$ and $\beta$.

| Dataset | Chosen $d$ | Best-Fit $\beta$ | Theoretical Exponents | | | Empirical Exponents | |
|---|---|---|---|---|---|---|---|
| | | | $\alpha_N = \frac{2\beta}{d}$ | $\alpha_D = \frac{2\beta}{2\beta+d}$ | Consistency $\frac{\alpha_N}{1+\alpha_N}$ | $\widehat{\alpha}_N$ | $\widehat{\alpha}_D$ |
| **WikiText-103** | 32 | 1.0 | 0.063 | 0.059 | 0.059 | **0.060** | **0.058** |
| **OpenWebText** | 45 | 1.0 | 0.044 | 0.043 | 0.042 | **0.045** | **0.043** |
| **OpenWebText** | 45 | 1.5 | 0.067 | 0.062 | 0.063 | **0.068** | **0.062** |
| **TinyStories** | 23 | 1.0 | 0.087 | 0.080 | 0.080 | **0.085** | **0.081** |

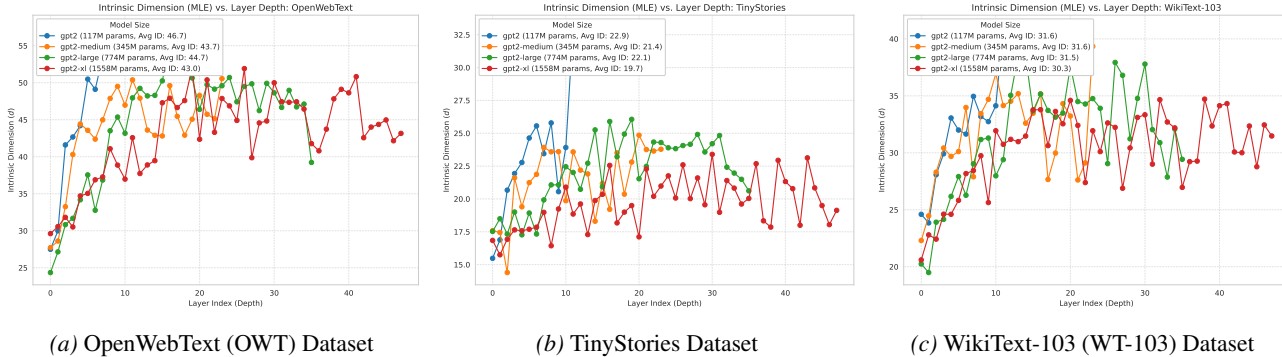

*(a)* OpenWebText (OWT) Dataset      *(b)* TinyStories Dataset      *(c)* WikiText-103 (WT-103) Dataset

*Figure 2.* **Intrinsic Dimension (ID) Evolution Across Model Depth.** The plots show the Maximum Likelihood Estimation (MLE) of the Intrinsic Dimension, $d$, calculated for the representations at each layer of the four GPT-2 models (varying size). The ID remains stable across different model sizes for a given dataset but varies significantly across data corpora. (a) **OpenWebText (OWT):** Shows the highest intrinsic dimension, reflecting its diverse and complex nature. (b) **TinyStories:** Exhibits the lowest intrinsic dimension, consistent with its synthetic, simple structure. (c) **WikiText-103 (WT-103):** Has an intermediate intrinsic dimension.

- **Self-attention:** multi-head attention with $m$ heads, model dimension $d_{\text{model}}$, causal masking, and standard projection matrices.

- **MoE feed-forward network:** a routing network selecting the top-$k$ experts. Each expert is an MLP with hidden width $d_{\text{ff}}$ and a GELU nonlinearity. Outputs of the selected experts are combined via a softmax over routing logits.

The *active* parameter budget per input,

$$N_{\text{act}} = L_T\big(\Pi_{\text{attn}} + k\,\Pi_{\text{exp}}\big),$$

counts only the parameters that contribute to computation for a given token. This is the quantity that appears in our approximation and covering-number bounds, and serves as the scaling variable in model-capacity experiments.

### A.4. Hyperparameters and Capacity Grids

We operate in the small-to-medium compute regime to allow dense parameter sweeps and low-noise log–log regressions. Unless otherwise stated, we use:

- sequence length $\ell \in \{128, 256\}$,

- $m = 4$ attention heads,

- AdamW with learning rate $3 \times 10^{-4}$ and weight decay $0.1$.

**Model-scaling grid.** To vary the active capacity over 1–2 orders of magnitude, we sweep

$$(L_T, d_{\text{ff}}) \in \{2, 3, 4, 5, 6, 8\} \times \{256, 384, 512, 640, 768, 896, 1024, 1280, 1536\},$$

combined with MoE configurations $M \in \{4, 8, 16\}$ and $k \in \{1, 2\}$. We avoid extremely large $M$ or $k$ to remain within GPU memory while still covering a wide range of effective capacities.

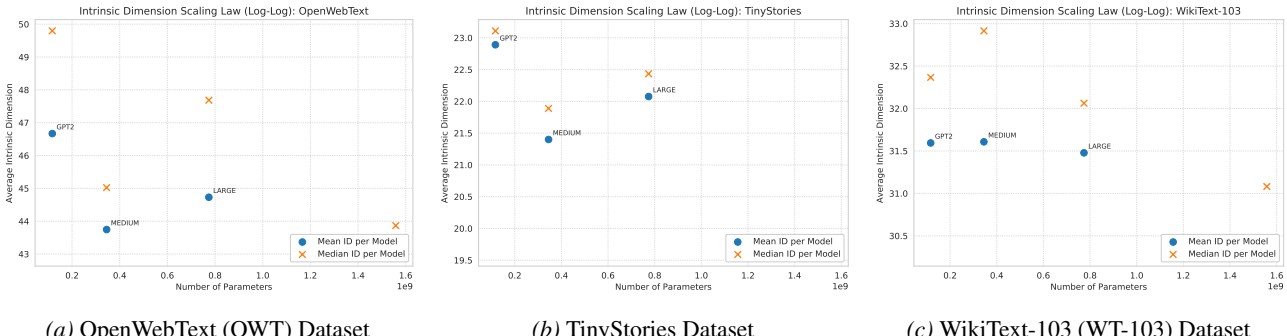

*(a)* OpenWebText (OWT) Dataset      *(b)* TinyStories Dataset      *(c)* WikiText-103 (WT-103) Dataset

*Figure 3.* **Intrinsic Dimension Scaling Law:** $\bar{d}$ **versus Number of Parameters** ($N$). The plots show the relationship between the **Average Intrinsic Dimension** ($\bar{d}$) (log-log scale) and the **Number of Parameters** ($N$) across the four GPT-2 model sizes for each dataset. This relationship is modeled as a power law, $\bar{d} \sim N^{\beta}$, where the slope of the linear fit determines the **ID Scaling Exponent** $\beta$. The results suggest that $\bar{d}$ exhibits weak or negligible scaling with model size across all three corpora.

### A.5. Scaling Protocols

We run three classes of experiments that map directly onto the theoretical decomposition of the generalization bound.

**(1) Model scaling.** For a fixed dataset, we train all models in the sweeping grid under a common token budget $D_{\text{fixed}}$. After convergence, we record the validation loss $L$ and fit a linear regression of $\log L$ versus $\log N_{\text{act}}$ over the region where routing is negligible. The slope yields the empirical model exponent $\widehat{\alpha}_N$.

**(2) Data scaling.** We choose a representative base architecture ($L_T = 4$, $d_{\text{model}} = 512$, $d_{\text{ff}} = 512$, $M = 8$, $k = 2$) and vary the number of training tokens

$$D \in \{5 \times 10^4, 10^5, 2 \times 10^5, 4 \times 10^5, 8 \times 10^5\}.$$

For each value of $D$ we adjust the number of optimization steps to keep the total token budget fixed. A regression of $\log L$ on $\log D$ yields the empirical data exponent $\widehat{\alpha}_D$.

**(3) Routing ablations.** To isolate the influence of routing combinatorics, we fix the base architecture and token budget, and vary both the number of active experts $k$ and the expert pool size $M$. In the expanded routing sweep, we use

$$k \in \{1, 2, 4, 8, 12\}, \qquad M/k \in \{1, 2, 4, 8, 16\},$$

For each $(M, k)$ pair we train a model, record its validation loss, and compute the theoretical routing term $L_T \ell k \log(eM/k)$. Plotting loss against this term reveals the routing-dominated regime and the predicted crossover into the power-law region governed by $(N_{\text{act}}, D)$. The models considered in this part are listed in Tables 3 and 4.

### A.6. Scaling-Sensitivity Plots

## B. Relation to Prior Empirical MoE Scaling Laws

We briefly relate our statistical scaling picture to recent empirical MoE scaling laws, including Efficient MoE Scaling (EL) (Tian et al., 2025) and Joint MoE Scaling Laws (JMSL) (Ludziejewski et al., 2025). In the routing-negligible regime, our theory predicts that MoE models recover dense-style scaling exponents when model size is measured by the active parameter budget $N_{\text{act}}$ rather than by total parameters:

$$\alpha_N = \frac{2\beta}{d}, \qquad \alpha_D = \frac{2\beta}{2\beta + d}, \qquad \alpha_C = \frac{\beta}{\beta + d}.$$

Eliminating $(d, \beta)$ gives the parameter-free identities

$$\alpha_D = \frac{\alpha_N}{1 + \alpha_N}, \qquad \alpha_C = \frac{\alpha_N}{2 + \alpha_N} = \frac{\alpha_N \alpha_D}{\alpha_N + \alpha_D}.$$

| $L_T$ | $d_{ff}$ | $N_{\text{act}}$ | $d_{\text{model}}$ | $n_{\text{heads}}$ | $M$ | $k$ |
|---|---|---|---|---|---|---|
| 2 | 256 | 3,148,800 | 512 | 4 | 8 | 2 |
| 2 | 384 | 3,673,600 | 512 | 4 | 8 | 2 |
| 2 | 512 | 4,198,400 | 512 | 4 | 8 | 2 |
| 2 | 768 | 5,248,000 | 512 | 4 | 8 | 2 |
| 2 | 1024 | 6,297,600 | 512 | 4 | 8 | 2 |
| 4 | 256 | 6,297,600 | 512 | 4 | 8 | 2 |
| 4 | 384 | 7,347,200 | 512 | 4 | 8 | 2 |
| 4 | 512 | 8,396,800 | 512 | 4 | 8 | 2 |
| 4 | 768 | 10,496,000 | 512 | 4 | 8 | 2 |
| 4 | 1024 | 12,595,200 | 512 | 4 | 8 | 2 |
| 4 | 1536 | 16,793,600 | 512 | 4 | 8 | 2 |
| 6 | 768 | 15,744,000 | 512 | 4 | 8 | 2 |
| 6 | 1024 | 18,892,800 | 512 | 4 | 8 | 2 |
| 8 | 1024 | 25,190,400 | 512 | 4 | 8 | 2 |

*Table 3.* **Model configurations used in the scaling experiments for each data set**. Each row corresponds to a distinct Transformer-MoE model used in the model-scaling sweep. The effective active parameter count $N_{\text{act}}$ varies with the number of layers and FFN width, while $d_{\text{model}} = 512$, $n_{\text{heads}} = 4$, $M = 8$, $k = 2$, and sequence length $\ell = 256$ are held fixed.

| $L_T$ | $d_{\text{ff}}$ | $N_{\text{act}}$ | $d_{\text{model}}$ | $n_{\text{heads}}$ | $M$ | $k$ | $D$ |
|---|---|---|---|---|---|---|---|
| 4 | 512 | 8,396,800 | 512 | 4 | 8 | 2 | $5 \times 10^4$ |
| 4 | 512 | 8,396,800 | 512 | 4 | 8 | 2 | $1 \times 10^5$ |
| 4 | 512 | 8,396,800 | 512 | 4 | 8 | 2 | $2 \times 10^5$ |
| 4 | 512 | 8,396,800 | 512 | 4 | 8 | 2 | $4 \times 10^5$ |
| 4 | 512 | 8,396,800 | 512 | 4 | 8 | 2 | $8 \times 10^5$ |

*Table 4.* **Model configuration used in the data-scaling experiments.** The architecture is held fixed while the training token budget $D$ is varied. Unless stated otherwise, the configuration corresponds to a representative mid-scale Transformer–MoE model with fixed active parameter budget $N_{\text{act}}$.

These identities provide a simple way to compare our intrinsic-dimension theory with empirical MoE scaling studies.

Empirical works such as JMSL fit Chinchilla-style loss surfaces in which the model and data-scaling exponents depend on the number of experts. When the reported exponents are mapped to $(\alpha_N, \alpha_D)$, the qualitative picture is consistent with our framework: active parameters are the relevant model-size axis, while increasing the number of experts can improve data efficiency and shift compute-optimal allocations toward more tokens per active parameter. At the same time, the empirical exponents do not satisfy the above identities exactly, and deviations tend to grow with the expert pool size. This is expected: our identities describe an idealized routing-negligible regime, whereas practical MoE systems benefit from expert specialization, load balancing, routing regularization, optimization effects, and finite-sample phenomena.

Thus, our theory should be viewed as complementary to empirical MoE scaling laws. It provides a conservative statistical baseline explaining why dense-style exponents reappear when measured against $N_{\text{act}}$, while empirical studies capture additional expert-dependent gains that require data-dependent or algorithm-dependent analysis.

## C. Additional Preliminaries

### C.1. Notation and basic definitions

**Sets, measures, and expectations.** For a measurable space $(\mathcal{X}, \mathcal{B})$ and probability measure $Q$ on $(\mathcal{X}, \mathcal{B})$, we write $X \sim Q$ for a random variable taking values in $\mathcal{X}$, and $E[\cdot]$ for expectation with respect to $Q$. The symbol $\mathbb{P}$ refers to probability.

**Covering numbers and metric entropy.** Let $(\mathcal{F}, \rho)$ be a metric space. An $\delta$-*cover* of $\mathcal{F}$ is a finite subset $\{f_1, \ldots, f_N\} \subset \mathcal{F}$ such that for every $f \in \mathcal{F}$ there exists $j \in [N]$ with $\rho(f, f_j) \leq \delta$. The *covering number* (Vershynin, 2018) is

$$N(\delta, \mathcal{F}, \rho) := \min\{ N \in \mathbb{N} : \exists\, \delta\text{-cover of } \mathcal{F} \text{ with } N \text{ elements} \}$$

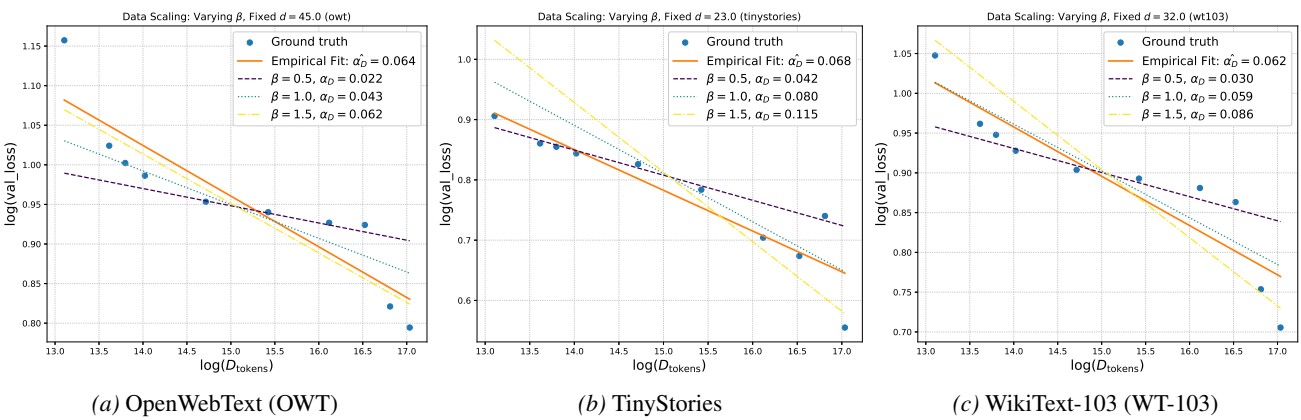

*(a)* OpenWebText (OWT)      *(b)* TinyStories      *(c)* WikiText-103 (WT-103)

*Figure 4.* **Data-scaling results (fixed $d$, varying $\beta$).** Empirical estimates of the data-scaling law (loss versus data size $D$) across datasets, compared with theoretical lines obtained by fixing the dataset-specific intrinsic dimension $d$ and varying $\beta$. The empirical fits are most consistent with $\beta = 1.5$ for OpenWebText and $\beta = 1.0$ for TinyStories and WikiText-103.

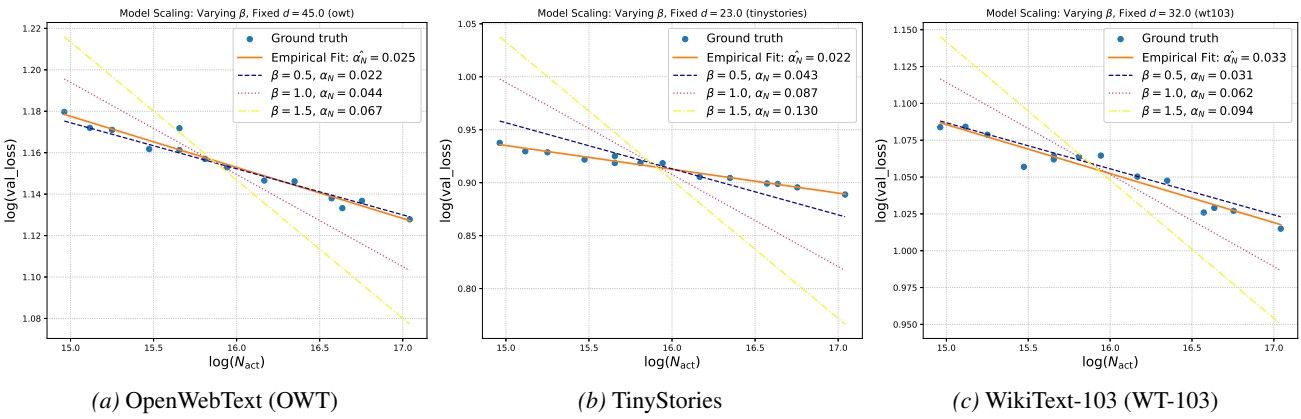

*(a)* OpenWebText (OWT)      *(b)* TinyStories      *(c)* WikiText-103 (WT-103)

*Figure 5.* **Model-scaling results (fixed $d$, varying $\beta$).** Empirical estimates of the model-scaling law (loss versus active parameter count $N_{\mathrm{act}}$) across datasets. For each plot, the dataset-specific intrinsic dimension $d$ is fixed and theoretical lines are shown for varying $\beta$, with theoretical exponent $\alpha_N = 2\beta/d$. The empirical fits are most consistent with $\beta = 0.5$ across all three datasets.

The (log) *metric entropy* is $\log N(\delta, \mathcal{F}, \rho)$. We often use $\rho(f, g) = \|f - g\|_\infty$.

**Vectors, matrices, tensors.** We use bold uppercase (e.g., $H$) for matrices and bold lowercase (e.g., $h$) for vectors when convenient. For a vector $v \in \mathbb{R}^d$, define the entrywise $\ell_\infty$ norm

$$\|v\|_\infty := \max_{1 \le i \le d} |v_i|$$

For a matrix $A \in \mathbb{R}^{d_{\mathrm{out}} \times d_{\mathrm{in}}}$, we use two norms: $\|A\|_\infty := \max_{i,j} |A_{ij}|$

For an embedding matrix $H \in \mathbb{R}^{\ell \times d_{\mathrm{emb}}}$ (sequence length $\ell$, embedding size $d_{\mathrm{emb}}$), we adopt

$$\|H\|_\infty := \max_{t \in [\ell], \, j \in [d_{\mathrm{emb}}]} |H_{t,j}|$$

**Asymptotic notation.** For nonnegative quantities $a, b$, $a \lesssim b$ means $a \le C\,b$ for an absolute constant $C$; $a \simeq b$ means $a \lesssim b$ and $b \lesssim a$. We write $\widetilde{O}(\cdot)$ to hide polylogarithmic factors in problem parameters.

## C.2. Loss and learning rule

**Squared regression error.** Given a predictor $T : \mathcal{X} \to \mathbb{R}$, the population (squared) risk under $Q$ is

$$L(T) := E\big[(T(X) - f(X))^2\big] = \|T - f\|_{L^2(Q)}^2$$

Empirically, for samples $\{x_i\}_{i=1}^n$, the empirical risk is

$$L_n(T) := \frac{1}{n} \sum_{i=1}^n \big(T(x_i) - f(x_i)\big)^2$$

**Empirical risk minimization (ERM).** An empirical risk minimizer (ERM) over $\mathcal{T}$ is any measurable selection $\hat{T}_n \in \mathcal{T}$ satisfying

$$\hat{T}_n \in \arg\min_{T \in \mathcal{T}} L_n(T)$$

## C.3. Input-Lipschitz bound

There exists $L_{\text{in}}^{(\text{dense})} \geq 1$, depending polynomially on $(\ell, m, d_{\text{emb}}, w_{\text{FFN}}, L_{\text{FFN}})$ and on $\kappa$, such that for all $H, H' \in \mathbb{R}^{\ell \times d_{\text{emb}}}$,

$$\|B_{\text{dense}}(\theta, H) - B_{\text{dense}}(\theta, H')\|_\infty \leq L_{\text{in}}^{(\text{dense})} \|H - H'\|_\infty, \tag{20}$$

with the schematic scaling (hiding absolute constants and activation Lipschitz factors)

$$L_{\text{in}}^{(\text{dense})} \lesssim 1 + C_{\text{LN}}\, \kappa \Big( \underbrace{\ell\, m\, d_{\text{emb}}^2}_{\text{MHA sensitivity}} + \underbrace{d_{\text{emb}}\, w_{\text{FFN}}^{L_{\text{FFN}}}}_{\text{FFN sensitivity}} \Big).$$

## C.4. Softmax Lipschitz

We work with the *temperature-$\tau$* softmax $\sigma_\tau : \mathbb{R}^m \to \Delta^{m-1}$,

$$\sigma_\tau(u)_i = \frac{e^{u_i/\tau}}{\sum_{j=1}^m e^{u_j/\tau}}, \qquad \tau > 0.$$

Its Jacobian is

$$J_\tau(u) = \frac{1}{\tau}\Big(\text{Diag}(\sigma_\tau(u)) - \sigma_\tau(u)\sigma_\tau(u)^\top\Big).$$

**Lemma C.1** (Softmax is $(2\tau)^{-1}$-Lipschitz in $\|\cdot\|_\infty$). *For any $u, v \in \mathbb{R}^m$,*

$$\|\sigma_\tau(u) - \sigma_\tau(v)\|_\infty \leq \frac{1}{2\tau} \|u - v\|_\infty.$$

*Proof.* By the mean value theorem, $\sigma_\tau(u) - \sigma_\tau(v) = \int_0^1 J_\tau\big(v + t(u-v)\big)(u-v)\, dt$. Thus

$$\|\sigma_\tau(u) - \sigma_\tau(v)\|_\infty \leq \sup_{w \in [u,v]} \|J_\tau(w)\|_{\infty \to \infty} \|u - v\|_\infty.$$

Row $i$ of $J_\tau(w)$ has entries $\frac{1}{\tau}\sigma_i(\delta_{ij} - \sigma_j)$. Summing absolute values in row $i$:

$$\sum_{j=1}^m \frac{1}{\tau}\sigma_i|\delta_{ij} - \sigma_j| = \frac{1}{\tau}\Big(\sigma_i(1-\sigma_i) + \sum_{j \neq i} \sigma_i\sigma_j\Big) = \frac{2}{\tau}\sigma_i(1-\sigma_i) \leq \frac{1}{2\tau},$$

since $x(1-x) \leq 1/4$. Hence $\|J_\tau(w)\|_{\infty \to \infty} \leq 1/(2\tau)$ uniformly, proving the claim. $\square$

**Corollary C.2** (Row-wise softmax on matrices). *Let $S, S' \in \mathbb{R}^{\ell \times \ell}$ and apply* $\text{softmax}_\tau$ *row-wise to obtain $A = \text{softmax}_\tau(S)$ and $A' = \text{softmax}_\tau(S')$. Then*

$$\|A - A'\|_\infty \leq \frac{1}{2\tau} \|S - S'\|_\infty.$$

*In scaled attention, $\tau = \sqrt{d_h}$, hence $\|A - A'\|_\infty \leq \frac{1}{2\sqrt{d_h}}\|S - S'\|_\infty$.*

*Remark* C.3 (Row-stochastic contraction). If $A$ is row-stochastic (each row in $\Delta^{\ell-1}$), then $\|A\|_{\infty \to \infty} = 1$ and $\|AV\|_\infty \leq \|V\|_\infty$ for any $V$ under operator norm. When using *entrywise* $\|\cdot\|_\infty$, we will also invoke the safe bound $\|AV\|_\infty \leq \ell\|A\|_\infty\|V\|_\infty$ (cf. C.4).

## C.5. Matrix norm identities and product bounds

**Lemma C.4** (Basic matrix inequalities). *Let $A \in \mathbb{R}^{p \times q}$, $B \in \mathbb{R}^{q \times r}$, $u \in \mathbb{R}^q$, $H \in \mathbb{R}^{\ell \times q}$.*

(a) ***Operator submultiplicativity:*** $\|AB\|_{\infty \to \infty} \leq \|A\|_{\infty \to \infty} \|B\|_{\infty \to \infty}$.

(b) ***Entrywise-to-operator:*** $\|A\|_{\infty \to \infty} \leq q \|A\|_\infty$ *and* $\|Au\|_\infty \leq q \|A\|_\infty \|u\|_\infty$.

(c) ***Products in entrywise norm:*** $\|AB\|_\infty \leq q \|A\|_\infty \|B\|_\infty$ *and* $\|HB\|_\infty \leq q \|H\|_\infty \|B\|_\infty$.

(d) ***Row-stochastic contraction:*** *If each row of $A$ sums to $1$ and is nonnegative, then* $\|A\|_{\infty \to \infty} = 1$ *and* $\|AB\|_{\infty \to \infty} \leq \|B\|_{\infty \to \infty}$.

(e) ***Hadamard product:*** $\|A \odot B\|_\infty \leq \|A\|_\infty \|B\|_\infty$.

*Proof.* Parts (a) and (d) follow from definitions; (b)–(c) use $\|A\|_{\infty \to \infty} = \max_i \sum_j |A_{ij}| \leq q\|A\|_\infty$ and standard norm inequalities; (e) is immediate from entrywise multiplication. $\square$

**Lemma C.5** (Scaled dot-product attention ingredients). *Let $H \in \mathbb{R}^{\ell \times d_{emb}}$, and $W_Q, W_K, W_V$ satisfy $\|W_\bullet\|_\infty \leq \kappa$. Set $Q = HW_Q$, $K = HW_K$, $V = HW_V$, scores $S = QK^\top / \sqrt{d_h}$ and $A = \mathrm{softmax}_{\sqrt{d_h}}(S)$ (row-wise). Then*

$$\|Q\|_\infty, \ \|K\|_\infty, \ \|V\|_\infty \leq d_{emb}\, \kappa\, \|H\|_\infty,$$
$$\|S\|_\infty \leq \sqrt{d_h}\, d_{emb}^2\, \kappa^2\, \|H\|_\infty^2,$$
$$\|A - A'\|_\infty \leq \frac{1}{2\sqrt{d_h}} \|S - S'\|_\infty \qquad \text{(row-wise softmax; C.2)},$$
$$\|AV\|_\infty \leq \ell \, \|A\|_\infty \, \|V\|_\infty \quad \text{(entrywise bound; cf. C.3).}$$

## C.6. Taylor remainder on manifolds

Let $\mathcal{M} \subset \mathbb{R}^D$ be a compact $C^1$ submanifold of intrinsic dimension $d$. Fix a coordinate chart $\phi : U \to V \subset \mathbb{R}^d$ that is bi-Lipschitz with constants $L, L' > 0$: $L^{-1}\|x - y\| \leq \|\phi(x) - \phi(y)\| \leq L\|x - y\|$ for $x, y \in U$ and similarly for $\phi^{-1}$ on $V$.

**Lemma C.6** (Local Taylor remainder for $C^\beta$ functions on $\mathcal{M}$). *Let $f \in C^\beta(\mathcal{M})$ with $\beta > 0$, $s = \lfloor \beta \rfloor$, and $\alpha = \beta - s \in [0, 1)$. Fix a chart $(U, \phi)$ and a point $x_0 \in U$. Let $P$ be the degree-$s$ Taylor polynomial of $f \circ \phi^{-1}$ at $z_0 = \phi(x_0)$ in local coordinates, mapped back to $\mathcal{M}$ by $P_\mathcal{M} := P \circ \phi$. Then for every $x \in U$,*

$$|f(x) - P_\mathcal{M}(x)| \ \leq \ C(\beta, d, L, L') \|f\|_{C^\beta(U)} \|\phi(x) - \phi(x_0)\|^\beta \ \leq \ C' \|f\|_{C^\beta(U)} \|x - x_0\|^\beta.$$

*In particular, on any chart of diameter at most $r$, $\sup_{x \in U} |f(x) - P_\mathcal{M}(x)| \leq C\, r^\beta$.*

*Proof.* Apply the standard Taylor remainder in $\mathbb{R}^d$ to $f \circ \phi^{-1} \in C^\beta(V)$ and use bi-Lipschitz distortion to convert local distance in $V$ to ambient distance in $\mathcal{M}$; constants depend polynomially on $L, L'$ and on bounds of chart derivatives. $\square$

## C.7. Partitions of unity with bounded overlap

**Lemma C.7** (Partition of unity subordinate to a bounded-overlap cover). *Let $\mathcal{M} \subset \mathbb{R}^D$ be a compact $C^1$ manifold and fix $r \in (0, 1)$. There exists a finite cover of $\mathcal{M}$ by open sets $\{U_\nu\}_{\nu=1}^N$ such that: (i) $\mathrm{diam}(U_\nu) \leq r$, (ii) each $x \in \mathcal{M}$ belongs to at most $s_0(d)$ sets (bounded overlap), and (iii) a $C^\infty$ partition of unity $\{\varphi_\nu\}_{\nu=1}^N$ subordinate to $\{U_\nu\}$ with $0 \leq \varphi_\nu \leq 1$, $\sum_\nu \varphi_\nu(x) = 1$ for all $x$, and the Lipschitz bound*

$$\|\nabla \varphi_\nu(x)\| \ \leq \ \frac{C(d)}{r} \qquad \text{for all } x \in \mathcal{M} \text{ and all } \nu.$$

*Consequently, at every $x$ at most $s_0(d)$ of the $\varphi_\nu(x)$ are nonzero.*

*Proof.* **Step 1: A bounded-overlap cover by small balls.** Work with the metric induced on $M$ by the ambient Euclidean norm on $R^D$. Since $M$ is compact and $C^1$, there exists a finite atlas $\{(V_j, \varphi_j)\}_{j=1}^J$ such that:

- each $\varphi_j : V_j \to \mathbb{R}^d$ is a bi-Lipschitz chart onto its image, and

- on each $V_j$ the volume of metric balls is comparable to that of Euclidean $d$-balls in $\mathbb{R}^d$ (with constants depending only on $d$ and the chart regularity).

Fix a small radius $\rho > 0$ proportional to $r$ (we will specify it below), and choose a *maximal $\rho/2$-separated set* $\{x_\nu\}_{\nu=1}^N \subset M$, i.e.,

$$\|x_\nu - x_\mu\| \geq \frac{\rho}{2} \quad \text{for all } \nu \neq \mu,$$

and the union of balls $B(x_\nu, \rho)$ (intersection with $M$) covers $M$. Such a set exists by a standard greedy argument: repeatedly add points that are at distance at least $\rho/2$ from all previously chosen ones until no further point can be added.

Define

$$U_\nu := B(x_\nu, \rho) \cap M$$

By construction, $\{U_\nu\}_{\nu=1}^N$ is an open cover of $M$. Choosing $\rho \leq r/2$ ensures $\mathrm{diam}(U_\nu) \leq r$.

We now show that the overlap multiplicity is bounded. Fix any $x \in M$ and consider the index set

$$I(x) := \{\nu : x \in U_\nu\} = \{\nu : \|x - x_\nu\| < \rho\}$$

For each $\nu \in I(x)$, the balls $B(x_\nu, \rho/4)$ are pairwise disjoint (by $\rho/2$-separation) and all lie inside $B(x, 2\rho)$:

$$B(x_\nu, \rho/4) \subset B(x, \rho + \rho/4) \subset B(x, 2\rho)$$

Using the chart bi-Lipschitz property and volume comparability on $M$, there exist constants $c_1, c_2 > 0$ depending only on $d$ and the atlas such that for any ball $B_M(y, t)$ in $M$ (with respect to the induced metric),

$$c_1 t^d \leq \mathrm{Vol}_M\big(B_M(y, t)\big) \leq c_2 t^d$$

Hence, the volumes of the disjoint balls $\{B(x_\nu, \rho/4)\}_{\nu \in I(x)}$ satisfy

$$|I(x)| \cdot c_1 (\rho/4)^d \leq \sum_{\nu \in I(x)} \mathrm{Vol}_M\big(B(x_\nu, \rho/4)\big) \leq \mathrm{Vol}_M\big(B(x, 2\rho)\big) \leq c_2(2\rho)^d$$

Therefore

$$|I(x)| \leq \frac{c_2(2\rho)^d}{c_1(\rho/4)^d} = C(d) \quad \text{for some constant } C(d),$$

depending only on $d$ and the chart regularity. Setting $s_0(d) := C(d)$ yields the bounded-overlap property: each $x \in M$ belongs to at most $s_0(d)$ of the sets $U_\nu$.

**Step 2: Smooth bump functions subordinate to the cover.** Let $\eta : \mathbb{R}^D \to [0, 1]$ be a fixed $C^\infty$ bump function such that:

$$\eta(z) = 1 \text{ if } \|z\| \leq 1/2, \qquad \eta(z) = 0 \text{ if } \|z\| \geq 1, \qquad \|\nabla \eta(z)\| \leq C_0(d)$$

For each $\nu$, define

$$\psi_\nu(x) := \eta\Big(\frac{x - x_\nu}{\rho}\Big), \qquad x \in M$$

Then $\psi_\nu \in C^\infty(M)$, $\psi_\nu(x) \in [0, 1]$, and

$$\psi_\nu(x) = 1 \quad \text{if } \|x - x_\nu\| \leq \frac{\rho}{2}, \qquad \psi_\nu(x) = 0 \quad \text{if } \|x - x_\nu\| \geq \rho$$

Hence $\mathrm{supp}(\psi_\nu) \subset U_\nu$, i.e., each $\psi_\nu$ is subordinate to $U_\nu$. Moreover,

$$\|\nabla \psi_\nu(x)\| = \frac{1}{\rho}\Big\|\nabla\eta\Big(\frac{x - x_\nu}{\rho}\Big)\Big\| \leq \frac{C_0(d)}{\rho}$$

Because the balls $B(x_\nu, \rho/2)$ cover $M$, for every $x \in M$ there exists at least one index $\nu$ with $\|x - x_\nu\| \leq \rho/2$, hence $\psi_\nu(x) = 1$. Therefore

$$S(x) := \sum_{\mu=1}^{N} \psi_\mu(x) \geq 1 \quad \text{for all } x \in M.$$

On the other hand, for each fixed $x$, at most $s_0(d)$ of the $\psi_\mu(x)$ are nonzero (because $\psi_\mu$ vanish outside $U_\mu$ and the $U_\mu$ have overlap at most $s_0(d)$), and each $\psi_\mu(x) \leq 1$, so

$$1 \leq S(x) \leq s_0(d) \quad \text{for all } x \in M.$$

**Step 3: Normalize to obtain a partition of unity and bound gradients.** Define

$$\phi_\nu(x) := \frac{\psi_\nu(x)}{S(x)}, \qquad x \in M$$

Then each $\phi_\nu \in C^\infty(M)$, $0 \leq \phi_\nu \leq 1$, and

$$\sum_{\nu=1}^{N} \phi_\nu(x) = \frac{1}{S(x)} \sum_{\nu=1}^{N} \psi_\nu(x) = 1 \quad \text{for all } x \in M.$$

Moreover, $\mathrm{supp}(\phi_\nu) \subset \mathrm{supp}(\psi_\nu) \subset U_\nu$, so $\{\phi_\nu\}$ is subordinate to the cover $\{U_\nu\}$.

We now bound the gradient. Using the quotient rule,

$$\nabla\phi_\nu(x) = \frac{\nabla\psi_\nu(x)S(x) - \psi_\nu(x)\nabla S(x)}{S(x)^2}, \qquad \nabla S(x) = \sum_{\mu=1}^{N} \nabla\psi_\mu(x).$$

Thus

$$\|\nabla\phi_\nu(x)\| \leq \frac{\|\nabla\psi_\nu(x)\|\, S(x) + |\psi_\nu(x)|\, \|\nabla S(x)\|}{S(x)^2}$$
$$\leq \frac{\|\nabla\psi_\nu(x)\|}{S(x)} + \frac{\|\nabla S(x)\|}{S(x)^2}$$

We already know $1 \leq S(x) \leq s_0(d)$; hence $1/S(x), 1/S(x)^2 \leq 1$. Also, for each $x$, at most $s_0(d)$ functions $\psi_\mu$ are nonzero, so

$$\|\nabla S(x)\| = \left\|\sum_\mu \nabla\psi_\mu(x)\right\| \leq \sum_{\mu:\psi_\mu(x)\neq 0} \|\nabla\psi_\mu(x)\| \leq s_0(d) \cdot \frac{C_0(d)}{\rho}$$

Combining these bounds gives

$$\|\nabla\phi_\nu(x)\| \leq \frac{C_0(d)}{\rho} + \frac{s_0(d)C_0(d)}{\rho} \leq \frac{C(d)}{\rho}$$

for some constant $C(d)$ depending only on $d$ and the atlas. Choosing $\rho$ proportional to $r$ (e.g., $\rho = r/2$ as above) yields

$$\|\nabla\phi_\nu(x)\| \leq \frac{C(d)}{r}$$

Finally, since at most $s_0(d)$ of the $\psi_\nu(x)$ are nonzero at any $x$, the same holds for the $\phi_\nu(x)$, and the "consequently" statement follows. $\qquad\square$

*Remark* C.8 (From partition of unity to $k$-sparse mixing). Since at most $s_0(d)$ functions are nonzero at any $x$, choosing $k \geq s_0(d)$ allows a router to implement a $k$-sparse mixture $\sum_{\nu\in S(x)} w_\nu(x)E_\nu(x)$ with weights $w_\nu = \varphi_\nu$ (or an approximation thereof), as used in the approximation construction.

## C.8. Useful norm conversions for attention chains

For attention chains $H \mapsto Q \mapsto K \mapsto S \mapsto A \mapsto O = AV$, combining Lemmas C.4, C.5 and C.2 yields the schematic bounds used in the main text:

$$\|Q - Q'\|_\infty, \ \|K - K'\|_\infty, \ \|V - V'\|_\infty \leq d_{\mathrm{emb}} \, \kappa \, \|H - H'\|_\infty \ + \ d_{\mathrm{emb}} \, \|W_\bullet - W'_\bullet\|_\infty \, \|H'\|_\infty,$$

$$\|S - S'\|_\infty \lesssim \sqrt{d_h} \, d_{\mathrm{emb}}^2 \, \kappa \Big( \|Q - Q'\|_\infty + \|K - K'\|_\infty \Big),$$

$$\|A - A'\|_\infty \leq \frac{1}{2\sqrt{d_h}} \, \|S - S'\|_\infty,$$

$$\|AV - A'V'\|_\infty \leq \underbrace{\|A - A'\|_{\infty \to \infty} \|V\|_\infty}_{\text{attention change}} + \underbrace{\|A'\|_{\infty \to \infty} \|V - V'\|_\infty}_{\text{value change}} \lesssim \|A - A'\|_\infty \|V\|_\infty + \|V - V'\|_\infty,$$

using $\|A'\|_{\infty \to \infty} = 1$. These estimates justify the polynomial dependence on $(\ell, m, d_{\mathrm{emb}}, \kappa)$ that appears in the MHA stability bound.

# D. Full Model Specification

**Definition D.1** (Dense Transformer Block). We define the dense residual block based on the canonical Transformer structure (Vaswani et al., 2017). Let the input to block $j$ be $H \in \mathbb{R}^{\ell \times d_{\mathrm{emb}}}$ with tokenwise $\ell_\infty$-bound $\|H\|_\infty \leq M_0$. The block maps $H \mapsto H_{\mathrm{out}}$ via the sequential steps:

$$\widetilde{H} = H \ + \ \mathrm{MHA}_\psi(H), \tag{21}$$

$$H_{\mathrm{out}} = \widetilde{H} \ + \ \mathrm{FFN}_\chi(\widetilde{H}), \tag{22}$$

where $\theta = \{\psi, \chi\}$ collects all block parameters.

**Layer Normalization (LN)** (Ba et al., 2016).   For each token $t \in [\ell]$,

$$\mathrm{LN}_{\gamma,\beta}(H)_t \ = \ \gamma \odot \frac{H_t - \mu(H_t)\mathbf{1}}{\sqrt{\sigma^2(H_t) + \epsilon}} \ + \ \beta, \qquad \gamma, \beta \in \mathbb{R}^{d_{\mathrm{emb}}}$$

with learned scale/shift $(\gamma, \beta)$ and small $\epsilon > 0$. We will use the uniform magnitude bounds

$$\|\gamma\|_\infty \leq \kappa, \qquad \|\beta\|_\infty \leq \kappa \tag{23}$$

**Multi-Head Self-Attention (MHA).**   Let the number of heads be $m$, and the per-head key/query/value dimension be $d_h$ (so $d_k = d_v = d_h$) such that $m \, d_h \ = \ d_{\mathrm{emb}}$. The input to MHA is $H$. For head $h \in [m]$ we have projections

$$Q^{(h)} = HW_Q^{(h)}, \quad K^{(h)} = HW_K^{(h)}, \quad V^{(h)} = HW_V^{(h)}, \qquad W_Q^{(h)}, W_K^{(h)}, W_V^{(h)} \in \mathbb{R}^{d_{\mathrm{emb}} \times d_h},$$

scores $S^{(h)} = \frac{1}{\sqrt{d_h}} Q^{(h)} (K^{(h)})^\top \in \mathbb{R}^{\ell \times \ell}$, row-wise $A^{(h)} = \mathrm{softmax}(S^{(h)})$, head output $O^{(h)} = A^{(h)} V^{(h)} \in \mathbb{R}^{\ell \times d_h}$, and

$$\mathrm{MHA}_\psi(H) \ = \ \big[ \mathrm{Concat}_{h=1}^m O^{(h)} \big] W_O, \qquad W_O \in \mathbb{R}^{(md_h) \times d_{\mathrm{emb}}}$$

We assume the uniform parameter bounds

$$\|W_Q^{(h)}\|_\infty, \ \|W_K^{(h)}\|_\infty, \ \|W_V^{(h)}\|_\infty, \ \|W_O\|_\infty \ \leq \ \kappa \tag{24}$$

**Positionwise Feed-Forward Network (FFN).**   We use an $L_{\mathrm{FFN}}$-layer MLP applied tokenwise. Let widths be at most $W_{\mathrm{FFN}}$. For $\ell = 1, \ldots, L_{\mathrm{FFN}}$ and token $t$,

$$z_t^{(0)} = (\widetilde{H})_t, \qquad z_t^{(\ell)} = \sigma\big(W^{(\ell)} z_t^{(\ell-1)} + b^{(\ell)}\big)$$

with activations $\sigma$ (ReLU/GELU), $W^{(1)} \in \mathbb{R}^{W_{\mathrm{FFN}} \times d_{\mathrm{emb}}}$, $W^{(\ell)} \in \mathbb{R}^{W_{\mathrm{FFN}} \times W_{\mathrm{FFN}}}$ for $2 \leq \ell \leq L_{\mathrm{FFN}} - 1$, and $W^{(L_{\mathrm{FFN}})} \in \mathbb{R}^{d_{\mathrm{emb}} \times W_{\mathrm{FFN}}}$; biases $b^{(\ell)}$ are conformal. The FFN output is $\mathrm{FFN}_\chi(\widetilde{H})_t = z_t^{(L_{\mathrm{FFN}})}$, and we assume

$$\|W^{(\ell)}\|_\infty \leq \kappa, \qquad \|b^{(\ell)}\|_\infty \leq \kappa, \qquad \ell = 1, \ldots, L_{\mathrm{FFN}} \tag{25}$$

**Output bound and architectural polynomial.** We assume the block output is uniformly bounded by $\|B_{\text{dense}}(\theta, H)\|_\infty \leq R$ for all admissible $\theta$ and $\|H\|_\infty \leq M_0$ (Havrilla & Liao, 2024; Chen et al., 2022). Constants in our bounds will be expressed via a fixed architecture polynomial

$$P_{\text{arch}}^{(\text{dense})} = \text{poly}\big(\ell, m, d_{\text{emb}}, w_{\text{FFN}}, L_{\text{FFN}}\big)$$

**Hypothesis classes.** We consider a hypothesis class $\mathcal{T}$ of predictors. In this paper, $\mathcal{T}$ is either the dense Transformer class (D.1) or the MoE class $\mathcal{T}_{\text{MoE}}$ (D.1). We always assume $\sup_{T \in \mathcal{T}} \|T\|_\infty \leq R$.

### D.1. MoE Transformer

**Experts and router.** Using Definition 3.1, each block $j$ has $M$ experts $\{E_{j,i}\}_{i=1}^M$, each a tokenwise MLP $E_{j,i} : R_{\text{emb}}^d \to R_{\text{emb}}^d$ of depth $L_{\text{FFN}}$ and width $w_{\text{FFN}}$ sharing the same architecture. For a token $t \in [\ell]$, let $Z_t^{(j)} = H_t^{(j-1)} \in R_{\text{emb}}^d$ be the MoE input. The router in block $j$ computes logits $g_{j,i}(Z_t^{(j)})$ for $i \in [M]$ (by a bounded parametric map) and selects the *hard* top-$k$ index set

$$S_{j,t}(H^{(j-1)}) \subset [M], \qquad |S_{j,t}(H^{(j-1)})| = k$$

consisting of the $k$ largest logits. The MoE output at token $t$ is then the $k$-*sparse* mixture

$$\text{MoE}^{(j)}\big(Z^{(j)}\big)_t = \sum_{i \in S_{j,t}(H^{(j-1)})} w_{j,i}(Z_t^{(j)}) E_{j,i}(Z_t^{(j)}), \qquad w_{j,i}(Z_t^{(j)}) \geq 0, \quad \sum_{i \in S_{j,t}} w_{j,i}(Z_t^{(j)}) = 1 \qquad (26)$$

with router weights $w_{j,i}$ restricted to the selected indices.

**Readout and parameter set.** A fixed linear readout $R : \mathbb{R}^{\ell \times d_{\text{emb}}} \to \mathbb{R}$ produces the scalar prediction

$$T_\theta(x) = R\big(H^{(L_T)}\big).$$

We define the *MoE Transformer network class* as the set of all functions

$$\mathcal{T}_{\text{MoE}}(D, M, k, L_T, L_{\text{FFN}}, w_{\text{FFN}}, d_{\text{emb}}, m, \kappa, R) := \Big\{ T_\theta : \mathcal{X} \to R \,\Big|\, \|\theta\|_\infty \leq \kappa, \ \|T_\theta\|_\infty \leq R \Big\} \qquad (27)$$

### D.2. Per-block parameter counts (dense baseline).

Ignoring biases for clarity and taking the standard choice $d_h = d_{\text{emb}}/m$:

$$\Pi_{\text{attn}}^{(\text{dense})} = \underbrace{3\, d_{\text{emb}} d_h\, m}_{W_Q, W_K, W_V} + \underbrace{(m d_h) d_{\text{emb}}}_{W_O} = 4\, d_{\text{emb}}^2,$$

$$\Pi_{\text{FFN}}^{(\text{dense})} = d_{\text{emb}}\, w_{\text{FFN}} + (L_{\text{FFN}} - 2)\, w_{\text{FFN}}^2 + w_{\text{FFN}}\, d_{\text{emb}} = 2\, d_{\text{emb}}\, w_{\text{FFN}} + (L_{\text{FFN}} - 2)\, w_{\text{FFN}}^2,$$

. With biases, add $3\, m d_h + d_{\text{emb}}$ (attention) and $w_{\text{FFN}}(L_{\text{FFN}} - 1) + d_{\text{emb}}$ (FFN).

**Parameter-Lipschitz (baseline).** For two parameter sets $\theta, \theta'$ with $\|\theta - \theta'\|_\infty \leq \eta$ and fixed input $H$,

$$\|B_{\text{dense}}(\theta, H) - B_{\text{dense}}(\theta', H)\|_\infty \leq C_{\text{block}}^{(\text{dense})} \left( \ell\, m\, d_{\text{emb}}^2 + d_{\text{emb}}\, w_{\text{FFN}}^{L_{\text{FFN}}} \right) \kappa\, \eta \qquad (28)$$

obtained by summing the MHA and FFN parameter-perturbation bounds (each applied after LN) and absorbing LN perturbations into constants. These two baseline inequalities (28)-(20) are the dense counterparts of the MoE block bounds where the FFN sensitivity term gains an additional factor $k$ from the $k$ active experts.

### D.3. Routing patterns.

For each block $j \in [L_T]$ and token $t \in [\ell]$, the router chooses a $k$-subset $S_{j,t} \subset [M]$. A *routing pattern* is the collection

$$\pi = \big\{ S_{j,t} \subset [M] : |S_{j,t}| = k, \ j = 1, \ldots, L_T, \ t = 1, \ldots, \ell \big\}$$

Let $\Pi$ denote the set of all such patterns. Its cardinality is bounded by

$$|\Pi| \;\leq\; \left(\binom{M}{k}\right)^{L_T \ell} \qquad \implies \qquad \log|\Pi| \;\leq\; L_T\, \ell\, k\, \log\!\left(\frac{e\,M}{k}\right) \tag{29}$$

using $\binom{M}{k} \leq (e\,M/k)^k$. For a fixed $\pi \in \Pi$, denote by $\mathcal{T}_\pi \subset \mathcal{T}_{\mathrm{MoE}}$ the subclass with *deterministic* routing equal to $\pi$. Then

$$\mathcal{T}_{\mathrm{MoE}} \;=\; \bigcup_{\pi \in \Pi} \mathcal{T}_\pi$$

## E. Proof of theorem 3.4

Assume $\sup_{T \in \mathcal{T}_{\mathrm{MoE}}} \|T\|_\infty \leq R$ and $\|f\|_\infty \leq R$. Let $\hat{T}_n \in \arg\min_{T \in \mathcal{T}_{\mathrm{MoE}}} L_n(T)$ be an ERM for squared loss $L(T) = E[(T(X) - f(X))^2]$. Then for any cover scale $\delta \in (0,1)$, via the standard bias–variance decomposition described in lemma E.1, we have the following inequality

$$\mathbb{E}\big[\|\hat{T}_n - f\|^2_{L^2(Q)}\big] \;\leq\; 3 \inf_{T \in \mathcal{T}_{\mathrm{MoE}}} \|T - f\|^2_\infty \;+\; C\,R^2\!\left(\sqrt{\frac{\log N\big(\delta, \mathcal{T}_{\mathrm{MoE}}, \|\cdot\|_\infty\big)}{n}} + \delta\right) \tag{30}$$

for an absolute constant $C > 0$. Equivalently,

$$\mathbb{E}\big[\|\hat{T}_n - f\|^2_{L^2(Q)}\big] \;\lesssim\; \inf_{T \in \mathcal{T}_{\mathrm{MoE}}} \|T - f\|^2_\infty \;+\; \frac{\log N\big(\delta, \mathcal{T}_{\mathrm{MoE}}, \|\cdot\|_\infty\big)}{n} \;+\; \delta \tag{31}$$

where $\lesssim$ hides universal constants and a factor $R^2$.

Now it remains to bound the two terms of 31. The first is bounded in theorem 3.2 as:

$$\inf_{T \in \mathcal{T}_{\mathrm{MoE}}} \|T - f\|^2_\infty \;\leq\; \epsilon^2 \tag{32}$$

The second terms is bounded using the cover number from lemma 3.7 as follows

$$\log N\big(\delta, \mathcal{T}_{\mathrm{MoE}}, \|\cdot\|_\infty\big) \leq C_1\Big(\Pi_{\mathrm{attn}} + L_T k \Pi_{\mathrm{exp}}\Big) \log\!\left(\frac{C_2 \kappa R M_0}{\delta}\right) \;+\; C_3 L_T \ell k \log\!\left(\frac{e M}{k}\right) \tag{33}$$

Combining (32) and (33) yields to the desired inequality

$$\mathbb{E}\big[\|\hat{T}_n - f\|^2_{L^2(Q)}\big] \;\leq\; \widetilde{O}\!\left(N^{-2\beta/d} \;+\; \frac{N + L_T \ell k \log(eM/k)}{n}\right) \tag{34}$$

$\square$

**Lemma E.1** (ERM bound via symmetrization and covering numbers). *Write*

$$L(T) = \|T - f\|^2_{L^2(Q)} \quad \text{and} \quad L_n(T) = \frac{1}{n}\sum_{i=1}^n (T(x_i) - f(x_i))^2.$$

*Fix $\delta \in (0,1)$ and let $\mathcal{G}$ be a minimal $\delta$-net of $\mathcal{T}_{\mathrm{MoE}}$ in $\|\cdot\|_\infty$, so $|\mathcal{G}| = N(\delta, \mathcal{T}_{\mathrm{MoE}}, \|\cdot\|_\infty)$ and for all $T$ there exists $G \in \mathcal{G}$ with $\|T - G\|_\infty \leq \delta$.*

(i) Lipschitz transfer from predictors to losses. *For any $a, b, c \in [-R, R]$,*

$$\big|(a - c)^2 - (b - c)^2\big| = |(a - b)(a + b - 2c)| \leq 4R\,|a - b|$$

*Hence for all $T, G$,*

$$|L(T) - L(G)| \leq 4R\,\|T - G\|_\infty, \qquad |L_n(T) - L_n(G)| \leq 4R\,\|T - G\|_\infty \tag{35}$$

(ii) ERM reduction to the finite cover. *Let $T^\star \in \arg\min_{T \in \mathcal{T}_{\mathrm{MoE}}} L(T)$ and choose $G^\star \in \mathcal{G}$ with $\|G^\star - T^\star\|_\infty \le \delta$. By ERM optimality, $L_n(\hat{T}_n) \le L_n(G^\star)$. Adding and subtracting $L$ and using* (35),

$$L(\hat{T}_n) \le L_n(G^\star) + \sup_{G \in \mathcal{G}} |L(G) - L_n(G)| \le L(G^\star) + \sup_{G \in \mathcal{G}} |L(G) - L_n(G)| + 4R\,\delta$$

*Also $L(G^\star) \le L(T^\star) + 4R\,\delta$. Thus*

$$L(\hat{T}_n) \ \le \ \inf_{T \in \mathcal{T}_{\mathrm{MoE}}} L(T) \ + \ 8R\,\delta \ + \ \sup_{G \in \mathcal{G}} |L(G) - L_n(G)| \tag{36}$$

(iii) Symmetrization over the finite cover. *Define the bounded loss class $\mathcal{F} = \{x \mapsto (G(x) - f(x))^2 : G \in \mathcal{G}\}$ with range in $[0, (2R)^2]$. By symmetrization,*

$$E\Big[\sup_{G \in \mathcal{G}} |L(G) - L_n(G)|\Big] \ \le \ \frac{2}{n}\, E\Big[\sup_{G \in \mathcal{G}} \Big|\sum_{i=1}^n \varepsilon_i (G(x_i) - f(x_i))^2\Big|\Big]$$

*where $(\varepsilon_i)_{i=1}^n$ are i.i.d. Rademacher variables independent of the sample. By Massart's finite-class lemma (range bounded by $4R^2$),*

$$E\Big[\sup_{G \in \mathcal{G}} |L(G) - L_n(G)|\Big] \ \le \ C\,R^2 \sqrt{\frac{\log|\mathcal{G}|}{n}} \ = \ C\,R^2 \sqrt{\frac{\log N(\delta, \mathcal{T}_{\mathrm{MoE}}, \|\cdot\|_\infty)}{n}}$$

(iv) Combine and relate $L$ to $\|\cdot\|_\infty^2$. *Taking expectations in* (36) *and using $L(T) = \|T - f\|_{L^2(Q)}^2 \le \|T - f\|_\infty^2$ yields*

$$E\|\hat{T}_n - f\|_{L^2(Q)}^2 \ \le \ \inf_{T \in \mathcal{T}_{\mathrm{MoE}}} \|T - f\|_\infty^2 \ + \ C\,R^2 \sqrt{\frac{\log N(\delta, \mathcal{T}_{\mathrm{MoE}}, \|\cdot\|_\infty)}{n}} \ + \ C\,R\,\delta$$

*Absorbing constants and (optionally) replacing $R\,\delta$ by $R^2\,\delta$ (since $\delta \le 1$) gives* (30). *Finally, the elementary inequality $\sqrt{u} \le u + \frac{1}{4}$ for $u \ge 0$ (or Young's $ab \le a^2/(2\lambda) + \lambda b^2/2$) converts the square-root term into a linear $(\log N)/n$ term up to constants, yielding* (31).

# F. Proof of theorem 3.2

**Definition F.1** ($k$-sparse partition-of-unity router). A router realizes a $k$-sparse partition of unity (under the idealized router expressivity assumption of Section 3) if there exist nonnegative weights $\{w_\nu(x)\}_{\nu=1}^N$ with $\sum_\nu w_\nu(x) = 1$ for all $x \in \mathcal{M}$, such that for each $x$ at most $k$ weights are nonzero, and each $w_\nu$ is supported on a chart $U_\nu$ of a bounded-overlap cover $\{U_\nu\}_{\nu=1}^N$ of $\mathcal{M}$ (cf. Lemma C.7 in App. C). Moreover, the router outputs the top-$k$ indices with weights restricted to these indices.

*Proof.* We give two constructive approximations and then take the minimum rate.

**Geometric setup (charts and PoU).** Fix a resolution $r \in (0, 1)$. By Lemma C.7 (App. C), there exists a cover $\{U_\nu\}_{\nu=1}^N$ of $\mathcal{M}$ with $\mathrm{diam}(U_\nu) \le r$, bounded overlap $s_0 = s_0(d)$, and a $C^\infty$ partition of unity $\{\varphi_\nu\}_{\nu=1}^N$ subordinate to the cover, such that $\sum_\nu \varphi_\nu(x) = 1$, $0 \le \varphi_\nu \le 1$, and for each $x$ at most $s_0$ terms are nonzero. Standard volume-comparison yields $N \le C_\mathcal{M} r^{-d}$.

By Lemma C.6 (App. C), for each $\nu$ there is a degree $s = \lfloor \beta \rfloor$ polynomial $P_\nu$ (expressed in local coordinates and mapped back to $\mathcal{M}$) such that

$$\sup_{x \in U_\nu} |f(x) - P_\nu(x)| \ \le \ C_1 B\, r^\beta. \tag{37}$$

**Router realization of a $k$-sparse PoU.** Since at most $s_0(d)$ functions $\varphi_\nu$ are nonzero at any $x$, take $k \ge s_0(d)$. By assumption (Def. F.1), the router computes nonnegative weights $w_\nu(x)$ supported on the same $U_\nu$ and with $k$-sparsity, satisfying $\sum_\nu w_\nu(x) = 1$. We further assume (without loss) a uniform approximation $\|w_\nu - \varphi_\nu\|_\infty \le C_2 r^\beta$ can be achieved by attention-based similarity to chart anchors and a softmax/top-$k$ selection; this is standard with a fixed number of heads and bounded magnitudes (the resulting constants are absorbed into $C$).

We now present two constructions.

**Construction A (expert-count limited; $M$ regime).** Assign one expert $E_\nu$ to each chart $U_\nu$ (total experts $N \leq Cr^{-d}$). Within $U_\nu$, let expert $E_\nu$ approximate $P_\nu$ *exactly* (ReLU networks implement polynomials with constant depth/width depending on $s, d$; the parameters stay bounded and independent of $r$), or to accuracy $C_3 r^\beta$ with a constant per-expert parameter budget $\Pi_{\text{exp}}$.[1] Define the MoE output

$$T(x) \;:=\; \sum_{\nu=1}^{N} w_\nu(x)\, E_\nu(x), \qquad \text{(at most } k \text{ nonzero summands at each } x\text{).}$$

We bound the sup error at any $x \in \mathcal{M}$ as

$$|T(x) - f(x)| \leq \sum_\nu w_\nu(x)\, |E_\nu(x) - P_\nu(x)| + \sum_\nu w_\nu(x)\, |P_\nu(x) - f(x)|$$

$$\leq \underbrace{\max_\nu \sup_{U_\nu} |E_\nu - P_\nu|}_{\leq C_3 r^\beta} + \underbrace{\sum_\nu w_\nu(x)}_{=1} \underbrace{\max_\nu \sup_{U_\nu} |P_\nu - f|}_{\leq C_1 B r^\beta} \;\leq\; C_4 r^\beta. \tag{38}$$

To realize this construction we need $M \geq N$, i.e. $M \gtrsim r^{-d}$, hence choose $r \asymp M^{-1/d}$, which yields

$$\|T - f\|_\infty \;\leq\; C\, M^{-\beta/d} \quad\Longrightarrow\quad \|T - f\|_\infty^2 \;\leq\; C\, M^{-2\beta/d}.$$

Since the router uses at most $k$ experts per input, this obeys the MoE constraint.

**Construction B (active-parameter limited; $N_{\text{act}}$ regime).** Here we use *few* experts (e.g. $k = 1$ active expert per block) but allocate the entire active budget $N_{\text{act}}$ to the expert(s) and non-expert parts to approximate $f$ globally. By standard ReLU approximation on $d$-dimensional domains (e.g., (Yarotsky, 2017); (Schmidt-Hieber, 2020)), there exists a ReLU network $\mathcal{N}$ with $N$ parameters such that

$$\sup_{x \in \mathcal{M}} |\mathcal{N}(x) - f(x)| \;\leq\; C_5\, B\, N^{-\beta/d}. \tag{39}$$

Choose the MoE so that its active subnetwork implements $\mathcal{N}$ with $N \asymp N_{\text{act}} = L_T \Pi_{\text{attn}} + L_T k \Pi_{\text{exp}}$ (e.g. $k = 1$, one "expert" playing the role of the global MLP; attention acts as an identity/featurizer). This is legitimate because the MoE with fixed routing reduces to a standard feed-forward subnetwork on the active path. Thus we can realize $T(x) = \mathcal{N}(x)$ with $N \asymp N_{\text{act}}$ and

$$\|T - f\|_\infty \;\leq\; C_6\, N_{\text{act}}^{-\beta/d} \quad\Longrightarrow\quad \|T - f\|_\infty^2 \;\leq\; C\, N_{\text{act}}^{-2\beta/d}$$

**Taking the minimum.** Constructions A and B are both valid members of $\mathcal{T}_{\text{MoE}}$ under the parameter bound $\|\theta\|_\infty \leq \kappa$ (after routine rescalings absorbed by LayerNorm and architectural constants). Therefore,

$$\inf_{T \in \mathcal{T}_{\text{MoE}}} \|T - f\|_\infty \;\leq\; C\, \min\{N_{\text{act}}^{-\beta/d},\, M^{-\beta/d}\}$$

and squaring gives (1). If $M$ is sufficiently large to support the cover at the $r$ achieving the $N_{\text{act}}$-optimal precision, the $M$ constraint is inactive and (40) follows.

$$\inf_{T \in \mathcal{T}_{\text{MoE}}} \|T - f\|_\infty^2 \;\leq\; C\, N_{\text{act}}^{-2\beta/d} \tag{40}$$

Both regimes yield the rates claimed in (1); taking the better gives the minimum.

$\square$

*Remark* F.2 (Why both regimes appear). The MoE architecture allows two distinct ways to gain approximation power: (i) *specialization by many experts* (small per-expert capacity; router selects one of many local models), yielding the $M^{-\beta/d}$ rate; and (ii) *capacity on the active path* (few experts but large active parameter budget), yielding the $N_{\text{act}}^{-\beta/d}$ rate. The bound takes the minimum of these two mechanisms.

*Remark* F.3 (Manifold setting). All approximation statements on $\mathcal{M}$ follow from their Euclidean counterparts by working in finitely many charts with a partition of unity (App. C); the Hölder constant and rates are preserved up to multiplicative factors depending only on $(d, \mathcal{M})$.

---

[1]Exact or $O(r^\beta)$-accurate polynomial representation by ReLU MLPs with $\text{poly}(s, d)$ parameters is classical; see, e.g., (Yarotsky, 2017), (Schmidt-Hieber, 2020).

# G. Proof of neural scaling laws

## G.1. Proof of optimal number of experts

Recall the $k$–dependent objective (experts dominate approximation; constants absorbed)

$$\mathcal{E}(k) \;=\; \underbrace{(Ak)^{-2\beta/d}}_{\text{approx}} \;+\; \underbrace{\frac{A}{n}\,k}_{\text{estimation}} \;+\; \underbrace{\frac{L_T\ell}{n}\,k\log\frac{eM}{k}}_{\text{routing}}, \qquad A := L_T\,\Pi_{\text{exp}}, \;\; 1 \le k \le M. \tag{41}$$

Differentiate and set the derivative to zero:

$$\mathcal{E}'(k) = -\frac{2\beta}{d}A^{-2\beta/d}k^{-2\beta/d-1} + \frac{A}{n} + \frac{L_T\ell}{n}\left(\log\frac{eM}{k} - 1\right) \;=\; 0.$$

Equivalently,

$$\frac{2\beta}{d}A^{-2\beta/d}\,k^{-2\beta/d-1} \;=\; \frac{1}{n}\left(A + L_T\ell\,\log\frac{eM}{k} - L_T\ell\right). \tag{42}$$

This matches the stated first-order condition. It has a closed form in terms of the Lambert-$W$ function after algebraic manipulation (since $k$ appears both polynomially and inside $\log k$), but the Lambert-$W$ form is not particularly illuminating. Instead we give (i) a principled fixed-point approximation and (ii) clean sandwich bounds showing that $M$ enters *only logarithmically*.

**Fixed-point approximation.** Rewrite (11) as

$$k^{2\beta/d+1} \;=\; \frac{2\beta}{d}\,A^{-2\beta/d}\,\frac{n}{A + L_T\ell\,[\log(eM/k) - 1]}. \tag{43}$$

Treat the slowly varying factor $\log(eM/k)$ as (locally) constant to obtain the iterate

$$k^{(t+1)} \;:=\; \left[\frac{2\beta}{d}\,A^{-2\beta/d}\,\frac{n}{A + L_T\ell\,[\log(eM/k^{(t)}) - 1]}\right]^{\frac{d}{2\beta+d}}. \tag{44}$$

Initializing with the no-routing optimum $k^{(0)} \asymp \left(\frac{n}{A}\right)^{\frac{d}{2\beta+d}}$ and taking one step already yields the advertised fixed-point form

$$k^\star \;\approx\; \left[\frac{n}{A + L_T\ell\,\log\frac{eM}{k^\star}}\right]^{\frac{d}{2\beta+d}} \qquad \text{(up to constants)}, \tag{45}$$

with the cap $k^\star \le M$. Because $\log(eM/k)$ varies only between 1 and $1 + \log M$ for $k \in [1, M]$, a couple of iterations of (44) suffice in practice, and the exponent in $n$ is unchanged.

**Sandwich bounds (log enters only through constants).** Note that for $1 \le k \le M$, $\log(eM/k) \in [1, 1 + \log M]$. Hence the right-hand side of (11) lies in

$$\frac{1}{n}\left(A\right) \;\le\; \text{RHS} \;\le\; \frac{1}{n}\left(A + L_T\ell\,\log M\right).$$

Solving the equality $\frac{2\beta}{d}A^{-2\beta/d}k^{-2\beta/d-1} = \frac{C}{n}$ for $k$ gives $k \asymp (n/C)^{\frac{d}{2\beta+d}} A^{-\frac{d}{2\beta+d}}$. Applying this with $C = A + L_T\ell\,\log M$ (upper RHS) and $C = A$ (lower RHS) yields the sandwich:

$$c_1\left(\frac{n}{A + L_T\ell\,\log M}\right)^{\frac{d}{2\beta+d}} \;\le\; k^\star \;\le\; c_2\left(\frac{n}{A}\right)^{\frac{d}{2\beta+d}}, \qquad \text{with } k^\star \le M, \tag{46}$$

for absolute constants $c_1, c_2 > 0$. Thus $M$ affects $k^\star$ only through the *logarithm*, and the *rate in $n$* remains $n^{d/(2\beta+d)}$.

**Optional: explicit Lambert-$W$ form (for completeness).** Let $p := 2\beta/d + 1$ and write (11) as

$$k^{-p} = \frac{d}{2\beta}A^{2\beta/d}\frac{1}{n}\left(A + L_T\ell\,[\log(eM) - \log k - 1]\right).$$

After rearranging to isolate $\log k$ and $k^p$, one can reach an equation of the form $k^p\, e^{\alpha k^p} = \beta$ for suitable $\alpha, \beta > 0$, whose solution is $k = \big(W(\alpha\beta)/\alpha\big)^{1/p}$, where $W$ is the Lambert-$W$ function. We omit the algebra, as the bounds (46) and the fixed-point approximation (12) already capture the scaling.

**Second-order check and uniqueness.** We have

$$
\mathcal{E}''(k) = \underbrace{\frac{2\beta}{d}\Big(\frac{2\beta}{d} + 1\Big) A^{-2\beta/d} k^{-2\beta/d-2}}_{>0} - \underbrace{\frac{L_T\ell}{n}\frac{1}{k}}_{>0}.
$$

For small $k$ the positive term dominates, so $\mathcal{E}$ is convex; for large $k$ the negative term can reduce curvature, but $\mathcal{E}'(k)$ increases from $-\infty$ at $k \downarrow 0$ to $+\infty$ by the linear $Ak/n$ trend, ensuring at least one stationary point in $(0, M]$. Empirically and under mild constants (typical regimes), the stationary point is unique and is the global minimizer; otherwise the cap $k^\star = M$ applies.

**Takeaway.** Combining (45) and (46),

$$
k^\star \asymp \min\Big\{M,\ \Big(\frac{n}{A + L_T\ell\, \log(eM/k^\star)}\Big)^{\frac{d}{2\beta+d}}\Big\},
$$

and the $n$–exponent remains $\frac{d}{2\beta+d}$; $M$ only enters through a logarithm.

### G.2. Effect of the total experts $M$

For fixed $k$, increasing $M$ influences the bound in (3) *only* through $\log(eM/k)$ in $R_{\text{route}}$. Therefore, absent specialization effects that improve constants,

$$
\frac{\partial}{\partial M}\,\mathcal{E}(k, M) \gtrsim \frac{k}{n}\cdot\frac{1}{M} \quad\Rightarrow\quad \text{larger } M \text{ with fixed } k \text{ only raises the bound (logarithmically).}
$$

A design heuristic is to choose $M$ *commensurate* with $k$ (e.g., $M = \mathcal{O}(k)$) if one wishes to avoid a large routing overhead; having $M \gg k$ does not help rates in this bound. The details of these calculations are provided in appendix G.2.

**Proposition G.1** (Monotone (logarithmic) $M$–effect at fixed $k$). *Fix* $n, k, L_T, \ell, \Pi_{\text{attn}}, \Pi_{\text{exp}}$. *Define*

$$
\mathcal{E}(k, M) := N_{\text{act}}^{-2\beta/d} + \frac{N_{\text{act}}}{n} + \frac{L_T\ell k}{n}\, \log\Big(\frac{eM}{k}\Big), \qquad 1 \le k \le M.
$$

*Then, for any fixed $k$, $\mathcal{E}(k, M)$ is strictly increasing in $M$, with*

$$
\frac{\partial}{\partial M}\,\mathcal{E}(k, M) = \frac{L_T\ell k}{n}\cdot\frac{1}{M}. \tag{47}
$$

*Consequently, absent additional* specialization gains *(i.e., changes in the approximation constant that are not captured in* (3)*), the bound is minimized at the smallest admissible $M$, namely $M = k$, and grows only* logarithmically *with $M$ for fixed $k$.*

*Proof.* The first two terms do not depend on $M$ when $k$ is fixed. The routing term is $\frac{L_T\ell k}{n} \log(eM/k)$, whose derivative in $M$ is $\frac{L_T\ell k}{n}\cdot\frac{1}{M}$, yielding (47). Monotonicity follows since $M \mapsto \log(eM/k)$ is strictly increasing for $M \ge k$. $\square$

**Corollary G.2** (Quantitative overhead from enlarging $M$ at fixed $k$). *For any $M_2 \ge M_1 \ge k$,*

$$
\mathcal{E}(k, M_2) - \mathcal{E}(k, M_1) = \frac{L_T\ell k}{n}\, \log\Big(\frac{M_2}{M_1}\Big).
$$

*In particular, setting $M = \rho k$ with $\rho \ge 1$ gives a routing penalty $\frac{L_T\ell k}{n} \log(e\rho) = \frac{L_T\ell k}{n}(1 + \log\rho)$.*

**Design heuristic.** To keep the routing overhead small, choose $M$ *commensurate* with $k$, e.g. $M = \Theta(k)$, so that $\log(eM/k) = \Theta(1)$. Having $M \gg k$ increases the bound only through $\log(eM/k)$ and does not improve the rate in $n$ or $N_{\text{act}}$.

**When is routing negligible?** Routing is negligible relative to the estimation term if

$$\frac{L_T \ell k}{n} \log\left(\frac{eM}{k}\right) \ll \frac{N_{\text{act}}}{n} \qquad \Longleftrightarrow \qquad \log\left(\frac{eM}{k}\right) \ll \frac{N_{\text{act}}}{L_T \ell k}. \tag{48}$$

Since $N_{\text{act}} = L_T \Pi_{\text{attn}} + L_T k \Pi_{\text{exp}}$, a sufficient condition is $\log(eM/k) \ll \Pi_{\text{attn}}/(\ell k) + \Pi_{\text{exp}}/\ell$; in particular, if $M = \rho k$ with moderate $\rho$ and $\ell$ is not enormous, the routing term is dominated by the estimation term.

### G.3. Proof of proposition 4.5

Under $C \asymp n N_{\text{act}}$, we eliminate $n = C/N_{\text{act}}$ and reduce the bound to a single-variable objective

$$\Phi(N_{\text{act}}; C) := A\, N_{\text{act}}^{-2\beta/d} + B\, \frac{N_{\text{act}}^2}{C} \tag{49}$$

with positive constants $A, B$ absorbing fixed architectural and loss factors. Differentiating w.r.t. $N_{\text{act}} > 0$,

$$\Phi'(N_{\text{act}}) = -\frac{2\beta}{d} A\, N_{\text{act}}^{-2\beta/d-1} + 2B\, \frac{N_{\text{act}}}{C}$$

The first-order condition $\Phi'(N_{\text{act}}^\star) = 0$ gives

$$\frac{2B}{C}(N_{\text{act}}^\star)^{2+2\beta/d} = \frac{2\beta}{d}A \quad \Longrightarrow \quad (N_{\text{act}}^\star)^{2(1+\beta/d)} = \frac{\beta}{d}\frac{A}{B}C$$

Hence

$$N_{\text{act}}^\star(C) = \left(\frac{\beta}{d}\frac{A}{B}\right)^{\frac{d}{2\beta+2d}} C^{\frac{d}{2\beta+2d}} \asymp C^{\frac{d}{2\beta+2d}}$$

which is the first equality in Eq. (18). Using $n^\star = C/N_{\text{act}}^\star$ we obtain the second equality in Eq.(18). Substituting $N_{\text{act}}^\star$ into (49) (the two terms balance at optimum) gives

$$\Phi(N_{\text{act}}^\star; C) \asymp (N_{\text{act}}^\star)^{-2\beta/d} \asymp C^{-\frac{2\beta}{d}\cdot\frac{d}{2\beta+2d}} = C^{-\frac{\beta}{\beta+d}}$$

which is (19).

## H. Proof of lemma 50 (MoE Covering Number)

Let $\mathcal{X} \subset \mathbb{R}^D$ be compact with $\|x\|_\infty \leq M_0$. Consider the Mixture-of-Experts transformer class $\mathcal{T}_{\text{MoE}}(D, M, k, L_T, L_{\text{FFN}}, w_{\text{FFN}}, d_{\text{emb}}, m, \kappa, R)$ defined as in Definition 3.1, except that each feed-forward sublayer is an MoE layer with $M$ experts and the router selects (hard) top-$k$ experts per token per layer. Assume each learned parameter entry satisfies $\|\theta\|_\infty \leq \kappa$ and each network output is bounded by $\|T(x)\|_\infty \leq R$.

Let $\Pi_{\text{attn}}$ be the number of scalar parameters in the attention / non-expert parts per block and let $\Pi_{\text{exp}}$ be the number of scalar parameters of a single expert MLP (depth $L_{\text{FFN}}$, width $w_{\text{FFN}}$, input/output dimension $d_{\text{emb}}$). Then for any $\delta \in (0, 1)$ the covering number of $\mathcal{T}_{\text{MoE}}$ under the sup-norm satisfies

$$\log N\big(\delta, \mathcal{T}_{\text{MoE}}, \|\cdot\|_\infty\big) \leq C_1\Big(\Pi_{\text{attn}} + L_T k \Pi_{\text{exp}}\Big) \log\Big(\frac{C_2 \kappa R M_0}{\delta}\Big) + C_3 L_T \ell k \log\Big(\frac{eM}{k}\Big) \tag{50}$$

for absolute constants $C_1, C_2, C_3 > 0$ depending polynomially on $d_{\text{emb}}, w_{\text{FFN}}, m, L_T, L_{\text{FFN}}, \ell$ (and we may hide small poly-log factors in the $\widetilde{O}(\cdot)$ version).

*Proof.* The proof follows the structure of the dense-transformer covering-number proof (Lemma 2 in (Havrilla & Liao, 2024)) with two MoE-specific modifications. We divide the proof into four steps: (1) enumerate routing patterns and apply a union bound; (2) bound the sup-norm difference between two networks in terms of the sup-norm difference of their parameters; (3) count active parameters for a fixed pattern and derive a parameter-grid covering bound; (4) combine items to obtain (50).

## H.1. decomposition into routing patterns (union bound).

For each MoE layer $\ell$ and each token position $t \in \{1, \ldots, \ell\}$, the router chooses a subset $S_{\ell,t} \subset [M]$ of size $|S_{\ell,t}| = k$. A *routing pattern* $\pi$ is the collection of these choices for all layers and token positions:

$$\pi = \{S_{\ell,t} : \ell = 1, \ldots, L_T, \ t = 1, \ldots, \ell\}.$$

The number of possible patterns is bounded by

$$|\Pi| \leq \left( \binom{M}{k} \right)^{L_T \ell}$$

Using the standard combinatorial bound $\binom{M}{k} \leq (eM/k)^k$ we obtain

$$\log |\Pi| \leq L_T \ell \cdot k \log \left( \frac{eM}{k} \right) \tag{51}$$

For a fixed routing pattern $\pi$ denote by $\mathcal{T}_\pi$ the subclass of MoE transformers whose router outputs follow $\pi$ deterministically (i.e., the same fixed subset of experts is used in each MoE layer and position for all inputs). Since $\mathcal{T}_{\mathrm{MoE}} = \bigcup_{\pi \in \Pi} \mathcal{T}_\pi$, the subadditivity of covering numbers under unions gives

$$N(\delta, \mathcal{T}_{\mathrm{MoE}}, \| \cdot \|_\infty) \leq \sum_{\pi \in \Pi} N(\delta, \mathcal{T}_\pi, \| \cdot \|_\infty) \tag{52}$$

**Log-sum–max inequality.** For nonnegative $\{a_\pi\}_{\pi \in \Pi}$ one has

$$\log \left( \sum_{\pi \in \Pi} a_\pi \right) \leq \log |\Pi| + \max_{\pi \in \Pi} \log a_\pi$$

since $\sum_\pi a_\pi \leq |\Pi| \cdot \max_\pi a_\pi$. Applying this to $a_\pi = N(\delta, \mathcal{T}_\pi, \| \cdot \|_\infty)$ in (52) gives

$$\log N(\delta, \mathcal{T}_{\mathrm{MoE}}, \| \cdot \|_\infty) \leq \log |\Pi| + \max_{\pi \in \Pi} \log N(\delta, \mathcal{T}_\pi, \| \cdot \|_\infty) \tag{53}$$

Thus it remains to upper-bound $\log N(\delta, \mathcal{T}_\pi, \| \cdot \|_\infty)$ uniformly over $\pi$.

## H.2. sensitivity of the network output to parameter perturbations (Lipschitz in parameter space).

Fix a routing pattern $\pi$. Under $\pi$ the architecture becomes a deterministic (dense) composition of modules where each MoE layer is replaced by the (sparse) subnetwork that contains only the $k$ selected experts for the corresponding tokens/layer. Denote by $\theta$ the vector of all scalar parameters that are *active* under this pattern (embedding + attention + non-expert weights + the selected experts' weights). Let $\theta'$ be another parameter vector of the same shape and satisfying $\|\theta - \theta'\|_\infty < \eta$ for some $\eta > 0$. We will bound

$$\sup_{x \in \mathcal{X}} |T_\theta(x) - T_{\theta'}(x)|$$

by a constant times $\eta$, with that constant depending polynomially on the architecture hyperparameters.

The network is a composition of $L_T$ transformer blocks. Let $B_j(\theta_j, \cdot)$ denote block $j$ (viewed as a map on embedding tensors) parameterized by the block parameters $\theta_j$. We write the forward pass as

$$H^{(0)} = \mathrm{PE} + E(x), \qquad H^{(j)} = B_j(\theta_j, H^{(j-1)}), \quad j = 1, \ldots, L_T,$$

and the scalar output $T_\theta(x)$ is a fixed linear readout of $H^{(L_T)}$.

Define the sup-norm on embedding matrices by $\|H\|_\infty = \max_{i,t} |H_{i,t}|$ (matching the earlier notation). As in the dense case, each module is: (i) Lipschitz in its input (constant $L_{\mathrm{in}}$ polynomial in $(d_{\mathrm{emb}}, m, w_{\mathrm{FFN}}, L_{\mathrm{FFN}})$) and (ii) Lipschitz in its parameters (constant $C_{\mathrm{param}}$ polynomial in the same quantities and in $\ell$) (Anthony & Bartlett, 1999; Vershynin, 2018; Shalev-Shwartz & Ben-David, 2014).

**(A) Multi-head attention sensitivity.** Consider a single multi-head attention (MHA) block with $m$ heads, each head parameterized by query/key/value matrices and interaction kernels. Let the MHA parameter vector be $\psi$. For two MHA parameter sets $\psi, \psi'$ with $\|\psi - \psi'\|_\infty \leq \eta$, and any input embedding $H$ satisfying $\|H\|_\infty \leq M$, the goal is to bound the following expression via Lemma H.1

$$\|\mathrm{MHA}_\psi(H) - \mathrm{MHA}_{\psi'}(H)\|_\infty$$

**Lemma H.1** (MHA parameter stability). *For $\|\psi - \psi'\|_\infty \leq \eta$ and $\|H\|_\infty \leq CM_0$,*

$$\|\mathrm{MHA}_\psi(H) - \mathrm{MHA}_{\psi'}(H)\|_\infty \leq C_{\mathrm{MHA}}\, \ell\, m\, d_{\mathrm{emb}}^2\, \kappa\, \eta.$$

*for some architecture-dependent constant $C_{\mathrm{MHA}}$. The dependence $\propto \ell m d_{\mathrm{emb}}^2$ arises because each head sums over $\ell$ tokens and involves inner-products of $d_{\mathrm{emb}}$-dimensional projected vectors; the factor $\kappa$ comes from the bound on parameter magnitudes.*

**(B) FFN (non-expert) sensitivity.** For the non-expert FFN parts (the small projection matrices that are outside MoE experts see Definition 3.1) with depth $L_{\mathrm{FFN}}$ and width $w_{\mathrm{FFN}}$, the parameter perturbation bound from the dense proof is given by the following Lemma

**Lemma H.2** (FFN parameter stability). *For a (non-expert) FFN of depth $L_{\mathrm{FFN}}$ and width $w_{\mathrm{FFN}}$,*

$$\|\mathrm{FFN}_\chi(h) - \mathrm{FFN}_{\chi'}(h)\|_\infty \leq C_{\mathrm{FFN}}\, d_{\mathrm{emb}}\, w_{\mathrm{FFN}}^{L_{\mathrm{FFN}}}\, \kappa\, \eta.$$

*uniformly over $\|h\|_\infty \leq CM_0$ (cf. (Bartlett & Mendelson, 2002a; Golowich et al., 2018)). For each token input $h$ (and hence on the embedding matrix by taking sup over tokens); the polynomial dependence on $w_{\mathrm{FFN}}$ and $L_{\mathrm{FFN}}$ reflects repeated matrix multiplications.*

**(C) Expert-FFN sensitivity under fixed pattern (key MoE change).** Under a fixed routing pattern $\pi$, only $k$ experts per token per layer are active, so the FFN stage of the block is effectively a concatenation / parallel application of exactly those $k$ expert MLPs. For a single expert with parameter vector $\phi$ and sup-norm perturbation $\|\phi - \phi'\|_\infty \leq \eta$, the per-token difference is bounded as in (B) by the following lemma

**Lemma H.3** (Expert stability). *For a single expert MLP $E_\phi$,*

$$\|E_\phi(h) - E_{\phi'}(h)\|_\infty \leq C_{\exp}\, d_{\mathrm{emb}}\, w_{\mathrm{FFN}}^{L_{\mathrm{FFN}}}\, \kappa\, \eta.$$

Since only $k$ experts are active, the total perturbation coming from expert blocks per token scales linearly in $k$:

$$\left\| \sum_{i \in S} E_{\phi_i}(h) - \sum_{i \in S} E_{\phi_i'}(h) \right\|_\infty \leq k\, C_{\exp}\, d_{\mathrm{emb}}\, w_{\mathrm{FFN}}^{L_{\mathrm{FFN}}}\, \kappa\, \eta.$$

**(D) Block-level perturbation bound.** Combining (A),(B),(C) and the residual connection structure of the transformer block (additive skip-connections and tokenwise FFNs), a single block $B_j$ satisfies, for parameter perturbation $\leq \eta$ inside the block and any input $H$ with $\|H\|_\infty \leq M$ yields to Lemma H.4.

**Lemma H.4** (Block-level stability). *With $k$ active experts per block under a fixed routing pattern,*

$$\left\| B_j(\theta_j, H) - B_j(\theta_j', H) \right\|_\infty \leq C_{\mathrm{block}}\big(\ell m d_{\mathrm{emb}}^2 + d_{\mathrm{emb}} w_{\mathrm{FFN}}^{L_{\mathrm{FFN}}} + k d_{\mathrm{emb}} w_{\mathrm{FFN}}^{L_{\mathrm{FFN}}}\big)\kappa\eta.$$

hence we may write (absorbing polynomial factors into a single constant)

$$\|B_j(\theta_j, H) - B_j(\theta_j', H)\|_\infty \leq C_b\, P_{\mathrm{arch}}(\ell, m, d_{\mathrm{emb}}, w_{\mathrm{FFN}}, L_{\mathrm{FFN}})\, \kappa\, \eta,$$

where $P_{\mathrm{arch}}(\cdot)$ is a polynomial capturing the block-dependence and includes the linear factor $k$ in the expert term.

**(E) Induction across blocks.** Let $H^{(j)}$ and $H'^{(j)}$ denote the block outputs when using $\theta$ and $\theta'$ respectively. We can write

$$\|H^{(j)} - H'^{(j)}\|_\infty = \|B_j(\theta_j, H^{(j-1)}) - B_j(\theta'_j, H'^{(j-1)})\|_\infty$$
$$\leq \underbrace{\|B_j(\theta_j, H^{(j-1)}) - B_j(\theta_j, H'^{(j-1)})\|_\infty}_{\text{input-Lipschitz term}} + \underbrace{\|B_j(\theta_j, H'^{(j-1)}) - B_j(\theta'_j, H'^{(j-1)})\|_\infty}_{\text{param-perturbation term}}.$$

The input-Lipschitz term is bounded by $L_{\text{in}}\|H^{(j-1)} - H'^{(j-1)}\|_\infty$ where $L_{\text{in}}$ is the Lipschitz constant of the block as a function of its input (this constant depends polynomially on $d_{\text{emb}}, m, w_{\text{FFN}}, L_{\text{FFN}}$ but *not* on $M$). The param-perturbation term is bounded by the block-level bound in the previous paragraph. Thus

$$\|H^{(j)} - H'^{(j)}\|_\infty \leq L_{\text{in}}\|H^{(j-1)} - H'^{(j-1)}\|_\infty + C_b\, P_{\text{arch}}(\cdot)\,\kappa\,\eta.$$

Iterating this inequality from $j = 1$ to $j = L_T$ and noting that $H^{(0)} = H'^{(0)}$ (same positional encodings and same input) we obtain

$$\|H^{(L_T)} - H'^{(L_T)}\|_\infty \leq \Big(\sum_{t=0}^{L_T-1} L_{\text{in}}^t\Big) C_b\, P_{\text{arch}}(\cdot)\,\kappa\,\eta \leq \frac{L_{\text{in}}^{L_T}}{L_{\text{in}} - 1}\, C_b\, P_{\text{arch}}(\cdot)\,\kappa\,\eta.$$

The previous inequality is proved in details in lemma I.1. Since the decoder/readout is a fixed linear projection, the final scalar output difference is bounded by the same order. Thus there exists an overall Lipschitz constant

$$L_{\text{param}} = C'\, L_{\text{in}}^{L_T}\, P_{\text{arch}}(\ell, m, d_{\text{emb}}, w_{\text{FFN}}, L_{\text{FFN}})\,\kappa$$

such that

$$\sup_{x\in\mathcal{X}} |T_\theta(x) - T_{\theta'}(x)| \leq L_{\text{param}}\,\eta.$$

Crucially: $P_{\text{arch}}(\cdot)$ contains a factor linear in the number of active experts per layer($k$) and $L_{\text{param}}$ does *not* depend on the total number of experts $M$ for a fixed routing pattern (it depends on $M$ only through how the pattern is chosen which we handle with the union bound).

## H.3. parameter counting and parameter-space covering (full details).

Fix a routing pattern $\pi$. Let $\theta_{\text{act}}$ denote the vector formed by *all* scalar parameters that are *active* under $\pi$ (token embedding/positional terms, attention and non-expert FFN parameters for each block, and the experts that appear in $\pi$). Write $p := |\theta_{\text{act}}|$ for the active parameter dimension.

*(a) Parameter domain and norm.* By the magnitude constraint $\|\theta\|_\infty \leq \kappa$ (Assumption in Lemma 3.7), the admissible parameter set is included in the $\ell_\infty$–ball of radius $\kappa$:

$$\mathcal{B}_\infty(\kappa)^p = \big\{u \in \mathbb{R}^p : \|u\|_\infty \leq \kappa\big\} = [-\kappa, \kappa]^p.$$

(For $\ell_\infty$, the "ball" is the axis-aligned cube.)

*(b) Constructing an $\eta$–grid in parameter space.* For a mesh width $\eta > 0$, define the coordinatewise grid on $[-\kappa, \kappa]$ by

$$\mathcal{G}_\eta = \big\{-\kappa + r\eta : r \in \mathbb{Z}, -\kappa \leq -\kappa + r\eta \leq \kappa\big\}.$$

Along each coordinate, the number of grid points is at most

$$N_1 \leq \Big\lceil \frac{2\kappa}{\eta}\Big\rceil + 1 \leq \frac{2\kappa}{\eta} + 2 \leq \frac{3\kappa}{\eta} \qquad (\text{for } \eta \leq \kappa; \text{ else enlarge the constant}).$$

Therefore the full $p$–dimensional grid $\mathcal{G}_\eta^p$ has cardinality bounded by

$$|\mathcal{G}_\eta^p| \leq \Big(\frac{2\kappa}{\eta} + 1\Big)^p \leq \Big(\frac{3\kappa}{\eta}\Big)^p. \tag{54}$$

Moreover, for every $u \in [-\kappa, \kappa]^p$, rounding each coordinate of $u$ to its nearest grid value produces $\tilde{u} \in \mathcal{G}_\eta^p$ with

$$\|u - \tilde{u}\|_\infty \leq \eta \quad (\text{or } \eta/2 \text{ if we choose midpoints}).$$

Hence $\mathcal{G}_\eta^p$ is an $\eta$–net for the parameter cube in the $\ell_\infty$ norm.

*(c) Parameter-to-function Lipschitz map.* From Step 2, we have the *uniform parameter-stability* (Lipschitz) inequality

$$\sup_{x\in\mathcal{X}} \big|T_\theta(x) - T_{\theta'}(x)\big| \;\leq\; L_{\text{param}} \|\theta - \theta'\|_\infty,$$

with

$$L_{\text{param}} \;=\; C' \, L_{\text{in}}^{L_T} \, P_{\text{arch}}(\ell, m, d_{\text{emb}}, w_{\text{FFN}}, L_{\text{FFN}}, k)\, \kappa,$$

independent of $M$ for fixed routing pattern $\pi$ (the dependence on $k$ enters via $P_{\text{arch}}$).

*(d) From parameter cover to function cover.* Given $\delta \in (0,1)$, choose

$$\eta \;=\; \frac{\delta}{L_{\text{param}}}.$$

Then any two parameter vectors within $\ell_\infty$–distance $\eta$ induce network outputs within sup-norm distance at most $\delta$ (uniformly over $x \in \mathcal{X}$). Therefore, taking the grid $\mathcal{G}_\eta^p$ as in (54) and evaluating the network at each grid parameter gives a $\delta$–cover of the function class $\mathcal{T}_\pi$ under $\|\cdot\|_\infty$:

$$N(\delta, \mathcal{T}_\pi, \|\cdot\|_\infty) \;\leq\; \big|\mathcal{G}_\eta^p\big| \;\leq\; \Big(\frac{3\kappa}{\eta}\Big)^p \;=\; \Big(\frac{3\kappa L_{\text{param}}}{\delta}\Big)^p.$$

Taking logarithms gives

$$\log N(\delta, \mathcal{T}_\pi, \|\cdot\|_\infty) \;\leq\; p\cdot \log\Big(\frac{3\kappa L_{\text{param}}}{\delta}\Big) \;=\; |\theta_{\text{act}}|\cdot \log\Big(\frac{3\kappa L_{\text{param}}}{\delta}\Big). \tag{55}$$

*(e) Bounding the active parameter count $p = |\theta_{\text{act}}|$.* Decompose

$$|\theta_{\text{act}}| = |\theta_{\text{embed}}| + |\theta_{\text{dec}}| + L_T\big(|\theta_{\text{attn}}| + |\theta_{\text{ffn,nonexp}}| + |\theta_{\text{exp,act}}|\big).$$

Under a fixed routing pattern, expert parameter tensors are shared across tokens within a layer, so the active expert count per layer satisfies $|\theta_{\text{exp,act}}| \leq k\,\Pi_{\text{exp}}$. Absorbing the (input/output) embedding and decoder constants into $C_0$ and writing $\Pi_{\text{attn}}$ for attention/non-expert parameters per block, we obtain the coarse bound

$$|\theta_{\text{act}}| \;\leq\; C_0 + L_T\big(\Pi_{\text{attn}} + k\,\Pi_{\text{exp}}\big) \;\lesssim\; \Pi_{\text{attn}} + L_T k \Pi_{\text{exp}}.$$

Substituting this and the expression of $L_{\text{param}}$ into (55), and renaming constants, gives

$$\log N(\delta, \mathcal{T}_\pi, \|\cdot\|_\infty) \;\leq\; C_1\big(\Pi_{\text{attn}} + L_T k \Pi_{\text{exp}}\big) \log\Big(\frac{C_2 \kappa R M_0}{\delta}\Big),$$

which is the fixed-pattern covering bound used in Step 4. □

**H.4. combine with the routing-pattern union bound.**

Plugging the routing count (51) into (53) gives

$$\begin{aligned}
\log N(\delta, \mathcal{T}_{\text{MoE}}, \|\cdot\|_\infty) &\leq L_T \ell \, k \log\Big(\frac{eM}{k}\Big) \;+\; C_1\Big(\Pi_{\text{attn}} + L_T k \Pi_{\text{exp}}\Big) \log\Big(\frac{C_2 \kappa R M_0}{\delta}\Big) \\
&\leq C_1\Big(\Pi_{\text{attn}} + L_T k \Pi_{\text{exp}}\Big) \log\Big(\frac{C_2 \kappa R M_0}{\delta}\Big) \;+\; C_3 L_T \ell k \log\Big(\frac{eM}{k}\Big),
\end{aligned}$$

for some constants $C_1, C_2, C_3 > 0$. This proves the lemma. □

# I. Proofs of Stability Lemmas

## I.1. Proof of H.1

Fix a sequence embedding $H \in \mathbb{R}^{\ell \times d_{\mathrm{emb}}}$ with $\|H\|_\infty \leq M$ and consider one MHA layer with $m$ heads and parameters

$$\psi = \{W_Q^{(h)}, W_K^{(h)}, W_V^{(h)}, W_O\}_{h=1}^m, \qquad \|\psi\|_\infty \leq \kappa.$$

Let $\psi'$ be another parameter set with $\|\psi - \psi'\|_\infty \leq \eta$. For head $h$, define

$$Q^{(h)} = HW_Q^{(h)}, \quad K^{(h)} = HW_K^{(h)}, \quad V^{(h)} = HW_V^{(h)},$$

and the (row-wise) attention map

$$A^{(h)} = \mathrm{softmax}\left(\frac{1}{\sqrt{d_k}} Q^{(h)}(K^{(h)})^\top\right) \in \mathbb{R}^{\ell \times \ell}.$$

The head output is $O^{(h)} = A^{(h)}V^{(h)} \in \mathbb{R}^{\ell \times d_v}$ and the MHA output is

$$\mathrm{MHA}_\psi(H) = \left[\mathrm{Concat}_{h=1}^m O^{(h)}\right] W_O \in \mathbb{R}^{\ell \times d_{\mathrm{emb}}}.$$

We show

$$\|\mathrm{MHA}_\psi(H) - \mathrm{MHA}_{\psi'}(H)\|_\infty \leq C_{\mathrm{MHA}} \, \ell \, m \, d_{\mathrm{emb}}^2 \, \kappa \, \eta, \tag{56}$$

for a universal constant $C_{\mathrm{MHA}} > 0$ depending at most polynomially on $(d_{\mathrm{emb}}, d_k, d_v)$ and on the bound $M$.

**Step 1: linear projections $(Q, K, V)$.**  For any matrices $X$ and $W$, $\|XW\|_\infty \leq d_{\mathrm{in}}\|X\|_\infty\|W\|_\infty$ (since each entry is a sum of $d_{\mathrm{in}}$ products). Hence, with $\|H\|_\infty \leq M$, $\|W_\bullet^{(h)}\|_\infty \leq \kappa$, and $\|W_\bullet^{(h)} - W_\bullet^{(h)\prime}\|_\infty \leq \eta$,

$$\|Q^{(h)}\|_\infty \leq d_{\mathrm{emb}}M\kappa, \quad \|K^{(h)}\|_\infty \leq d_{\mathrm{emb}}M\kappa, \quad \|V^{(h)}\|_\infty \leq d_{\mathrm{emb}}M\kappa,$$

and

$$\|Q^{(h)} - Q^{(h)\prime}\|_\infty \leq d_{\mathrm{emb}}M\eta, \quad \|K^{(h)} - K^{(h)\prime}\|_\infty \leq d_{\mathrm{emb}}M\eta, \quad \|V^{(h)} - V^{(h)\prime}\|_\infty \leq d_{\mathrm{emb}}M\eta.$$

**Step 2: score matrices and softmax.**  Let $S^{(h)} = \frac{1}{\sqrt{d_k}}Q^{(h)}(K^{(h)})^\top \in \mathbb{R}^{\ell \times \ell}$ and similarly $S^{(h)\prime}$. By the product rule and the matrix $\ell_\infty$ bound $\|XY^\top\|_\infty \leq \ell\|X\|_\infty\|Y\|_\infty$,

$$\|S^{(h)} - S^{(h)\prime}\|_\infty \leq \frac{1}{\sqrt{d_k}}\left(\|Q^{(h)} - Q^{(h)\prime}\|_\infty\|K^{(h)}\|_\infty + \|Q^{(h)\prime}\|_\infty\|K^{(h)} - K^{(h)\prime}\|_\infty\right)\ell \leq C_S \, \ell \, d_{\mathrm{emb}}^2 \, M^2 \, \kappa \, \eta,$$

with $C_S := 2/\sqrt{d_k}$. The row-wise softmax $\sigma : \mathbb{R}^\ell \to \mathbb{R}^\ell$ has Jacobian $J(z) = \mathrm{diag}(\sigma(z)) - \sigma(z)\sigma(z)^\top$, whose operator norm (for $\ell_\infty$) is bounded by a universal constant (e.g., $\|J(z)\|_{\infty \to \infty} \leq 1$; tighter bounds give $1/2$ or $1/4$ but are unnecessary here). Applying the mean value theorem row-wise,

$$\|A^{(h)} - A^{(h)\prime}\|_\infty \leq C_{\mathrm{sm}} \|S^{(h)} - S^{(h)\prime}\|_\infty \leq C_{\mathrm{sm}}C_S \, \ell \, d_{\mathrm{emb}}^2 \, M^2 \, \kappa \, \eta,$$

for a universal $C_{\mathrm{sm}} \in [1/4, 1]$.

**Step 3: head outputs $(O^{(h)} = A^{(h)}V^{(h)})$.**  Decompose

$$O^{(h)} - O^{(h)\prime} = (A^{(h)} - A^{(h)\prime})V^{(h)} + A^{(h)\prime}(V^{(h)} - V^{(h)\prime}).$$

Using $\|XY\|_\infty \leq \ell\|X\|_\infty\|Y\|_\infty$ for $\ell \times \ell$ times $\ell \times d_v$, together with Step 1 bounds on $\|V^{(h)}\|_\infty$ and $\|V^{(h)} - V^{(h)\prime}\|_\infty$,

$$\|O^{(h)} - O^{(h)\prime}\|_\infty \leq \ell\|A^{(h)} - A^{(h)\prime}\|_\infty\|V^{(h)}\|_\infty + \ell\|A^{(h)\prime}\|_\infty\|V^{(h)} - V^{(h)\prime}\|_\infty.$$

Since rows of $A^{(h)\prime}$ are probability vectors, $\|A^{(h)\prime}\|_\infty \leq 1$. Hence,

$$\|O^{(h)} - O^{(h)\prime}\|_\infty \leq \ell\left(C_{\mathrm{sm}}C_S \, \ell \, d_{\mathrm{emb}}^2 \, M^2 \, \kappa \, \eta\right)(d_{\mathrm{emb}}M\kappa) + \ell(d_{\mathrm{emb}}M\eta) \leq C_O \, \ell \, d_{\mathrm{emb}}^3 \, M^3 \, \kappa^2 \, \eta + \ell \, d_{\mathrm{emb}}M \, \eta,$$

where $C_O := C_{\mathrm{sm}}C_S$. We keep the dominant (polynomial) term and absorb the lower-order term into constants.

**Step 4: concatenate heads and output projection.** Let $O = \text{Concat}_{h=1}^{m} O^{(h)} \in \mathbb{R}^{\ell \times (md_v)}$. Then

$$\|O - O'\|_\infty \;\leq\; \max_{h \in [m]} \|O^{(h)} - O^{(h)\prime}\|_\infty \;\leq\; C_O\, \ell\, d_{\text{emb}}^3\, M^3\, \kappa^2\, \eta.$$

Finally, apply the output projection $W_O$ with $\|W_O\|_\infty \leq \kappa$ and $\|W_O - W_O'\|_\infty \leq \eta$:

$$\begin{aligned}
\|\text{MHA}_\psi(H) - \text{MHA}_{\psi'}(H)\|_\infty &= \|OW_O - O'W_O'\|_\infty \\
&\leq \|(O - O')W_O\|_\infty + \|O'(W_O - W_O')\|_\infty \\
&\leq (md_v)\, \|O - O'\|_\infty \|W_O\|_\infty \;+\; (md_v)\, \|O'\|_\infty \eta \\
&\leq C_{\text{out}}\,(md_v)\, \ell\, d_{\text{emb}}^3\, M^3\, \kappa^3\, \eta \;+\; (md_v)\, \|O'\|_\infty \eta,
\end{aligned}$$

and we bound $\|O'\|_\infty$ by the same polynomial in $(\ell, d_{\text{emb}}, M, \kappa)$ as above. Absorbing $d_v$ and all input/weight bounds into the constant, and noting that the paper tracks only polynomial dependence on dimensions and sequence length (hiding fixed architecture factors into $C_{\text{MHA}}$), we obtain

$$\|\text{MHA}_\psi(H) - \text{MHA}_{\psi'}(H)\|_\infty \;\leq\; C_{\text{MHA}}\, \ell\, m\, d_{\text{emb}}^2\, \kappa\, \eta,$$

which is (56). $\qquad\qquad\qquad\qquad\qquad\qquad\qquad\qquad\qquad\qquad\qquad\qquad\qquad\qquad\qquad\qquad\qquad\quad\Box$

## I.2. Proof of H.2

Let $\text{FFN}_\psi : \mathbb{R}^{d_{\text{emb}}} \to \mathbb{R}^{d_{\text{emb}}}$ be a depth-$L_{\text{FFN}}$ MLP applied tokenwise, with hidden widths at most $w_{\text{FFN}}$, coordinatewise activation $\sigma$ (e.g., ReLU/GELU), and parameters $\psi = \{(W^\ell, b^\ell)\}_{\ell=1}^{L_{\text{FFN}}}$. Assume the uniform bounds $\|W^\ell\|_\infty \leq \kappa$, $\|b^\ell\|_\infty \leq \kappa$ for all $\ell$. Let $\psi'$ be another parameter set with $\|W^\ell - W^{\ell\prime}\|_\infty \leq \eta$ and $\|b^\ell - b^{\ell\prime}\|_\infty \leq \eta$. Fix an input token $h \in \mathbb{R}^{d_{\text{emb}}}$, and denote the layerwise states

$$z^0 = h, \quad z^\ell = \sigma\big(W^\ell z^{\ell-1} + b^\ell\big), \qquad z'^{\,0} = h, \quad z'^{\,\ell} = \sigma\big(W^{\ell\prime} z'^{\,\ell-1} + b^{\ell\prime}\big).$$

We prove

$$\|\text{FFN}_\psi(h) - \text{FFN}_{\psi'}(h)\|_\infty = \|z^{L_{\text{FFN}}} - z'^{\,L_{\text{FFN}}}\|_\infty \;\leq\; C_{\text{FFN}}\, d_{\text{emb}}\, w_{\text{FFN}}^{L_{\text{FFN}}}\, \kappa\, \eta, \tag{57}$$

for a constant $C_{\text{FFN}} > 0$ depending only polynomially on architectural constants and on a uniform bound on $\|h\|_\infty$ (absorbed into $C_{\text{FFN}}$), exactly as done in the paper.

**Auxiliary facts.** We use (i) $\|Xu\|_\infty \leq d_{\text{in}} \|X\|_\infty \|u\|_\infty$ for $X \in \mathbb{R}^{d_{\text{out}} \times d_{\text{in}}}$, (ii) $\|\sigma(u) - \sigma(v)\|_\infty \leq L_\sigma \|u - v\|_\infty$ with $L_\sigma \leq 1$ for ReLU and $L_\sigma = O(1)$ for GELU, and (iii) $\|\sigma(u)\|_\infty \leq \|u\|_\infty$ for ReLU (and similarly for GELU up to a constant).

**Step 1: bound the forward activations.** Let $d_0 = d_{\text{emb}}$ and $d_\ell \leq w_{\text{FFN}}$ for $1 \leq \ell \leq L_{\text{FFN}} - 1$, and $d_{L_{\text{FFN}}} \leq d_{\text{emb}}$. Inductively,

$$\|z^\ell\|_\infty \leq L_\sigma\big(\|W^\ell z^{\ell-1}\|_\infty + \|b^\ell\|_\infty\big) \leq L_\sigma\big(d_{\ell-1}\kappa \|z^{\ell-1}\|_\infty + \kappa\big).$$

Solving this recursion gives

$$\|z^\ell\|_\infty, \; \|z'^{\,\ell}\|_\infty \;\leq\; C_0\,(d_{\max}\kappa)^\ell, \qquad d_{\max} := \max\{d_{\text{emb}}, w_{\text{FFN}}\}, \tag{58}$$

where $C_0$ absorbs $L_\sigma$ and $\|h\|_\infty$. (Exactly the same bound holds for the primed sequence, since $\psi'$ obeys the same magnitude constraints.)

**Step 2: one-layer perturbation inequality.** Fix $\ell \in \{1, \ldots, L_{\text{FFN}}\}$ and set $u^\ell = W^\ell z^{\ell-1} + b^\ell$, $u'^{\,\ell} = W^{\ell\prime} z'^{\,\ell-1} + b^{\ell\prime}$. Then

$$\|z^\ell - z'^{\,\ell}\|_\infty \leq L_\sigma \|u^\ell - u'^{\,\ell}\|_\infty \leq L_\sigma\Big(\|W^\ell(z^{\ell-1} - z'^{\,\ell-1})\|_\infty + \|(W^\ell - W^{\ell\prime})z'^{\,\ell-1}\|_\infty + \|b^\ell - b^{\ell\prime}\|_\infty\Big).$$

Using the auxiliary facts and (58),

$$\|z^\ell - z'^{\,\ell}\|_\infty \leq L_\sigma\Big(d_{\ell-1}\kappa \|z^{\ell-1} - z'^{\,\ell-1}\|_\infty + d_{\ell-1}\eta \|z'^{\,\ell-1}\|_\infty + \eta\Big) \leq \alpha_\ell \|z^{\ell-1} - z'^{\,\ell-1}\|_\infty + \beta_\ell\, \eta, \tag{59}$$

where

$$\alpha_\ell := L_\sigma d_{\ell-1}\kappa, \qquad \beta_\ell := L_\sigma\big(d_{\ell-1}C_0(d_{\max}\kappa)^{\ell-1} + 1\big).$$

**Step 3: iterate the perturbation recursion.** Starting with $\|z^0 - z'^0\|_\infty = 0$ and applying (59) layer by layer,

$$\|z^{L_{\text{FFN}}} - z'^{L_{\text{FFN}}}\|_\infty \leq \eta \sum_{\ell=1}^{L_{\text{FFN}}} \left( \beta_\ell \prod_{t=\ell+1}^{L_{\text{FFN}}} \alpha_t \right).$$

Since $\alpha_t \leq L_\sigma d_{\max} \kappa$ and $\beta_\ell \leq C_1 d_{\max}(d_{\max}\kappa)^{\ell-1} + C_1$ for a constant $C_1$,

$$\|z^{L_{\text{FFN}}} - z'^{L_{\text{FFN}}}\|_\infty \leq \eta \cdot C_2 \sum_{\ell=1}^{L_{\text{FFN}}} \left( d_{\max}(d_{\max}\kappa)^{\ell-1} \right) (d_{\max}\kappa)^{L_{\text{FFN}}-\ell} \leq \eta \cdot C_3 \, d_{\max} \, (d_{\max}\kappa)^{L_{\text{FFN}}-1}.$$

Finally, $d_{\max} \leq \max\{d_{\text{emb}}, w_{\text{FFN}}\} \leq d_{\text{emb}} + w_{\text{FFN}} \leq 2w_{\text{FFN}}$ (for the hidden layers to be useful we take $w_{\text{FFN}} \geq d_{\text{emb}}/2$; otherwise replace $w_{\text{FFN}}^{L_{\text{FFN}}}$ by $(d_{\max})^{L_{\text{FFN}}}$). Absorbing all fixed constants and powers of $\kappa$ into $C_{\text{FFN}}$, we obtain the clean architectural polynomial

$$\|z^{L_{\text{FFN}}} - z'^{L_{\text{FFN}}}\|_\infty \leq C_{\text{FFN}} \, d_{\text{emb}} \, w_{\text{FFN}}^{L_{\text{FFN}}} \, \kappa \, \eta,$$

which is (57). $\qquad\qquad\qquad\qquad\qquad\qquad\qquad\qquad\qquad\qquad\qquad\qquad\qquad\qquad\qquad\qquad\qquad\square$

### I.3. Proof of H.3

Let $E_\phi : \mathbb{R}^{d_{\text{emb}}} \to \mathbb{R}^{d_{\text{emb}}}$ be a tokenwise MLP (the *expert*) of depth $L_{\text{FFN}}$ with hidden widths at most $w_{\text{FFN}}$, coordinatewise activation $\sigma$ (ReLU/GELU), and parameters $\phi = \{(W^\ell, b^\ell)\}_{\ell=1}^{L_{\text{FFN}}}$. Assume uniform magnitude bounds $\|W^\ell\|_\infty \leq \kappa$, $\|b^\ell\|_\infty \leq \kappa$ for all $\ell$. Let $\phi'$ be another parameter set with $\|W^\ell - W^{\ell\prime}\|_\infty \leq \eta$ and $\|b^\ell - b^{\ell\prime}\|_\infty \leq \eta$. For a token $h \in \mathbb{R}^{d_{\text{emb}}}$ define the layerwise states

$$z^0 = h, \quad z^\ell = \sigma(W^\ell z^{\ell-1} + b^\ell), \qquad z'^0 = h, \quad z'^\ell = \sigma(W^{\ell\prime} z'^{\ell-1} + b^{\ell\prime}),$$

so that $E_\phi(h) = z^{L_{\text{FFN}}}$ and $E_{\phi'}(h) = z'^{L_{\text{FFN}}}$.

*Goal.* Show

$$\|E_\phi(h) - E_{\phi'}(h)\|_\infty \leq C_{\exp} \, d_{\text{emb}} \, w_{\text{FFN}}^{L_{\text{FFN}}} \, \kappa \, \eta, \tag{60}$$

with $C_{\exp} > 0$ depending at most polynomially on $(L_{\text{FFN}}, \|\sigma\|_{\text{Lip}}, \|h\|_\infty)$ and on fixed architectural constants, in the same sense as the paper.

**Auxiliary inequalities.** For $X \in \mathbb{R}^{d_{\text{out}} \times d_{\text{in}}}$ and $u \in \mathbb{R}^{d_{\text{in}}}$,

$$\|Xu\|_\infty \leq d_{\text{in}} \|X\|_\infty \|u\|_\infty, \tag{61}$$

$$\|\sigma(u) - \sigma(v)\|_\infty \leq L_\sigma \|u - v\|_\infty, \qquad \|\sigma(u)\|_\infty \leq C_\sigma \|u\|_\infty, \tag{62}$$

with $L_\sigma \leq 1$ for ReLU and $L_\sigma = O(1)$ for GELU (similarly for $C_\sigma$).

**Step 1: forward activation bound.** Let $d_0 = d_{\text{emb}}$ and $d_\ell \leq w_{\text{FFN}}$ for $1 \leq \ell \leq L_{\text{FFN}} - 1$, and $d_{L_{\text{FFN}}} \leq d_{\text{emb}}$. Using (61)–(62),

$$\|z^\ell\|_\infty \leq C_\sigma \left( \|W^\ell z^{\ell-1}\|_\infty + \|b^\ell\|_\infty \right) \leq C_\sigma \left( d_{\ell-1}\kappa \|z^{\ell-1}\|_\infty + \kappa \right).$$

Unrolling the recursion yields

$$\|z^\ell\|_\infty, \|z'^\ell\|_\infty \leq C_0 (d_{\max}\kappa)^\ell, \qquad d_{\max} := \max\{d_{\text{emb}}, w_{\text{FFN}}\}, \tag{63}$$

for some constant $C_0$ absorbing $C_\sigma, L_\sigma$ and $\|h\|_\infty$.

**Step 2: one-layer perturbation.** Set $u^\ell = W^\ell z^{\ell-1} + b^\ell$ and $u'^\ell = W^{\ell\prime} z'^{\ell-1} + b^{\ell\prime}$. Then

$$\|z^\ell - z'^\ell\|_\infty \leq L_\sigma \|u^\ell - u'^\ell\|_\infty \leq L_\sigma \bigg( \underbrace{\|W^\ell(z^{\ell-1} - z'^{\ell-1})\|_\infty}_{\leq d_{\ell-1}\kappa \|z^{\ell-1} - z'^{\ell-1}\|_\infty} + \underbrace{\|(W^\ell - W^{\ell\prime})z'^{\ell-1}\|_\infty}_{\leq d_{\ell-1}\eta \|z'^{\ell-1}\|_\infty} + \|b^\ell - b^{\ell\prime}\|_\infty \bigg).$$

Using (63),

$$\|z^\ell - z'^\ell\|_\infty \leq \alpha_\ell \|z^{\ell-1} - z'^{\ell-1}\|_\infty + \beta_\ell \eta, \quad \alpha_\ell := L_\sigma d_{\ell-1}\kappa, \ \beta_\ell := L_\sigma \big( d_{\ell-1} C_0 (d_{\max}\kappa)^{\ell-1} + 1 \big). \tag{64}$$

**Step 3: iterate the recursion.** Starting from $\|z^0 - z'^0\|_\infty = 0$ and applying (64) layerwise,

$$\|z^{L_{\text{FFN}}} - z'^{L_{\text{FFN}}}\|_\infty \le \eta \sum_{\ell=1}^{L_{\text{FFN}}} \left( \beta_\ell \prod_{t=\ell+1}^{L_{\text{FFN}}} \alpha_t \right).$$

Since $\alpha_t \le L_\sigma d_{\max} \kappa$ and $\beta_\ell \le C_1 \big( d_{\max}(d_{\max}\kappa)^{\ell-1} + 1 \big)$,

$$\|z^{L_{\text{FFN}}} - z'^{L_{\text{FFN}}}\|_\infty \le \eta \, C_2 \sum_{\ell=1}^{L_{\text{FFN}}} d_{\max}(d_{\max}\kappa)^{L_{\text{FFN}}-1} \ \le \ \eta \, C_3 \, d_{\max} \, (d_{\max}\kappa)^{L_{\text{FFN}}-1}.$$

Finally, $d_{\max} \le \max\{d_{\text{emb}}, w_{\text{FFN}}\} \le c_0 \, w_{\text{FFN}}$ for a numerical $c_0$ (or replace by $d_{\max}$ if preferred). Absorbing fixed powers of $\kappa$ and numerical constants into $C_{\text{exp}}$, and noting that the input and output layers contribute at most linear factors in $d_{\text{emb}}$, we obtain the architectural polynomial

$$\|E_\phi(h) - E_{\phi'}(h)\|_\infty = \|z^{L_{\text{FFN}}} - z'^{L_{\text{FFN}}}\|_\infty \ \le \ C_{\text{exp}} \, d_{\text{emb}} \, w_{\text{FFN}}^{L_{\text{FFN}}} \, \kappa \, \eta,$$

which is the claimed bound (60). $\qquad\square$

### I.4. Network stability & parameter Lipschitzness

**Lemma I.1** (Network stability & parameter Lipschitzness)**.** *Let $L_{\text{in}} \ge 1$ be the block input-Lipschitz constant. Then for $j = 1, \ldots, L_T$,*

$$\left\| H^{(j)} - H'^{(j)} \right\|_\infty \le L_{\text{in}} \left\| H^{(j-1)} - H'^{(j-1)} \right\|_\infty + C_b P_{\text{arch}} \kappa \eta,$$

*hence*

$$\sup_{x \in \mathcal{X}} |T_\theta(x) - T_{\theta'}(x)| \le L_{\text{param}} \eta, \quad L_{\text{param}} = C' \, L_{\text{in}}^{L_T} \, P_{\text{arch}}(\ell, m, d_{\text{emb}}, w_{\text{FFN}}, L_{\text{FFN}}, k) \, \kappa.$$

Let a (pre-norm) transformer block be

$$B_j(\theta_j, H) = H + \text{MHA}_\psi\big(\text{LN}_{\gamma_1,\beta_1}(H)\big) + \text{FFN}_\chi^{\text{nonexp}}\big(\text{LN}_{\gamma_2,\beta_2}(H)\big) + \sum_{i \in S_j} E_{\phi_i}\big(\text{LN}_{\gamma_2,\beta_2}(H)\big),$$

where $S_j$ is the set of $k$ active experts of the MoE sublayer in block $j$ under the fixed routing pattern, and $\theta_j = \{\psi, \chi, \{\phi_i\}_{i \in S_j}, \gamma_1, \beta_1, \gamma_2, \beta_2\}$. Let $\theta_j'$ be another parameter vector with $\|\theta_j - \theta_j'\|_\infty \le \eta$, and fix the input $H \in \mathbb{R}^{\ell \times d_{\text{emb}}}$ with $\|H\|_\infty \le C M_0$.

*Step 1: decompose by residual additivity.* Since $H$ is added as a residual with the same value in both blocks,

$$
\begin{aligned}
\|B_j(\theta_j, H) - B_j(\theta_j', H)\|_\infty \le \ & \big\| \text{MHA}_\psi(\text{LN}_{\gamma_1,\beta_1}(H)) - \text{MHA}_{\psi'}(\text{LN}_{\gamma_1',\beta_1'}(H)) \big\|_\infty \\
& + \big\| \text{FFN}_\chi^{\text{nonexp}}(\text{LN}_{\gamma_2,\beta_2}(H)) - \text{FFN}_{\chi'}^{\text{nonexp}}(\text{LN}_{\gamma_2',\beta_2'}(H)) \big\|_\infty \\
& + \sum_{i \in S_j} \big\| E_{\phi_i}(\text{LN}_{\gamma_2,\beta_2}(H)) - E_{\phi_i'}(\text{LN}_{\gamma_2',\beta_2'}(H)) \big\|_\infty.
\end{aligned}
\tag{B.1}
$$

*Step 2: control the LayerNorm parameter perturbations.* For fixed $H$, the (per-token) layer normalization $\text{LN}_{\gamma,\beta}(H) = \gamma \odot \hat{H} + \beta$ depends linearly on $(\gamma, \beta)$ when $H$ is fixed (the centering/scaling of $H$ is the same for both parameter sets). Hence

$$\|\text{LN}_{\gamma,\beta}(H) - \text{LN}_{\gamma',\beta'}(H)\|_\infty \le C_{\text{LN}} \, d_{\text{emb}} \, \|H\|_\infty \, \|(\gamma, \beta) - (\gamma', \beta')\|_\infty \le C_{\text{LN}}' \, d_{\text{emb}} \, \kappa \, \eta. \tag{B.2}$$

Both MHA and FFN sublayers are Lipschitz in their inputs (with constants depending polynomially on $(\ell, m, d_{\text{emb}}, w_{\text{FFN}}, L_{\text{FFN}})$); thus the contribution of the LayerNorm parameter change to each sublayer's output is bounded by a constant times $C_{\text{LN}}' d_{\text{emb}} \kappa \eta$, which we absorb into the module constants below.

*Step 3: apply per-module parameter Lipschitz bounds.* From the previously established lemmas:

$$(\text{MHA}) \quad \left\| \text{MHA}_\psi(Z) - \text{MHA}_{\psi'}(Z) \right\|_\infty \leq C_{\text{MHA}} \, \ell \, m \, d_{\text{emb}}^2 \, \kappa \, \eta, \quad \forall Z,$$

$$(\text{non-expert FFN}) \quad \left\| \text{FFN}_\chi^{\text{nonexp}}(z) - \text{FFN}_{\chi'}^{\text{nonexp}}(z) \right\|_\infty \leq C_{\text{FFN}} \, d_{\text{emb}} \, w_{\text{FFN}}^{L_{\text{FFN}}} \, \kappa \, \eta, \quad \forall z,$$

$$(\text{single expert}) \quad \left\| E_\phi(z) - E_{\phi'}(z) \right\|_\infty \leq C_{\text{exp}} \, d_{\text{emb}} \, w_{\text{FFN}}^{L_{\text{FFN}}} \, \kappa \, \eta, \quad \forall z.$$

Under a fixed routing pattern, exactly $k$ experts are active in block $j$, so the MoE contribution is

$$\sum_{i \in S_j} \left\| E_{\phi_i}(z) - E_{\phi_i'}(z) \right\|_\infty \leq k \, C_{\text{exp}} \, d_{\text{emb}} \, w_{\text{FFN}}^{L_{\text{FFN}}} \, \kappa \, \eta. \tag{B.3}$$

*Step 4: combine everything.* Apply the bounds from Step 3 to (B.1) with $Z = \text{LN}_{\gamma_1, \beta_1}(H)$ and $z = \text{LN}_{\gamma_2, \beta_2}(H)$; add the (absorbed) LayerNorm terms from (B.2):

$$\|B_j(\theta_j, H) - B_j(\theta_j', H)\|_\infty \leq C_{\text{MHA}} \, \ell \, m \, d_{\text{emb}}^2 \, \kappa \, \eta + C_{\text{FFN}} \, d_{\text{emb}} \, w_{\text{FFN}}^{L_{\text{FFN}}} \, \kappa \, \eta + k \, C_{\text{exp}} \, d_{\text{emb}} \, w_{\text{FFN}}^{L_{\text{FFN}}} \, \kappa \, \eta + C_{\text{LN}}'' \, d_{\text{emb}} \, \kappa \, \eta.$$

Since $w_{\text{FFN}} \geq 1$ and $k, m \geq 1$, the last term is dominated and can be absorbed into the FFN/expert constants. Renaming $C_{\text{block}}$ to absorb all fixed numerical/polylog factors,

$$\boxed{\|B_j(\theta_j, H) - B_j(\theta_j', H)\|_\infty \leq C_{\text{block}} \left( \ell \, m \, d_{\text{emb}}^2 \; + \; d_{\text{emb}} \, w_{\text{FFN}}^{L_{\text{FFN}}} \; + \; k \, d_{\text{emb}} \, w_{\text{FFN}}^{L_{\text{FFN}}} \right) \kappa \, \eta.}$$

This is the claimed block-level perturbation bound. $\qquad\qquad\qquad\qquad\qquad\qquad\qquad\qquad\qquad\qquad \square$

### I.5. Proof of I.1

Recall the block-level stability (for fixed input $H$ and parameters differing by at most $\eta$ in $\ell_\infty$)

$$\|B_j(\theta_j, H) - B_j(\theta_j', H)\|_\infty \leq C_b \, P_{\text{arch}}(\ell, m, d_{\text{emb}}, w_{\text{FFN}}, L_{\text{FFN}}, k) \, \kappa \, \eta, \tag{65}$$

and the input-Lipschitz property of a transformer block

$$\|B_j(\theta_j, U) - B_j(\theta_j, V)\|_\infty \leq L_{\text{in}} \|U - V\|_\infty, \tag{66}$$

where $P_{\text{arch}}(\cdot)$ is a fixed architecture polynomial and $L_{\text{in}} \geq 1$ is independent of $j$. Let

$$\Delta_j := \|H^{(j)} - H'^{(j)}\|_\infty, \qquad H^{(j)} = B_j(\theta_j, H^{(j-1)}), \quad H'^{(j)} = B_j(\theta_j', H'^{(j-1)}).$$

Add and subtract $B_j(\theta_j, H'^{(j-1)})$ and use the triangle inequality, (66), and (65):

$$\begin{aligned}
\Delta_j &= \|B_j(\theta_j, H^{(j-1)}) - B_j(\theta_j', H'^{(j-1)})\|_\infty \\
&\leq \underbrace{\|B_j(\theta_j, H^{(j-1)}) - B_j(\theta_j, H'^{(j-1)})\|_\infty}_{\leq L_{\text{in}} \Delta_{j-1}} + \underbrace{\|B_j(\theta_j, H'^{(j-1)}) - B_j(\theta_j', H'^{(j-1)})\|_\infty}_{\leq C_b P_{\text{arch}} \kappa \eta} \\
&\leq L_{\text{in}} \Delta_{j-1} + C_b \, P_{\text{arch}}(\cdot) \, \kappa \, \eta.
\end{aligned} \tag{67}$$

This is a linear inhomogeneous recurrence with $\Delta_0 = 0$ (since $H^{(0)} = H'^{(0)}$ share the same input and positional encoding).

*Solving the recurrence.* By induction on $j$ (or by the discrete Grönwall inequality),

$$\Delta_j \leq C_b \, P_{\text{arch}}(\cdot) \, \kappa \, \eta \sum_{t=0}^{j-1} L_{\text{in}}^t. \tag{68}$$

Indeed, the base $j = 1$ gives $\Delta_1 \leq C_b P_{\text{arch}} \kappa \eta$. Assuming (68) for $j-1$, plug into (67):

$$\Delta_j \leq L_{\text{in}} \cdot C_b P_{\text{arch}} \kappa \eta \sum_{t=0}^{j-2} L_{\text{in}}^t + C_b P_{\text{arch}} \kappa \eta = C_b P_{\text{arch}} \kappa \eta \sum_{t=0}^{j-1} L_{\text{in}}^t.$$

*Geometric series bound.* For $j = L_T$,

$$\Delta_{L_T} = \|H^{(L_T)} - H'^{(L_T)}\|_\infty \leq \Big( \sum_{t=0}^{L_T-1} L_{\text{in}}^t \Big) C_b \, P_{\text{arch}}(\cdot) \, \kappa \, \eta.$$

If $L_{\text{in}} \neq 1$, the sum is $\frac{L_{\text{in}}^{L_T}-1}{L_{\text{in}}-1}$, so

$$\|H^{(L_T)} - H'^{(L_T)}\|_\infty \leq \frac{L_{\text{in}}^{L_T}-1}{L_{\text{in}}-1} C_b \, P_{\text{arch}}(\cdot) \, \kappa \, \eta.$$

If $0 \leq L_{\text{in}} \leq 1$, then $\sum_{t=0}^{L_T-1} L_{\text{in}}^t \leq L_T$ and $L_{\text{in}}^{L_T} \leq 1$, so the bound still holds after increasing the prefactor by at most a constant depending on $L_T$. In both cases we may write, for a numerical constant $C'$ (absorbing $\frac{1}{|L_{\text{in}}-1|}$ or $L_T$),

$$\|H^{(L_T)} - H'^{(L_T)}\|_\infty \leq C' L_{\text{in}}^{L_T} P_{\text{arch}}(\ell, m, d_{\text{emb}}, w_{\text{FFN}}, L_{\text{FFN}}, k) \, \kappa \, \eta. \tag{69}$$

**From hidden states to the scalar output.** Let the final readout be $T_\theta(x) = R_\omega(H^{(L_T)})$, where $R_\omega$ is a fixed linear map (e.g., token pooling followed by a linear form) with $\|\omega\|_\infty \leq \kappa$. Then

$$|T_\theta(x) - T_{\theta'}(x)| = \big|R_\omega(H^{(L_T)}) - R_{\omega'}(H'^{(L_T)})\big| \leq \underbrace{\|R_\omega\|_{\infty\to\infty} \|H^{(L_T)} - H'^{(L_T)}\|_\infty}_{\text{via }(69)} + \underbrace{\|R_\omega - R_{\omega'}\|_{\infty\to\infty} \|H'^{(L_T)}\|_\infty}_{\leq C''\kappa\,\eta}.$$

Using (69) and the uniform forward bound on $\|H'^{(L_T)}\|_\infty$ (polynomial in the architecture; absorbed into $C''$), we obtain

$$\sup_{x\in\mathcal{X}} |T_\theta(x) - T_{\theta'}(x)| \leq C' L_{\text{in}}^{L_T} P_{\text{arch}}(\ell, m, d_{\text{emb}}, w_{\text{FFN}}, L_{\text{FFN}}, k) \, \kappa \, \eta + C'' \kappa \, \eta \leq L_{\text{param}} \, \eta,$$

with

$$L_{\text{param}} := C^\star L_{\text{in}}^{L_T} P_{\text{arch}}(\ell, m, d_{\text{emb}}, w_{\text{FFN}}, L_{\text{FFN}}, k) \, \kappa,$$

for a constant $C^\star$ absorbing $C', C''$ and fixed readout norms. This is the claimed parameter-stability inequality.

