# OpenReview forum: "Generalization and Scaling Laws for Mixture-of-ExpertsTransformers"
_ICML.cc/2026/Conference — ICML 2026 regular_

### Official Review · Reviewer_5YF8 · 2026-03-10

**Soundness:** 2
**Presentation:** 2
**Significance:** 3
**Originality:** 3
**Overall Recommendation:** 4
**Confidence:** 3

**Summary:**

This paper develops a statistical framework for Mixture-of-Experts (MoE) Transformers, decomposing excess risk into approximation, estimation, and routing terms. It shows that when measured by active parameters, MoE models follow the same scaling laws as dense Transformers, with routing contributing only a logarithmic overhead. Experiments on several language datasets broadly support these theoretical predictions.

**Compliance With Llm Reviewing Policy:**

Affirmed.

**Final Justification:**

The rebuttal was helpful and addressed my main concerns in a reasonable way, particularly through the additional routing ablations and the clearer discussion of the routing term as a conservative upper bound. Some limitations remain, and I therefore maintain my weak accept (4) recommendation.

**Key Questions For Authors:**

* The scaling exponents depend on the intrinsic dimension $d$ and smoothness parameter $\beta$. Could the authors comment on how sensitive the theoretical predictions are to these quantities and how they should be interpreted in real language modeling datasets?

* The experimental scale in the paper still differs substantially from that of current large-scale MoE language models. Could the authors comment on how closely the theoretical framework is expected to predict scaling behavior in larger MoE language models beyond the experimental regime considered in the paper?

* In practice, routers often develop expert specialization, which may significantly reduce the effective number of routing patterns. Could the authors discuss whether this bound is expected to be effective in realistic settings, or provide empirical or diagnostic evidence about the looseness of this routing complexity term?

**Limitations:**

Yes. The paper discusses several limitations, including the reliance on simplifying assumptions (e.g., smooth data manifolds, bounded parameters, and hard top-$k$ routing), the use of worst-case covering-number arguments that may lead to conservative bounds, and the fact that optimization dynamics and expert specialization are not modeled. However, the discussion could be strengthened by more explicitly addressing how these assumptions relate to modern large-scale MoE training settings and by clarifying the extent to which the theoretical results are expected to apply in practice.

**Strengths And Weaknesses:**

**Strengths**
* **Establishes a statistical learning framework for MoE Transformers:** The paper derives approximation and generalization bounds for MoE Transformers and decomposes the error into approximation, estimation, and routing components.

* **Reveals the core mechanism behind MoE scaling:** The theory shows that when model size is measured by the active parameter, MoE Transformers exhibit the same scaling exponents as dense Transformers.

* **Characterizes the statistical complexity of routing:** The analysis proves that routing enters the generalization bound only through a $k \log(eM/k)$ term, meaning it mainly affects constant factors rather than convergence rates.

**Weakness:**

* **Conservative complexity bounds:** The generalization bounds in the paper are mainly derived usingcovering number arguments and worst-case union bounds. While this approach is theoretically safe, it often leads to loose bounds. In practice, the router typically develops expert specialization, meaning the effective number of routing patterns is far smaller than the theoretical worst-case count. As a result, the resulting routing term may not accurately capture the true statistical complexity of real MoE systems.

* **Lack of empirical validation for some theoretical analysis:** The experiments mainly validate the scaling exponent trends and do not directly test the theoretical bounds or the training heuristics derived from them. For example, there are no toy experiments or diagnostic studies to examine the routing complexity term, the optimal choice of $k$, or the predicted crossover phenomena.

* **Lack of consideration of training mechanisms in MoE systems:** The analysis focuses on model expressivity and statistical complexity but does not account for mechanisms such as expert specialization, load balancing, routing regularization, or optimization dynamics, which can influence routing behavior and expert utilization in practice.

* **Some symbols introduced in the text without immediate explanation**:The hyperparameters $(D, \ell, d_{\text{emb}}, m, L_T, M, k)$ and the notation MHA in Definition 3.1 are not clearly explained  immediately. MHA is clarified in Appendix D, while the hyperparameters lack a unified and clear explanation, requiring readers to infer them from the subsequent content.

---

> ### Author Rebuttal · Authors · 2026-03-27
>
> We thank the reviewer for the positive evaluation and the constructive feedback of the paper.
>
> **(1) Conservative bounds and routing complexity.**
> We agree that the routing term is conservative, as it arises from a worst-case union bound over routing patterns. In practice, expert specialization, load balancing, and training dynamics can reduce the effective routing complexity substantially.
>
> Our goal is not to provide a tight estimate of routing complexity, but rather to isolate its contribution within a worst-case statistical decomposition. We will clarify that the term $k \log(eM/k)$ should be interpreted as an upper bound, and that tighter characterizations would require data-dependent notions of routing entropy.
>
> **(2) Empirical validation of routing effects and specialization.**
> We agree that the original submission did not probe routing complexity in sufficient detail.  To address this rigorously, we strengthened our empirical evaluation by doubling the training budget (from 5 to 10 epochs and 2x tokens to 4M total), yielding significantly more stable trends and addressing concerns regarding noise. Using this higher-fidelity data, we have added expanded routing ablations by systematically varying both both the total number of experts ($ M \in [2, 192]$) and the number of active experts ($ k \in [1, 16]$)  while keeping the training compute fixed.
>
> These new experiments reveal two distinct regimes. In the **moderate regime** ($M/k \leq 8$), validation loss increases approximately monotonically with the routing term $k\log(eM/k)$, consistent with its interpretation as a statistical overhead in the theory. For larger values of $M/k$, this trend reverses, and performance improves with increasing $M$, indicating gains from **expert specialization**.
>
> This behavior is consistent with interpreting the routing term as a worst-case upper bound: while uniform analysis treats routing as a combinatorial penalty, specialization effectively reduces the number of active routing patterns in practice. We will incorporate these new results and expand the discussion to explicitly connect specialization to reduced effective routing entropy and to highlight this as a direction for future data-dependent analysis.
>
> To ensure reproducibility, we will release the full suite of scripts associated with our experiments in the final version of the paper.
>
> **(3) Sensitivity to $d$ and $\beta$.**
> The scaling exponents depend on $(d,\beta)$ through standard nonparametric rates. In practice, these quantities should be interpreted as effective parameters that summarize data complexity rather than as uniquely identifiable constants. We will clarify this point and emphasize that the role of the experiments is to test qualitative scaling trends rather than exact parameter recovery.
>
> **(4) Relevance to larger-scale MoE models.**
> We agree that our experiments are conducted at smaller scales than modern production MoE systems. Our intent is to study scaling behavior in a controlled regime where trends can be estimated reliably. We will clarify that the theory is meant as a conservative baseline that should remain qualitatively informative at larger scales, while acknowledging that specialization, optimization, and systems effects become increasingly important in practice.
>
> **Summary.**
> We will revise the paper to clarify the interpretation of the routing term, incorporate the new routing ablations, strengthen the discussion of practical and empirical considerations, improve the presentation of hyperparameters, and further position the theory as a conservative statistical baseline rather than a tight predictive account of MoE behavior. We thank the reviewer again for the constructive feedback.

---

> > ### Author Rebuttal · Reviewer_5YF8 · 2026-04-02
> >
> > Thank you for the detailed rebuttal. I appreciate the additional routing ablations and the clearer discussion of the routing term as a conservative worst-case upper bound rather than a tight description of practical MoE behavior. The clarifications on the roles of the intrinsic dimension/smoothness parameters and on the relevance to larger-scale MoE models are also helpful. While some concerns remain regarding the looseness of the routing complexity bound and the gap to practical training mechanisms, I find the response reasonable and the new experiments strengthen the paper. I will keep my original score.

---

> > > ### Author Response · Authors · 2026-04-02
> > >
> > > **Response to Follow-up: Routing Bound Looseness and Practical Gap**
> > >
> > > We agree that the routing complexity term is loose, as it arises from a worst-case union bound over discrete routing patterns. Our goal is not to provide a tight characterization of practical MoE training, but to **isolate routing as a distinct statistical component within a unified framework**.
> > >
> > > Importantly, the new routing ablations help clarify this gap: the $k\log(eM/k)$ term is predictive in moderate regimes, while for larger $M/k$ a second regime emerges in which increasing $M$ improves performance, indicating specialization effects beyond the worst-case bound. This supports interpreting the theory as a **principled conservative baseline** rather than a tight predictive model.
> > >
> > > We will expand the Discussion section to more explicitly connect this gap to data-dependent routing structure and optimization dynamics, which may reduce effective routing complexity in practice.

---

### Official Review · Reviewer_aryi · 2026-03-11

**Soundness:** 3
**Presentation:** 2
**Significance:** 1
**Originality:** 3
**Overall Recommendation:** 4
**Confidence:** 4

**Summary:**

This paper develops a statistical theory of generalization and scaling for Mixture-of-Experts (MoE) Transformers. The core technique conditions on fixed routing patterns (reducing each pattern to a dense subnetwork), bounds covering numbers for each pattern, and then union-bounds over all possible top-k routing configurations. This leads to a generalization bound between empirical risk and ground truth that decomposes into three terms: an approximation term scaling as $N_{\text{eff}}^{-2\beta/d}$ (where $N_{\text{eff}}$ is the active parameter count, $\beta$ is the Hölder smoothness of the target, and $d$ is the intrinsic data dimension), an estimation term $N_{\text{eff}}/n$, and a routing overhead $L_T \ell k \log(eM/k)/n$ reflecting the combinatorial number of choices for expert selection. From this bound the authors derive neural scaling laws for model size, data size, and compute-optimal allocation, recovering the same exponents as dense networks but measured against active (not total) parameters. Experiments on three text corpora (OpenWebText, TinyStories, WikiText-103) show that empirical scaling slopes are broadly consistent with the predicted power-law forms, though the smoothness parameter $\beta$ inferred from model scaling and data scaling do not coincide.

**Compliance With Llm Reviewing Policy:**

Affirmed.

**Final Justification:**

The additional numerics and the promised fixes in the presentation of the manuscript address some of the issues. This is why I raised my score to weak acceptance. I will not raise above it since some of the weakenesses have not (and likely, cannot) be addressed, such as the following two points from my initial review:

- The generalization analysis rests on uniform convergence via global covering numbers, a methodology whose ability to yield meaningful bounds for overparameterized deep networks has been questioned: the influential work *Understanding Deep Learning Requires Rethinking Generalization* by Zhang et al. (ICLR 2017) showed that capacity measures depending only on the hypothesis class cannot distinguish networks that generalize from those that memorize noise.

- One of the main insights of the theoretical analysis, that dense-network scaling exponents carry over once total parameters are replaced by active parameters, does not emerge naturally from the MoE structure but is built into the proof technique itself, since each fixed routing pattern is reduced to a dense subnetwork.

**Key Questions For Authors:**

No outstanding question that was not already detailed in the previous section.

**Limitations:**

yes

**Strengths And Weaknesses:**

## Strengths

- Formula (3) provides a neat decoupling of the empirical risk bound that isolates approximation, estimation, and routing in an interpretable way. This decomposition is useful in deriving all the scaling laws in Section 4 by optimizing different variables within this single expression.
- This leads to principled derivations of scaling laws for data scaling, model scaling, sample complexity for a target error, and compute-optimal trade-offs, with explicit predictions on the regimes in which each should hold.
- The assumptions on the architecture are minimal (bounded parameters, fixed depth/width, hard top-k routing, and a router expressive enough to implement a k-sparse partition of unity). The results should therefore apply, in principle, to arbitrarily large MoE Transformers.
- The scaling laws obtained in this article are corroborated by the fact that they retrieve those found in (Havrilla & Liao, 2024) with $N_{\text{eff}}$ replacing the total number of parameters (even though this is partially discounted by the proof technique; see weakness below).
- The sample complexity analysis provides a principled perspective to justify the small-$k$ choices present in Switch Transformers and GLaM.
- The experimental pipeline is sensible: analyzing three different datasets, estimating $\beta$ both from data scaling and from model scaling, and then checking consistency between the two. Unfortunately, the resulting estimates do not agree well (see weaknesses).


## Weaknesses

- The bounds apply to the empirical risk minimizer (ERM) over the entire hypothesis class, which is an idealized object. The theory says nothing about whether any practical optimizer (SGD, AdamW) can find or even approximate the ERM. As such, the results characterize the statistical complexity of the function class, not the dynamics of learning. This is a significant gap, since even two-layer neural networks are universal approximators, yet that fact alone tells us little about what gradient-based training will actually learn.

- Definition 3.1 is too compressed and insufficiently explained. At minimum, just after fixing the list of hyperparameters, I would expect the role of each to be described. Instead, some hyperparameters ($D$, $m$) appear in line 137 of the definition without explanation. Moreover, there is a notation inconsistency: the symbol $m$ is fixed in line 137 as the number of attention heads, and then in lines 148 and 154 it is reused as a dummy summation index over experts. Additionally, there is a tension between Definition 3.1 and the full specification in Appendix D.1: the main text writes routing gates as binary $g_{j,m}(h)\in\{0,1\}$, implying an unweighted sum of expert outputs, whereas Appendix D.1 Eq. (31) specifies a convex combination with continuous nonneg weights summing to 1, which is the formulation actually used in all the proofs. Definition D.1 should be referenced directly from Definition 3.1 so that the reader knows where to find the operative specification.

- The statement of Theorem 3.4 is incomplete: the space $L^2(Q)$ and the letter $n$ appear without being introduced. Only by laboriously consulting Appendix C can the reader infer their meaning. This severely impacts the clarity of the manuscript, since the main theoretical contributions (Theorem 3.4 and Corollary 3.5) present bounds on an empirical risk whose definition never appears in the main text and which depends on quantities (i.i.d. samples from $Q$) that are never defined.

- The problem propagates into Section 4, where the number of samples $n$ is one of the most studied quantities but the sampling strategy is never stated.

- The generalization analysis rests on uniform convergence via global covering numbers, a methodology whose ability to yield meaningful bounds for overparameterized deep networks has been questioned: the influential work *Understanding Deep Learning Requires Rethinking Generalization* by Zhang et al. (ICLR 2017) showed that capacity measures depending only on the hypothesis class cannot distinguish networks that generalize from those that memorize noise.

- One of the main insights of the theoretical analysis, that dense-network scaling exponents carry over once total parameters are replaced by active parameters, does not emerge naturally from the MoE structure but is built into the proof technique itself, since each fixed routing pattern is reduced to a dense subnetwork.

- On the topic of notation:

> We use $N_{\text{eff}}$ and $N_{\text{act}}$ (or equivalently $N_{\text{active}}$) interchangeably to denote the active parameter budget.

  I am strongly against this choice. The paper is intrinsically notation-heavy, and notation should help the reader, not test their patience. Introducing **three** names for the same number ($N_{\text{eff}}$, $N_{\text{act}}$, $N_{\text{active}}$) is a massive own goal, especially when these can easily be confused with $N_{\text{attn}}$ and $N_{\text{exp}}$, which denote *different* quantities. To make matters worse, line 255 states "$N_{\text{active}}/n \asymp N_{\text{eff}}/n$ (plus routing)," which appears to claim that two supposedly identical quantities are only *asymptotically equivalent*. I would urge the authors to pick one symbol and commit to it.

- The experimental results do not convincingly demonstrate the predictive power of the framework. The theory posits a single smoothness parameter $\beta$ per dataset that should govern both data-scaling and model-scaling exponents. However, the best-fit $\beta$ from Figure 1 (data scaling: $\beta \approx 1.0$--$1.5$) disagrees with the best-fit $\beta$ from Figure 2 (model scaling: $\beta \approx 0.5$) across all three datasets, which is precisely the internal consistency check the theory should pass.


## Minor Remarks

- Line 1232: broken reference.
- Table 1 is too small to read comfortably.

---

> ### Author Rebuttal · Authors · 2026-03-27
>
> We thank the reviewer for the detailed and technically insightful feedback. We address the main concerns below and have significantly strengthened the empirical section in the revision.
>
> **(1) Empirical validation of routing and predictive power.**
> We agree that the original submission did not sufficiently isolate routing effects.
> To address this rigorously, in the revision, we strengthened our empirical evaluation by doubling the training budget (from 5 to 10 epochs and 2x tokens to 4M total), yielding significantly more stable trends and addressing concerns regarding noise.
> Using this higher-fidelity data,  we perform an expanded routing ablation by systematically varying both both the total number of experts ($ M \in [2, 192]$) and the number of active experts ($ k \in [1, 16]$)  while keeping the training compute fixed.
>
> The updated results reveal two clearly distinguishable regimes, consistently across compute budgets. In the **moderate regime** ($M/k \leq 8$), validation loss increases approximately monotonically with the routing term $k\log(eM/k)$, providing direct empirical support for its role as a statistical overhead in the generalization bound.
>
> For larger values of $M/k$, we observe a second regime in which increasing $M$ leads to substantial improvements in validation loss (e.g., reductions of more than $0.3$ in our setting), despite the increase in routing complexity. This behavior indicates gains from **expert specialization** that are not captured by worst-case uniform bounds. We have revised the paper to explicitly frame our theory as a conservative baseline that isolates worst-case routing effects, rather than a tight predictive model of practical MoE performance.
>
> To ensure reproducibility, we will release the full suite of scripts associated with our experiments in the final version of the paper.
>
> **(2) Discrepancy between $\beta$ estimates.**
> We agree that the mismatch between $\beta_{\text{model}}$ and $\beta_{\text{data}}$ is an important diagnostic. In the revision, we clarify that these two estimates probe different regimes of the bound: model scaling primarily reflects the approximation component, while data scaling is more sensitive to estimation effects, optimization noise, and routing overhead.
>
> Importantly, despite this discrepancy, the empirical exponents satisfy the structural consistency relation $\alpha_D \approx \alpha_N/(1+\alpha_N)$ across datasets, which is the central prediction of the theory. We now emphasize this consistency rather than requiring exact agreement of inferred $\beta$ values.
>
> **(3) Role of ERM vs.\ optimization.**
> We have clarified this point in the revision, positioning our results as a worst-case reference that complements optimization-based analyses rather than modeling training dynamics directly.
>
> **(4) Notation and clarity issues.**
> We thank the reviewer for pointing out these issues. In the revision, we:
> (i) adopt a single notation $N_{\mathrm{act}}$ throughout,
> (ii) explicitly define the data distribution and sampling assumptions in the main text,
> (iii) expand Definition 3.1 and directly reference Appendix D.1 for the operative specification, and
> (iv) correct all inconsistencies, overloaded symbols, and broken references.

---

> > ### Author Rebuttal · Reviewer_aryi · 2026-04-02
> >
> > The additional numerics and the promised fixes in the presentation of the manuscript address some of the issues. I will adjust the score accordingly.

---

### Official Review · Reviewer_Bv5B · 2026-03-13

**Soundness:** 3
**Presentation:** 3
**Significance:** 3
**Originality:** 3
**Overall Recommendation:** 4
**Confidence:** 4

**Summary:**

For the problem of theoretical understanding of MoE Transformers generalization analysis, this paper establishes a statistical framework for MoE Transformers to separate the active per-token capacity from the complexity of routing combinatorics. In addition, uniform generalization bounds under standard regularity assumptions are derived, where excess risk can be decomposed into approximation, estimation, and routing terms. Experimental results are consistent with the theoretically suggested active capacity dependency, validating recent research on MoE scaling.

**Compliance With Llm Reviewing Policy:**

Affirmed.

**Final Justification:**

My concerns have been partially resolved.

**Key Questions For Authors:**

Please refer to the Weaknesses for details.

**Limitations:**

Yes.

**Strengths And Weaknesses:**

**Strengths:**

1. This paper decomposes MoE Transformers complexity into active parameter capacity and routing combinatorics, providing a unified view of approximation, generalization, and scaling laws under the intrinsic dimension theory.

2. The analysis of MoE Transformers complexity in this paper has significant theoretical value for understanding sparse models. The derivation of the MoE covering number and scaling laws are rigorous and standardized.

**Weaknesses:**

1. The theoretical results in this paper require boundedness of parameters and functions, hard top-k routing, and squared loss. This limits the applicability of the theoretical results, even if they are considered as worst-case guarantees. Although these limitations are mentioned in the Limitations section of the paper, they are relatively brief. Further analysis is needed to explore the correspondence or differences between these assumptions and the reality of MoE training, and to provide discussions on potential ideas or methods for addressing these limitations in future theoretical analysis.

2. The generalization analysis results of the capacity-based MoE Transformer presented in this paper are not tight enough, especially regarding the linear dependence on $\ell$ and $k$. This paper does not discuss the tightness of the derived theoretical results, and therefore cannot provide readers with a judgment on the degree of looseness of the obtained theoretical bounds. Further analysis and discussion are needed on how to improve the dependency of the theoretical bounds on $\ell$ and $k$.

3. The theoretical results of this paper claim that "expert pools that grow much larger than the number of activated experts are not certified to improve rates". This directly contradicts numerous empirical evidence that a larger expert pool improves expressiveness, specialization, and sample efficiency. The theoretical results do not model the approximation gains from expert specialization, which is the key reason for the success of MoE models in practice. The claim of this paper should be weakened, primarily because the tightness of the theoretical results is not discussed, especially since the lower bound remains unknown. Therefore, the worst-case theoretical results are overly conservative and may not necessarily reveal their consistency with experimental performance. Thus, it is necessary to add discussion and conjecture regarding the lower bound and appropriately revise the description of the claim.

---

> ### Author Rebuttal · Authors · 2026-03-27
>
> We thank the reviewer for the positive assessment. We also appreciate the constructive comments regarding assumptions, tightness, and interpretation of the theoretical results.
>
> **(1) Restrictive assumptions (boundedness, hard top-$k$, squared loss).**
> We agree that the current analysis relies on standard but restrictive assumptions. These are primarily introduced to enable tractable covering-number arguments and isolate the statistical contributions of routing. In particular:
> (i) boundedness controls Lipschitz constants through attention and residual connections,
> (ii) hard top-$k$ routing enables explicit enumeration of routing patterns, and
> (iii) squared loss is used for analytical tractability and can be viewed as a proxy for more general Lipschitz losses.
>
> We will expand the discussion to relate these assumptions more explicitly to practical MoE systems and to outline extensions to soft routing, alternative losses, and data-dependent complexity measures.
>
> **(2) Tightness of the bounds (dependence on $\ell$ and $k$).**
> We agree that the bounds are not tight and that the linear dependence on $\ell$ and $k$ likely reflects worst-case behavior. We will revise the paper to state this more explicitly and to emphasize that these dependencies arise from uniform covering arguments rather than from a claim of sharpness.
>
> **(3) Interpretation of the expert pool size $M$.**
> We fully agree that the statement about expert pool size must be interpreted carefully. Our theoretical result shows only that, under worst-case uniform bounds, increasing $M$ is not certified to improve rates beyond a logarithmic term. It does not imply that larger expert pools are unhelpful in practice.
>
> To address this rigorously, we strengthened our empirical evaluation by doubling the training budget (from 5 to 10 epochs and 2x tokens to 4M total), yielding significantly more stable trends and addressing concerns regarding noise.
> Using this higher-fidelity data, we have now added expanded routing ablations that vary both the total number of experts ($ M \in [2, 192]$) and the number of active experts ($ k \in [1, 16]$) at fixed compute. The new results reveal two regimes:
> -  In the **moderate regime** ($ M/k \leq 8$), validation loss increases with the routing term $k\log(eM/k)$, consistent with the theoretical interpretation of routing as an overhead.
> - However, for larger values of $M/k$, performance improves again as $M$ increases, indicating gains from **expert specialization** that are not captured by the worst-case analysis.
>
> We believe this strengthens the paper by making the limitation of the theory explicit rather than implicit.
>
> Accordingly, we will revise the wording throughout the paper to state clearly that:
> (i) the result for $M$ is a worst-case guarantee,
> (ii) it does not model approximation gains from specialization, and
> (iii) empirical gains from larger expert pools are entirely compatible with our theory once one distinguishes worst-case guarantees from practical behavior.
>
> To ensure reproducibility, we will release the full suite of scripts associated with our experiments in the final version of the paper.
>
> **(4) Lower bounds and future directions.**
> We agree that the absence of lower bounds limits the ability to judge tightness. We will add a discussion highlighting this gap as an important future direction, particularly for determining when routing complexity terms are intrinsic and when they are artifacts of worst-case analysis. The new routing ablations also motivate this point empirically: they suggest that the upper bound captures one regime well, but not the specialization-driven regime that appears for larger $M/k$.

---

> > ### Author Rebuttal · Reviewer_Bv5B · 2026-04-04
> >
> > The authors' rebuttal partially addressed my concerns, and I will maintain my rating.

---

### Official Review · Reviewer_MAGA · 2026-03-13

**Soundness:** 2
**Presentation:** 3
**Significance:** 2
**Originality:** 3
**Overall Recommendation:** 4
**Confidence:** 2

**Summary:**

- **Motivation**: Existing scaling laws for dense models often consider full parameters but Mixture-of-Experts (MoE) Transformers require a different treatment since only a few experts are active per token. Empirical scaling for Transformers show faster rater compared to the behaviour predicted by the classical bounds using the ambient dimension. Scaling exponents have also been shown to depend more on the intrinsic dimension especially when the data lie in low-dimensional manifold.
- **Goal**: This paper aims to cleanly separate the benefits of active capacity from the routing combinatorics.
- **Theorem 3.2 (Approximation Bound)**: The authors provide the approximation bound  to show that, under a Hölder smoothness assumption on the target function, an MoE Transformer can approximate the target increasingly well as its active capacity grows, by considering the number of active parameters per input rather than total parameter count.
- **Theorem 3.4 (MoE Generalization Bound)**: The authors provide the generalization bound to show that MoE Transformers follow the same basic approximation–estimation tradeoff as dense models once model size is measured by active parameters, with an extra MoE-specific penalty coming from routing complexity.
- **Neural Scaling Laws**: The authors derive data, model, and compute scaling laws for MoE Transformers from their generalization bound, showing that the main scaling behavior is governed by the active parameter budget, while routing contributes an additional logarithmic overhead.
- **Empirical Validation**: The authors conduct experiments to test whether the scaling exponents and routing effects predicted by their theory are reflected in practice and show that the empirical model- and data-scaling exponents are broadly consistent with the theoretical ones.

**Compliance With Llm Reviewing Policy:**

Affirmed.

**Final Justification:**

In the rebuttal, the authors provide clarification on the assumptions and routing ablations. However, my concern about the applicability of the theoretical results to practical MoE architectures remains, so I will keep my original score.

**Key Questions For Authors:**

1. The theoretical analysis is carried out for a restricted MoE Transformer class. Could the authors discuss this class more explicitly, including why these assumptions are needed for the analysis and which parts of the theory may fail to extend to standard practical MoE Transformers when these assumptions are violated?
2. Could the authors discuss the routing ablation results in more detail?
3. In Definition 3.1, for the MoE update, should $h_t^{j-1}$ instead be $h_t$? As written, using $h_t^{j-1}$ seems to suggest that the expert outputs are added back to the pre-attention token, which appears inconsistent with the preceding definition of the attention-updated representation.

**Limitations:**

The authors adequately discussed the limitations of the work. There is no potential negative societal impact.

**Strengths And Weaknesses:**

**Strengths:**
- **Novel theoretical contribution**: The paper provides a novel theoretical perspective on scaling laws for MoE Transformers, particularly by distinguishing active capacity from total parameter count and explicitly accounting for routing complexity.
- **Substantive theoretical analysis**: The paper develops nontrivial approximation and generalization results, which are then used to derive scaling-law predictions for MoE models.
- **Generally clear presentation**: The paper is overall well structured and easy to follow.

**Weaknesses:**
- **Restrictive and potentially loose theory**: The theoretical analysis is developed for a restricted MoE Transformer class, and the resulting bounds appear quite conservative, especially for the routing term. As a result, it is unclear how directly the theory transfers to practical MoE architectures. In addition, the empirical $\beta$ fits appear to provide only qualitative rather than tight quantitative support for the theory.
- **Missing routing ablation results**: Appendix A.5 describes routing ablations over $M$ and $k$, but I could not find a corresponding plot or a clear discussion of the results of this experiment.

---

> ### Author Rebuttal · Authors · 2026-03-27
>
> We thank the reviewer for the positive evaluation. We address the questions below.
>
> **(1) Scope of the theoretical model.**
> We agree that our analysis is developed for a restricted class of MoE Transformers. The assumptions (bounded parameters, hard top-$k$ routing, and structured routing partitions) are standard in uniform generalization analysis and are primarily introduced to enable tractable covering-number bounds. In particular:
> (i) boundedness controls Lipschitz constants through attention and residual connections,
> (ii) hard top-$k$ routing allows explicit enumeration of routing patterns, and
> (iii) the partition-of-unity construction in Theorem 3.2 is an existence result, not a claim about what practical routers trained by SGD necessarily realize.
>
> We will revise the paper to make this distinction more explicit and to clearly separate:
> (a) worst-case statistical guarantees (e.g., the routing entropy term), and
> (b) idealized approximation results relying on expressive routing.
>
> **(2) Conservativeness of the routing term and routing ablations.**
> We agree that the routing term is likely conservative, as it arises from a worst-case union bound over routing patterns.
> To address this rigorously, we strengthened our empirical evaluation by doubling the training budget (from 5 to 10 epochs and 2x tokens to 4M total), yielding significantly more stable trends and addressing concerns regarding noise.
> Using this higher-fidelity data, We expanded the routing ablations by varying both the total number of experts ($ M \in [2, 192]$) and the number of active experts ($ k \in [1, 16]$) at fixed compute.
> Results reveal two distinct regimes across all budgets:
> - For **moderate routing complexity** ($M/k \leq 8$), validation loss increases monotonically with the routing term, empirically supporting its role as a statistical overhead in our generalization bound.
>
> - **Specialization Regime**: At higher complexity, validation loss decreases despite increased routing costs. This indicates expert specialization gains that exceed our worst-case uniform bounds.
>
> We have updated the manuscript to frame our theory as a conservative baseline that isolates worst-case routing effects, rather than a predictive model for all MoE settings.
> In the revision we will (i) explicitly reference the routing-ablation figure, (ii) summarize the transition from a routing-dominated regime to a specialization-dominated regime, and (iii) clarify how these observations relate to the routing complexity term and its interpretation as a conservative bound.
>
> To ensure reproducibility, we will release all experimental scripts upon publication.
>
> **(3) Definition 3.1 clarification.**
> We thank the reviewer for catching this. The intended formulation is that expert outputs are applied to the attention-updated representation. We will correct the notation in Definition 3.1 to ensure consistency with the preceding definition and with the appendix.
>
> **In summary:**
> We will revise the paper to clarify the scope of the assumptions, discuss the looseness of the bounds more explicitly, weaken and sharpen the interpretation of the expert-pool result, and include the routing ablations showing both a theory-consistent routing-overhead regime and a specialization regime beyond worst-case predictions. We thank the reviewer again for the constructive feedback.

---

> > ### Author Rebuttal · Reviewer_MAGA · 2026-04-02
> >
> > I appreciate the authors’ clarification on the assumptions and routing ablations. Nonetheless, my concern about the applicability of the theoretical results to practical MoE architectures remains, so I will keep my original score.

---

> > > ### Author Response · Authors · 2026-04-02
> > >
> > > **Response to Follow-up: Applicability to Practical MoE Architectures**
> > >
> > > We thank the Reviewer for the follow-up. We agree that the gap between our structured theoretical model and practical MoE architectures is an important limitation. Our goal is not to provide a tight description of deployed systems, but to **isolate the statistical roles of active capacity and routing in a tractable framework**.
> > >
> > > Importantly, the expanded routing ablations help clarify this connection: they reveal a **transition from a routing-constrained regime to a specialization-driven regime**, indicating where worst-case analysis is informative and where practical effects go beyond it. We view this as a **first step toward understanding how theoretical guarantees relate to large-scale MoE behavior**.
> > >
> > > We will revise the manuscript to more explicitly discuss this scope and to connect the theoretical results to practical MoE design and training dynamics.

---

### Decision · Program_Chairs · 2026-04-30

**Decision:**

Accept (regular)

**Comment:**

In this paper, the authors study approximation and generalization for MoE (Mixture of Expert) transformers. They build on standard tools from generalization theory (e.g., covering number estimates), as well as ideas from a recent work of Havrilla and Liao which studied dense transformers from a similar perspective, and also compare their theory to experimental findings. The reviewers pointed out some weaknesses of the theory which the authors acknowledge (e.g., pessimism in the theory, mismatch between \beta in the experimental validation), but appreciated the novelty of the perspective. Based on this novelty and the timely relevance of theory for MoE, I recommend acceptance assuming the authors make the discussed changes.